# Decomposing Representation Space into Interpretable Subspaces with Unsupervised Learning

**Xinting Huang**   **Michael Hahn**
Saarland University
{xhuang, mhahn}@lst.uni-saarland.de

## ABSTRACT

Understanding internal representations of neural models is a core interest of mechanistic interpretability. Due to its large dimensionality, the representation space can encode various aspects about inputs. To what extent are different aspects organized and encoded in separate subspaces? Is it possible to find these "natural" subspaces in a purely unsupervised way? Somewhat surprisingly, we can indeed achieve this and find interpretable subspaces by a seemingly unrelated training objective. Our method, neighbor distance minimization (NDM), learns non-basis-aligned subspaces in an unsupervised manner. Qualitative analysis shows subspaces are interpretable in many cases, and encoded information in obtained subspaces tends to share the same abstract concept across different inputs, making such subspaces similar to "variables" used by the model. We also conduct quantitative experiments using known circuits in GPT-2; results show a strong connection between subspaces and circuit variables. We also provide evidence showing scalability to 2B models by finding separate subspaces mediating context and parametric knowledge routing.[1] Viewed more broadly, our findings offer a new perspective on understanding model internals and building circuits.

## 1  INTRODUCTION

Mechanistic Interpretability research aims to understand the internal mechanisms of neural networks. Unlike behavioral research, the community seeks deeper algorithmic descriptions or circuits of models. Insights from this kind of research can serve our purpose of building safer, more controllable, and more reliable AI systems (Bereska & Gavves, 2024), and making progress in other scientific fields by learning how data is modeled by these models (e.g. Futrell & Mahowald, 2025; Oota et al., 2023; Celeghin et al., 2023).

Researchers have made significant progress in this direction in recent years, uncovering circuits consisting of various types of basic units (or mediators (Mueller et al., 2024)), including components (attention heads and MLPs) of transformers (Wang et al., 2022; Hanna et al., 2023), sparse features obtained from Sparse Auto-Encoders (SAEs) or its variants (Cunningham et al., 2023; Bricken et al., 2023; Dunefsky et al., 2024; Ameisen et al., 2025), and subspaces (Geiger et al., 2024; Wu et al., 2023; Geiger et al., 2025). While providing valuable insights, they are limited in different ways. The information transferred between components is hard to understand, and researchers need to design specialized counterfactual data targeting certain aspect of the input to perform causal intervention, or use easy but much less reliable methods such as observing attention weights or projecting to vocabulary space. Sparse features provide a direct way to interpret (by reading from the inputs they activate on), but resulting circuits are input-dependent, in other words, specific for each input sequence, as different features are activated in different inputs. This approach also usually modifies the model's computation, replacing activations or MLPs with reconstruction from sparsity-encouraged new modules, in order to better analyze feature-to-feature interactions. Lastly, current subspace-based approaches require supervision from an abstract causal model specified by human. The abstract

---

[1] Link for code and trained matrices https://github.com/huangxt39/SubspacePartition

model represents a clear hypothesis of how the real model works, thus requiring anticipating possible mechanisms. We refer to Mueller et al. (2024) for a good summary of different mediators.

In this paper, we explore a new direction: unsupervised representation space decomposition. We show we can learn a partitioning of representation space by doing neighbor distance minimization (NDM) among subspace activations. We explain the intuition behind this training objective in terms of feature groups and mutual information and show that it looks for a partition such that subspaces are as independent as possible. Similar to Geiger et al. (2024), an orthogonal matrix is learned, which rotates and reflects the space before partitioning on the transformed basis. It captures the distributed nature of neural representations. While they aim to find the subspace for certain target variables by supervised learning, our method is unsupervised and makes use of "natural" structure inside the model. We first demonstrate the effectiveness of NDM in a toy setting, then move on to real language models. Somewhat surprisingly, we find that there is indeed meaningful inner structure of representation space in real-world models. We show that resulting subspaces are usually interpretable with InversionView (Huang et al., 2024b). We quantitatively test alignment between subspaces and variables in known circuits in GPT-2 Small using causal intervention. Moreover, we demonstrate its applicability on larger 2B models by showing that NDM finds separate subspaces mediating context knowledge and parametric knowledge.

Our method also points out a potential future direction: subspace circuits. Subspaces given by our method are suitable as new basic units for circuit analysis, as we can directly analyze model weights to understand, for example, which subspaces a certain attention head reads from and writes to. By building connections between subspaces across layers with model weights, we could construct *input-independent* circuits, as the space decomposition and model weights are input-independent. In these circuits, subspaces are like variables, each of which encapsulates a group of features it can take as values. Therefore, we believe our method and the possibilities it enables will benefit the interpretability community.

## 2 MOTIVATION AND BACKGROUND

One of the key concepts in interpretability literature is superposition, which means a model can represent more features than its dimensionality by representing them as non-orthogonal directions. The phenomenon is well-illustrated in (Elhage et al., 2022), where they use a toy model as follows:

$$\boldsymbol{h} = \mathbf{W}\boldsymbol{x} \qquad\qquad \boldsymbol{x}' = \mathrm{ReLU}(\mathbf{W}^T\boldsymbol{h} + \boldsymbol{b}) \qquad\qquad (1)$$

where $\boldsymbol{x}, \boldsymbol{x}' \in \mathbb{R}^z$, $\boldsymbol{h} \in \mathbb{R}^d$, $\mathbf{W} \in \mathbb{R}^{d \times z}$, $d < z$. The high-dimensional $\boldsymbol{x}$ is the ground truth feature vector, each entry $x_i$ represents a "feature" and its value ranges in [0, 1], it is encoded in low-dimensional $\boldsymbol{h}$. The model output $\boldsymbol{x}'$ is trained to reconstruct $\boldsymbol{x}$. So each column in $\mathbf{W}$ corresponds to a direction in the lower-dimensional space that represents a feature $x_i$.

In this toy setting, (Elhage et al., 2022) found that **sparsity** is necessary to allow superposition to occur. In this paper, we argue that **mutual exclusiveness** could be a more fundamental condition. More specifically, for a group of features, if only one (or none) occurs at a time, they are said to be mutually exclusive, and can be represented with superposition without much loss of information. Although sparsity encourages mutual exclusiveness, they are different, since the former concept treats features uniformly while the latter emphasizes a non-uniform view under which some sets of features can be more dependent on each other.

If mutual exclusiveness holds, superposition works well; the only discrepancy between $\boldsymbol{x}$ and $\boldsymbol{x}'$ is because of small interference between almost orthogonal features. On the contrary, if multiple features are activated, they might cancel each other in certain directions, causing loss of information, so that reconstructed $\boldsymbol{x}'$ can be totally different from $x$ (Fig. 6a and 6b in App.). In other words, *given two or more specific features*, if they often co-occur, superposing *these features* would not work well – incentivizing the model to represent them more orthogonally. On the other hand, if they are mutually exclusive (because the concepts they represent are mutually exclusive in the real world), the model would tend to make use of this fact and superpose their features.

**Feature Groups and Subspaces**    It would be more interesting if we consider groups of mutually exclusive features. Features are mutually exclusive with features in the same group but independent of features in different groups. So each model activation $h$ can result from multiple activating features, but each of them comes from a different group. As we discussed earlier, features in the same group tend to be superposed, while co-occurrence would encourage features from different groups to be more orthogonal. Therefore, in such an ideal case, each feature group can form a dedicated subspace, superposition happens inside the subspace, but subspaces are orthogonal to one another.

One might ask whether the mutual exclusiveness property is likely to happen in the real world. We think it is actually pervasive. For example, if the previous token is "the", it cannot be any other tokens in the vocabulary; if the subject of a sentence is "Alice", it cannot be "Bob"; if the sum of two digits is 5, it cannot be simultaneously other numbers. In model's processing, like an algorithm, it might have "variables" (like those searched by DAS (Geiger et al., 2024)) to encode certain aspects of information, and each variable can take only one value at a time, and each corresponds to a feature group, or a subspace. So we think such a mutual-exclusiveness property and group structure can be very common in the model, and there are "natural" partitions of orthogonal and interpretable subspaces. In the following section we show evidence in toy setting, and in Sec. 5, we show evidence in real neural language models. In addition, we find our notion of feature groups quite relevant to Multi-Dimensional Superposition Hypothesis (Engels et al., 2024), each feature group can be thought of as one multi-dimensional irreducible feature in this hypothesis, because of intra-group dependence and inter-group independence.

**Toy Model of Intra-group Superposition**    In the toy setting (Elhage et al., 2022) where we have access to real features, we can straightforwardly verify superposition and orthogonality. We use the aforementioned toy model (Eq. 1). As a start, we set $z = 40$, $d = 12$, and features are split into two groups, 20 for each. When sampling $x$, for each group, 0.25 of the time no feature is activated, 0.75 of the time one feature is randomly chosen and its value is sampled uniformly from [0, 1]. Features are equally important (not weighted when computing reconstruction loss).

After training, fraction of variance unexplained (FVU) is 0.059, and $\mathbf{W}^T\mathbf{W}$ is shown in Fig. 1a. We see that products of features coming from the same group are non-zero, while features in different groups are always orthogonal to each other. There are two orthogonal subspaces containing the two feature groups, respectively. Closer inspection of singular values shows that each feature group is encoded in 5-dimensional space. In App. C.1 we show different configurations of feature groups and the resulting $\mathbf{W}^T\mathbf{W}$. In sum, we can expect similar orthogonal subspaces when features follow the aforementioned group structure.

## 3    METHODOLOGY

If orthogonal structure exists, i.e., orthogonal partition of representation space such that each subspace contains one feature group, how can we find these subspaces? In toy setting, we can take all feature vectors from the same group, and then the subspace spanned by these is what we are looking for. But we aim to create a method that works for real-world models, i.e., without access to ground truth features $x$ or projection $\mathbf{W}$. We need a method that only requires model activations $h$. In this section, we describe our method, NDM, which finds the correct subspace given *only* model activations.

**Intuition**    The intuition behind our method is illustrated in Fig. 1b. In a 3D space, assuming xy plane contains a group of 3 features (blue arrows), and z dimension contains another group of 2 features (orange arrows), so there are 2 subspaces orthogonal to each other and each contains a feature group. If intra-group mutual exclusiveness holds, data points will only lie in the pink planes, as each point is a sum of only one blue and one orange. Given the feature groups, the correct partition is thus (xy, z). If the 3D space is partitioned in this way, the data points' projection on the xy subspace only lies on blue arrows, and similarly for the z subspace. Thus, given some data points, inside each subspace (consider their projection) points are concentrated on a few lines, and it is easy for each point to find another point in a close distance. However, if it is partitioned into xz and y, then the data points, after being projected to xz plane, cover the entire plane, causing larger distance between each point and their neighbor. In other words, the correct partition will make the distance to the nearest neighbor inside subspaces smaller, whereas incorrect partitions entail large distances because they arbitrarily combine features from different groups.

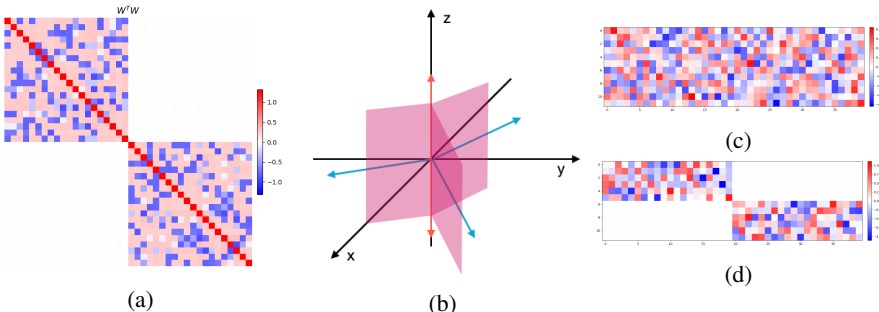

Figure 1: **(a)**: $\mathbf{W}^T\mathbf{W}$, $\mathbf{W} \in \mathbb{R}^{12 \times 40}$ (Sec. 2). Each column of $\mathbf{W}$ is a feature direction, its first 20 columns belong to one feature group, the last 20 to the other. We can see the bottom left and top right parts are zero, which means feature vectors of different groups are orthogonal. **(b)**: Illustration for the intuition of our method (Sec. 3), correct partition can make data points easier to model. **(c) (d)**: $\mathbf{RW}$ before and after NDM training (Sec. 4). The final dimension configuration is $d_1 = 6, d_2 = 6$. The corresponding toy model is shown in (a). After training, the first subspace (first 6 coordinates) only depends on the first feature group (first 20 coordinates), and likewise for the other subspace.

**Optimization** We define a *partition* as a tuple consisting of an orthogonal matrix $\mathbf{R}$ and a dimension configuration $c$ (number of subspaces and their dimensionality). $\mathbf{R}$ rotates and reflects the space as needed (similar to (Geiger et al., 2024)), then the transformed space is partitioned on its standard basis (simply grouping coordinates) according to $c$. We optimize $\mathbf{R}$ as follows: given $N$ model activations $[\boldsymbol{h_1} \cdots \boldsymbol{h_N}]$ where $\boldsymbol{h_n} \in \mathbb{R}^d$, and an orthogonal matrix to be trained $\mathbf{R} \in \mathbb{R}^{d \times d}$ [2], dimension configuration $c = [d_1, \cdots, d_S]$, where $S$ is the number of subspaces, $\sum_{s=0}^{S} d_s = d$, we do

$$[\hat{\boldsymbol{h}}_1 \cdots \hat{\boldsymbol{h}}_n] = \mathbf{R}[\boldsymbol{h_1} \cdots \boldsymbol{h_n}], \qquad \hat{\boldsymbol{h}}_n = \begin{bmatrix} \hat{\boldsymbol{h}}_n^{(1)} \\ \vdots \\ \hat{\boldsymbol{h}}_n^{(S)} \end{bmatrix}, \quad \hat{\boldsymbol{h}}_n^{(s)} \in \mathbb{R}^{d_s} \tag{2}$$

$$n^* = \underset{\substack{m=1,\ldots,N \\ m \neq n}}{\arg\min} \ \mathrm{dist}\left(\hat{\boldsymbol{h}}_n^{(s)}, \hat{\boldsymbol{h}}_m^{(s)}\right) \tag{3}$$

$$\min_{\mathbf{R}} \frac{1}{N} \sum_{s=1}^{S} \sum_{n=1}^{N} \mathrm{dist}\left(\hat{\boldsymbol{h}}_n^{(s)}, \hat{\boldsymbol{h}}_{n^*}^{(s)}\right) \quad \text{s.t. } \mathbf{R}^\top \mathbf{R} = \mathbf{I} \tag{4}$$

where the function $\mathrm{dist}(\cdot)$ is the distance metric we can do experiment with. So we optimize the orthogonal matrix to minimize the distance with the nearest neighbor in subspaces.

Besides the aforementioned perspective in terms of feature superposition, there is also a heuristic interpretation of our method as **minimizing total correlation** between subspaces, which generalizes mutual information (MI)'s definition from two variables to multiple variables and measures the redundancy and dependence among them. Loosely speaking, neighbor distance reflects the entropy; our method reduces entropy in subspaces. Entropy of the whole space is invariant under orthogonal transformation; thus, total correlation is reduced (details in App. B.1). In other words, we look for a partition of representation space such that the resulting subspaces are as independent as possible, thus they are suitable to serve as the elementary units of interpretation. This MI/total correlation perspective justifies our method even when the previous feature group hypothesis does not strongly or strictly hold in real neural models.

**Determining the Subspace Dimension Configuration** However, there is still a question: how to find the correct dimension configuration (number of subspaces and their dimensions)? It is not

---

[2]Libraries like Pytorch (Paszke et al., 2019) contain parametrization to ensure orthogonality of matrix

differentiable and is unclear what is the appropriate objective to optimize for. In our method, we use MI to address this problem. The procedure is as follows: we start with a configuration of small and equal-sized subspaces and train the orthogonal matrix. We measure the MI between subspaces regularly during training using KSG estimator (Kraskov et al., 2004). If the MI[3] is higher than a threshold, we merge the two subspaces, so the nearest neighbor should be found in the joint subspace afterwards and keep training the orthogonal matrix. We stop training when we cannot find a pair of subspaces whose MI is above the threshold. This procedure is in line with our goal of reducing the dependency between subspaces, as we merge them if their MI is unavoidably high. We describe our method in detail in Algorithm 1 and App. B. We also experimented with other methods for determining the configuration, see App. G.4. In sum, NDM finds a partition containing both $\mathbf{R}$ and $c$.

## 4    EXPERIMENTS IN TOY MODELS

We first test our method in the toy model setting described in Sec.2, using a slightly different and less scalable version of Algorithm 1 (see details in App. C.2). We use Euclidean distance as the $\text{dist}(\cdot)$ in Eq. 3 and 4, because we find cosine similarity to perform worse in preliminary experiments. The toy model setting is an ideal testbed as we have access to the ground truth feature vectors and subspaces. As described, we rotate and reflect the hidden activations $\boldsymbol{h}$ from Eq.1, which results in $\hat{\boldsymbol{h}} = \mathbf{RW}\boldsymbol{x}$. Subspace activations, which are defined to be a continuous span of coordinates of $\hat{\boldsymbol{h}}$, should ideally result from only one feature group. Therefore, a good $\mathbf{R}$ means rows in $\mathbf{RW}$ corresponding to one subspace (subvectors of $\hat{\boldsymbol{h}}$) have nonzero values only at columns corresponding to one feature group (subvectors of $\boldsymbol{x}$). So, the important question is, can we find subspaces encoding each feature group by minimizing distances? Despite our argument before, this still sounds uncertain a priori, as we do not optimize $\mathbf{R}$ to associate subspaces with underlying groups but rather optimize the *indirect* objective of neighbor distance, as we want to avoid using $\mathbf{W}$ and $\boldsymbol{x}$. Somewhat beyond our expectation, the answer is yes, we find good subspaces by using NDM.

The result is shown in Fig. 1d. We initialize $\mathbf{R}$ as an identity matrix, and the $\mathbf{RW}$ after training is almost perfect. We present more experiments in Fig. 10-13 in App. C.2, where numbers of features and groups vary. The results overall show a strong performance of NDM, we are able to find the subspace partition close to the ground truth in all cases. In sum, in the toy setting where there exist orthogonal subspaces dedicated to each group, we indeed can find these subspaces by NDM.

## 5    EXPERIMENTS IN LANGUAGE MODELS

The key question is NDM's applicability to real-world neural models. Unlike the toy setting, we do not know ground-truth features, so we design new ways to do evaluation. In this section, we denote residual stream activations as $\boldsymbol{h}^{l,\{mid,post\}}$, where $l$ is the layer index (starting from 0), $mid$ means after adding the attention sublayer's output, and $post$ means after adding the MLP sublayer's output.

### 5.1    QUANTITATIVE EVALUATION BASED ON GPT-2 CIRCUITS

The evaluation is based on subspace activation patching, or subspace interchange intervention (Geiger et al., 2024). The main intuition is that, when processing inputs, key intermediate results should ideally lie in a single subspace. For example, in an induction head circuit where the residual stream contains the previous token, the previous token should be placed in a single subspace, independent of which token it is from. Therefore, if we patch subspace activations with the value they take in sequences with different previous tokens, the effect should be concentrated on that correct subspace. We refer to the previous token as the target information, which is changed in the counterfactual input. Different target information should ideally be concentrated in different subspaces. We describe subspace patching in detail in App. D.1 and address potential concerns from (Makelov et al., 2023) in App. G.2. In sum, we aim to explore whether the high-level "variable" lies in the same subspace no matter what value it takes by measuring inequality or concentration of subspace patching effect.

**Test Suite and Metric**    We construct a test suite, making use of known circuits in GPT2 Small. It consists of 5 tests. The first 4 tests are based on the IOI circuit (Wang et al., 2022), where we intervene

---

[3]In practice, we divide the mutual information value by the sum of the two subspaces' dimensions.

| Method | test 1 | | | test 2 | | | test 3 | | | test 4 | | | test 5 | | | Avg |
|---|---|---|---|---|---|---|---|---|---|---|---|---|---|---|---|---|
| | - | $d_s$ | $\text{Var}_s$ | - | $d_s$ | $\text{Var}_s$ | - | $d_s$ | $\text{Var}_s$ | - | $d_s$ | $\text{Var}_s$ | - | $d_s$ | $\text{Var}_s$ | |
| Identity | 0.33 | 0.05 | 0.23 | 0.32 | 0.11 | 0.25 | 0.40 | 0.12 | 0.19 | 0.31 | 0.04 | 0.15 | 0.32 | 0.12 | 0.19 | 0.21 |
| Random | 0.36 | 0.10 | 0.11 | 0.36 | 0.16 | 0.16 | 0.32 | 0.11 | 0.12 | 0.33 | 0.10 | 0.12 | 0.39 | 0.16 | 0.18 | 0.21 |
| PCA 1 | 0.43 | 0.42 | 0.37 | 0.46 | 0.22 | 0.52 | 0.50 | 0.30 | 0.51 | 0.38 | 0.22 | 0.36 | 0.35 | 0.24 | 0.36 | 0.38 |
| PCA 2 | 0.66 | 0.56 | 0.39 | 0.26 | 0.19 | 0.24 | 0.28 | 0.19 | 0.29 | 0.40 | 0.20 | 0.19 | 0.40 | 0.26 | 0.27 | 0.32 |
| Feature | 0.60 | 0.44 | 0.60 | 0.61 | 0.41 | 0.67 | 0.73 | 0.39 | 0.54 | 0.74 | 0.31 | 0.54 | 0.71 | 0.56 | 0.61 | 0.56 |
| NDM | 0.89 | 0.89 | 0.90 | 0.71 | 0.72 | 0.83 | 0.63 | 0.70 | 0.38 | 0.50 | 0.55 | 0.79 | 0.72 | 0.67 | 0.78 | **0.71** |

Table 1: Results on GPT-2 Small test suite, higher is better. Test 1-5 measure the concentration of various target information in various places in GPT-2 Small (details in App. D.2) based on ground truth circuits (Wang et al., 2022; Hanna et al., 2023). Numbers are Gini coefficient over subspace patching effects ("-"), over effects normalized by subspace dimension ("$d_s$"), and over effects normalized by subspace variance ("$\text{Var}_s$"). Last column is the average values over the whole row. We compare NDM with 5 baselines. An identity matrix ("Identity"); a random orthogonal matrix ("Random"); PCA 1 and 2 are matrices composed of eigenvectors after applying PCA on activations. We use dimension configuration $[d_1, \ldots, d_S]$ of NDM result as the configuration for all baselines. $[d_1, \ldots, d_S]$ is sorted in descending order, eigenvectors can be sorted in either way according to eigenvalues, resulting in PCA 1 and 2 – meaning either that the smallest subspace receives the biggest ("PCA 1") or smallest ("PCA 2") directions of variance; and subspaces obtained by clustering SAE features ("Feature"), using a method adapted from (Engels et al., 2024)) (details are in App. F.2). Importantly, as shown in App. D.4, the degree of concentration of target information becomes satisfactory when Gini coefficient $> 0.6$.

on the following pieces of information respectively: (1) previous token information contained in layer 4 residual stream (2) subject position in layer 6 (3) subject position and (4) subject name in layer 8 residual stream. The last test is based on Greater-than circuit (Hanna et al., 2023): (5) information about the last two digits of the starting year in layer 9. We measure patching effect as the change in logit difference for IOI-based tests and change in valid answer probability for the Greater-than-based test. See more details in in App. D.2. We then quantify the inequality of patching effect using Gini coefficient. Though very unconventional in machine learning, we find Gini coefficient a suitable metric. Because we want to measure how strongly patching effect is focused on just one or a few subspaces, this notion is captured by Gini coefficient as it measures inequality or degree of concentration. If the target information is mainly encoded in one subspace, the patching effect for this one would be much higher than others, thus resulting in Gini coefficient close to 1. On the other hand, evenly distributed patching effect results in 0. In App. D.4 we show examples of patching effect distributions and their corresponding Gini coefficients, from which we conclude that the coefficient should be $> 0.6$ to be considered satisfactory (i.e., the effect is mainly concentrated in one subspace). We also calculate the Gini coefficient over patching effect normalized by dimension $d_s$ and variance $\text{Var}_s$ of subspaces.

**Results**   We apply NDM to find subspace partitions in GPT-2 Small (Radford et al., 2019), details are described in App. D.3. The results of NDM using the best hyperparameters are shown in Table 1, together with 5 baselines, which, like NDM, are unsupervised and purely computed from activations. *As we can see, NDM clearly outperform baselines*. Though there are a few cases where Gini coefficient is low for NDM, the overall average value shows a strong concentration of target information (0.71, significantly higher than 0.6). Figure 22, 29 and 30 show the qualitative examples of the subspaces obtained by NDM with the highest effect in Test 1, 2, 5 respectively (See later section for how to read them). In App. D.5 we show ablation studies, providing experimental data on how different factors affect final results. Meanwhile, the estimated MI between subspaces decreases significantly after training (Fig. 15), matching the explanation of NDM's training objective.

## 5.2   QUALITATIVE ANALYSIS OF OBTAINED SUBSPACES IN GPT-2

**Preliminaries**   (Huang et al., 2024b) show that we can interpret activations in terms of their *preimage*, which is the set of inputs that are mapped to the same activation by the model. Simply speaking, to interpret an activation (a vector, called query activation) at a certain place in the model (e.g., layer 3 residual stream, called activation site), we find those inputs that, when fed into the model, would give rise to roughly the same activation at that activation site. Therefore, when query activation is observed by downstream components, any input in preimage could be the current input,

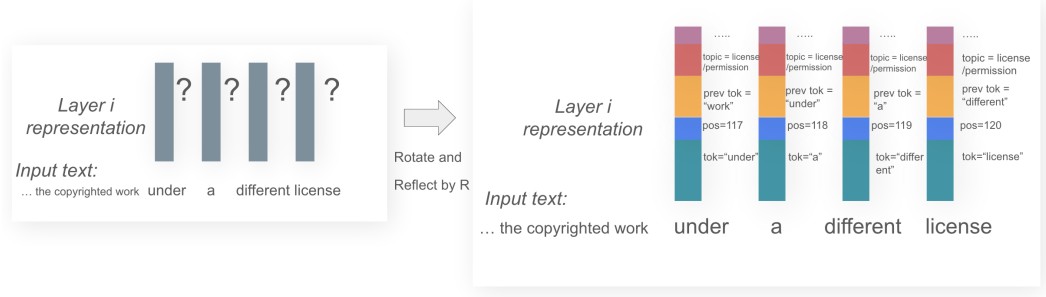

Figure 2: Illustration of our subspace decomposition. On the right, the interpretation of the column of token "a" is obtained by inspecting Figure 3. Likewise, each row from bottom to top corresponds to Figure 21, 23, 22, 24, respectively.

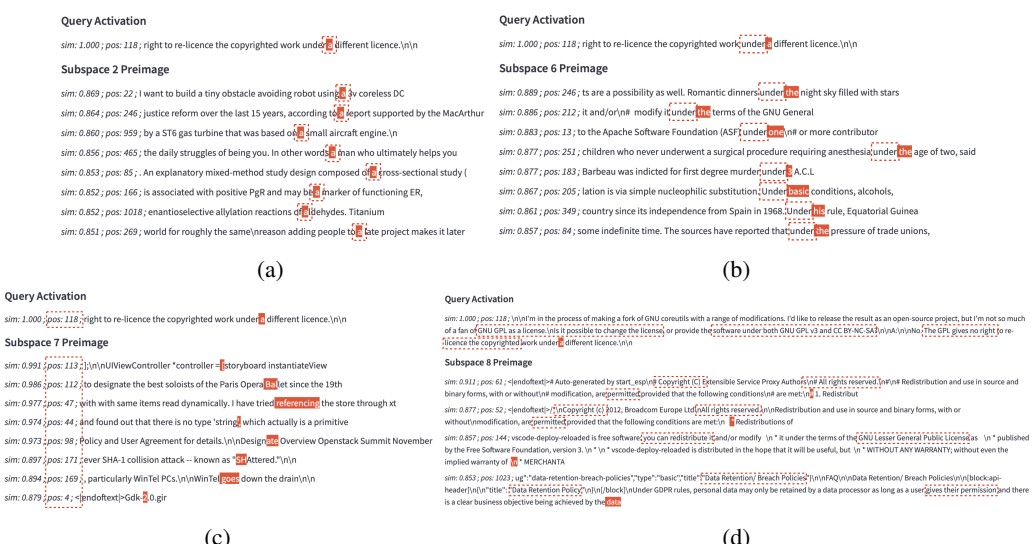

Figure 3: Preimages of the subspace activation corresponding to the same token "a" in 4 different subspaces of the residual stream $h^{4,post}$ in GPT-2 Small. Tokens highlighted with red background are those where the activations are taken from. Tokens highlighted with dashed boxes reflect the encoded information. These activations are projected into the subspaces and the first column "*sim*" is the similarity between this projection in the preimage and in the query activation. The second column "*pos*" is the position indices of the highlighted tokens. We are showing only the most recent part of the context and as well as 5 future tokens, the actual input sequences are much longer. Interpretation of encoded information: (a) current token "a". (b) the prior token "under". (c) current position, around 118. (d) the topic, the overall context is about licence/permission. In sum, we can see in different subspaces of the same layer, different aspects of the current context are encoded.

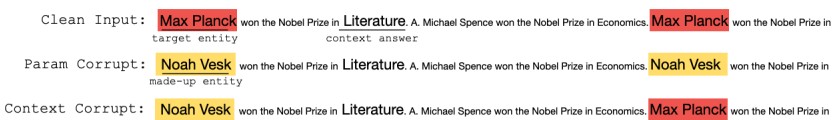

Figure 4: Clean and counterfactual inputs used in knowledge conflict experiments. Patching with activation from `Param Corrupt` can cause the model to increase the probability of context answer, while patching with `Context Corrupt` can increase the probability of parametric answer.

thus the commonality in preimage is the available information. In other words, query activation (any activation we want to interpret) "represents" the whole preimage.

**Setting**    Instead of the whole space, we apply it to subspaces. (See discussion in App. G.1 for why it works better for subspaces). Instead of training decoders as in (Huang et al., 2024b), we collect around 1M model activations and conduct vector search among them to find preimages (see details in App. E.1). Though without a decoder the content of preimage is restricted by a fixed set of inputs (2000 sequences from MiniPile + 2000 IOI and greater-than sequences), it is easier to implement. We use cosine similarity to define the preimage, so it includes inputs whose corresponding activations are in a similar direction to the query activation. We set 0.85 as the threshold for preimages in most cases. For all qualitative results in the paper (including the web application), the subspace partition is given by the best configuration of NDM according to the test results from Sec. 5.1 (last row of Table 1).

**Results**    Fig. 2 illustrates our findings. In Fig. 3 we show 4 preimages corresponding to the same token at the same layer's residual stream. We see clearly that in this contextualized representation, different aspects of the context are encoded in different subspaces, i.e., the current token, position, prior token, the topic. NDM indeed decomposes the representation space in a meaningful way. Moreover, we also find that the function/interpretation of subspaces is *consistent*. For example, Fig. 3(a) shows subspace 2 encodes the current token, and we find it always encodes the current token. This is also true for the other 3 subspaces shown in the other 3 subfigures (See Fig. 21 to 24 in App.). Though being consistent, we also find that the meaning of subspaces can vary, depending on factors such as whether a certain pattern occurs or a certain mechanism is triggered. We provide an extensive discussion for the qualitative results in App. E.3. In sum, qualitative results show that most of the time, the preimages are interpretable, and we can find a high-level concept to summarize the encoded information in subspaces across inputs because of good consistency. Due to space limits we only present a very limited number of examples, we strongly recommend checking more in our web application[4].

## 5.3   APPLICABILITY TO LARGER MODELS

To test the applicability on larger models, we apply NDM on residual stream activations of Qwen2.5-1.5B (Qwen et al., 2025) and Gemma-2-2B (Team et al., 2024)[5]. As no very detailed ground-truth circuits are available for them, we design prompts that are likely to elicit separate mechanisms inside the model, and see if they are mediated by separate subspaces. This would constitute strong evidence that our partition is meaningful.

**Setting**    We construct inputs that elicit knowledge conflicts: "Max Planck won the Nobel Prize in Literature. ... Max Planck won the Nobel Prize in". The model can either predict "Physics" (parametric answer) by using its parametric knowledge, or predict "Literature" (context answer) by copying the context knowledge. We refer "Max Planck" as target entity. We construct examples following this form with different target entities, using RAVEL dataset (Huang et al., 2024a). We do not use the dataset as the way it is designed, see App. D.6 for discussion and experimental details. To test whether a subspace plays a role in either of these two circuits quantitatively, we again use subspace patching and construct two types of counterfactual inputs (Fig. 4): 1) `Param Corrupt`: It aims to knock out the role of the parametric knowledge. We replace both occurrences of the target entity with a made-up entity, and other parts remain unchanged. Since the model does not have parametric knowledge for this made-up entity, the circuit for parametric knowledge is broken, while the circuit for context knowledge should not be affected much. 2) `Context Corrupt`: We replace only the first occurrence of the target entity with a made-up entity, other parts remain unchanged. This would presumably only break the context knowledge retrieval, as in the last sentence the target entity does not match any previous entity. We filter entities to ensure the model has parametric knowledge, and construct more than 500 prompts for both models, and measure the average patching effect over them. When patching, we patch *all* activations across token positions for a single subspace, since we do not know the underlying circuit, and our subspace partition is *uniform* across the entire sequence.

---

[4] `https://subspace-partition.streamlit.app`
[5] Both without post training

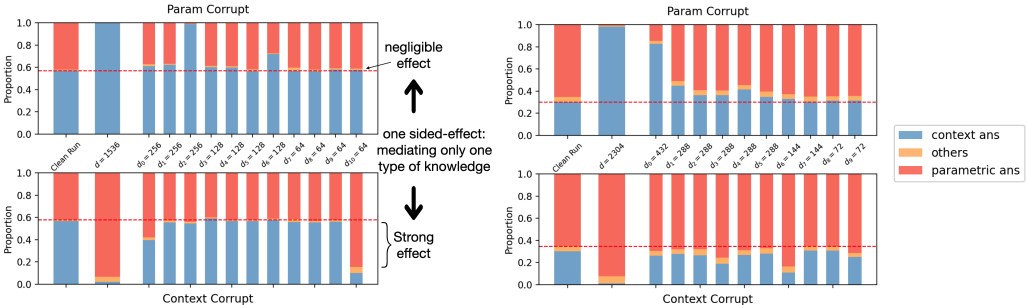

Figure 5: Subspace patching effect. **Left**: NDM partition on $h^{11,mid}$ of Qwen2.5-1.5B. The model originally predicts the context answer around 60% of the time ("Clean Run"). If the entire space is patched by activations from `Param Corrupt`, the model always chooses the context answer ("d=1536"), and likewise for `Context Corrupt`. NDM finds subspaces that only mediate parametric knowledge routing ("$d_2 = 256$", context answer is 100% under `Param Corrupt` while parametric answer rate remains the same as clean run under `Context Corrupt`) or only mediate context knowledge routing ("$d_{10} = 64$"). See Fig. 19 in App. for baseline results. **Right**: NDM on $h^{9,mid}$ of Gemma-2-2B. Similarly, some subspaces given by NDM have strong effect under one kind of counterfactual inputs but negligible effect under the other. See Fig. 20 in App. for baseline results.

**Results**   We show results in Fig. 5. We can see that NDM finds some subspaces that have non-trivial patching effect under one kind of counterfactual inputs, indicating they play a role in for only one mechanism. For example, in $h^{11,mid}$ of Qwen2.5-1.5B, subspace 2 seems to do the heavy lifting for routing parametric knowledge, while subspace 10 is crucial in routing context knowledge. Both subspaces are part of the same representation space. In contrast, baselines (Fig. 19, 20 in App.) typically show correlated patching effects under two types of counterfactual inputs. Importantly, we need to be careful when interpreting these results since the detailed circuits are unknown, see caveats in App. D.6. Moreover, we also try to qualitatively interpret some important subspaces, subspace 2 and 6 for Qwen2.5-1.5B. We found that the two subspaces indeed encode parametric knowledge while being in different stages of knowledge routing. Subspace 2 encodes local knowledge (consistently focused on the current entity/n-gram), while subspace 6 encodes knowledge propagated from elsewhere by attention layers (could be about a previous entity). See details in App. E.4.

## 6   DISCUSSIONS

**Related Work**   The internal mechanisms of a model can be analyzed and described using various units at different levels, such as components (Wang et al., 2022; Hanna et al., 2023; Stolfo et al., 2023; Prakash et al., 2024; Yu & Ananiadou, 2024), sparse features (Cunningham et al., 2023; Bricken et al., 2023; Dunefsky et al., 2024; Ameisen et al., 2025), token positions (Haklay et al., 2025; Bakalova et al., 2025), parameters (Braun et al., 2025; Bushnaq et al., 2025), and – most relevantly – subspaces. Interpretable and causally important non-neuron-basis-aligned multi-dimensional subspaces have been identified in many studies, though typically only those relevant to a specific sub-task or supporting certain arguments. For example, subspaces for gender bias (Bolukbasi et al., 2016; Belrose et al., 2023), linguistic concepts (Belrose et al., 2023; Guerner et al., 2023), greater-than and equality relationships (Wu et al., 2023; Geiger et al., 2024), estimation of current state (Shai et al., 2024), refusal behavior (Wollschläger et al., 2025), variables in abstract reasoning (Yang et al., 2025), days of the week, months, years (Engels et al., 2024; Clarke et al., 2024), and separate subspaces for digits of different significance in addition (Hu et al., 2025). Like the toy setting described in this paper, discussion of feature dependency leads to findings about interpretable subspaces (Engels et al., 2024; Clarke et al., 2024).

**Future Directions and Limitations**   As we have shown, the resulting subspaces often align with meaningful "variables", representing groups of infinitely many features sharing a higher-level concept. We believe they are better fundamental units to build circuits. Unlike feature circuits where connec-

tions are specific to concrete features, circuits defined over such "variables" capture relationships at a more abstract, high-level, and input-independent level, offering greater potential for algorithm-like description (see more discussion in App. G.3).

Our current approach shows a promising direction, but we expect that even stronger results could be achieved using substantial computational resources and engineering efforts from industry. With more compute, applications to larger models or learning better subspace partition on 2B models can produce more interesting subspaces (Compute usage of this paper is in App. H). More experiments can be done to explore more fine-grained partitions, instead of start merging from subspaces of at least 32 dimension. It is very likely that the model uses intermediate variables that take value from a small set (e.g., sentiment), thus encoded in tiny subspaces. Moreover, as we interpret directions in subspaces by using cosine similarity to define preimages, we rely on the Linear Representation Hypothesis (Elhage et al., 2022; Park et al., 2023), which could be untrue in some cases (Csordás et al., 2024). Furthermore, as mentioned in App. E.3, we do not find clear interpretations for some small subspaces. Besides reasons mentioned before, it could be because the orthogonal matrices require more training, or because we assume strong orthogonality between meaningful subspaces, or some abstract variables used internally by the model may be hard to interpret on the basis of inputs.

## 7 CONCLUSION

In this paper, we present a novel way to decompose representation space into non-basis-aligned multi-dimensional subspaces by optimizing under an unsupervised objective, neighbor distance minimization. Our method uncovers interpretable and less dependent subspaces from only data distribution. Different from previous work, we aim to investigate and describe the whole space by studying each subspace. We present both quantitative and qualitative evidence supporting our claim that obtained subspaces are interpretable and consistent most of the time. On the other hand, we also highlight some limitations of the current version of the method, e.g., partition is not fine-grained enough. We believe many of them can be improved in the future. In sum, we believe our method can serve as a useful tool for the interpretability and, more broadly, machine learning community.

## ACKNOWLEDGMENTS

Funded by the Deutsche Forschungsgemeinschaft (DFG, German Research Foundation) – Project-ID 232722074 – SFB 1102. We gratefully acknowledge the stimulating research environment of the GRK 2853/1 "Neuroexplicit Models of Language, Vision, and Action", funded by the Deutsche Forschungsgemeinschaft (DFG, German Research Foundation) under project number 471607914. We thank Aleksandra Bakalova, Yuekun Yao, and Kate McCurdy for discussions and feedback. We also thank anonymous reviewers for their feedback.

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

## A    ADDITIONAL FIGURES FOR MOTIVATION AND BACKGROUND

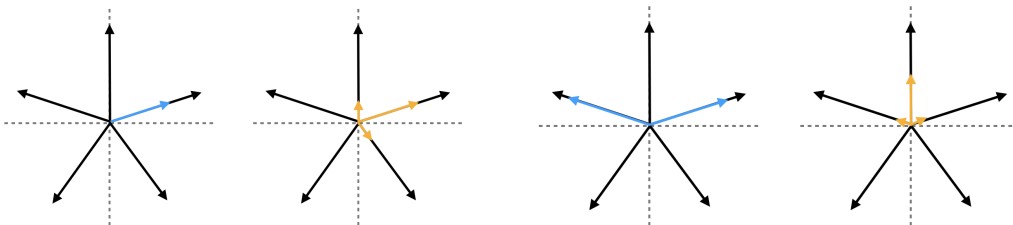

(a) Left: input features $x$, blue arrow represents the only activating feature; Right: yellow arrows represent reconstructed features $x'$

(b) Left: input features $x$, two features are activated, their sum $h$ is aligned with the y-axis; Right: reconstructed features $x'$

Figure 6: Toy model representing 5 features in 2D space, each black arrow corresponds to a column in its weight matrix $\mathbf{W}$

## B    METHODOLOGY DETAILS

### B.1    TOTAL CORRELATION PERSPECTIVE

The entropy of continuous random vector can be estimated based on the distance to $k$th-nearest neighbor (Kozachenko & Leonenko, 1987). Formally, given random samples $(x_1 \cdots x_N)$ of the random vector X, the entropy estimate $\hat{H}(X)$ is as follows

$$\hat{H}(X) = \frac{1}{N} \sum_{n=1}^{N} \log \epsilon(n) + \text{const.} \tag{5}$$

---

**Algorithm 1** NDM

---

**Require:** Activations Buffer B, $[\boldsymbol{h}_1, \cdots, \boldsymbol{h}_N]$, threshold $\tau$, initial configuration $c = [d_1, \ldots, d_S]$
where $\sum_{s=1}^{S} d_s = d$, initial orthogonal matrix $R \leftarrow I$, batch size $b$, search steps $n$, block size $m$,
merging start delay $p$, merging interval $t$, maximum update steps $K$.
  **for** $k = 1$ to K **do**
    $H_q \leftarrow$ B.pop($b$)                               ▷ a batch of activations, deleted after pop
    $\hat{H}_q \leftarrow RH_q$
    $H_{nearest} \leftarrow$ [[ ] **For** 1 to S]
    $D_{nearest} \leftarrow$ [[$\infty$ **For** 1 to b] **For** 1 to S ]
    Stop gradient tracking:
        **for** 1 to n **do**
            $H_k \leftarrow$ B.next()               ▷ a batch of size $m$, without being deleted from B
            $\hat{H}_k \leftarrow RH_k$
            **for** $s = 1$ to $S$ **do**
                D = pair_wise_dist($\hat{H}_q^{(s)}, \hat{H}_k^{(s)}$)               ▷ $D \in \mathbb{R}^{b \times m}$
                **for** $i = 1$ to $b$ **do**
                    **if** $\min(D[i]) < D_{nearest}[s][i]$ **then**
                        $D_{nearest}[s][i] \leftarrow \min(D[i])$
                        $H_{nearest}[s][i] \leftarrow H_k[\arg\min(D[i])]$
                  **end if**
                **end for**
            **end for**
        **end for**
    $D_{nearest} =$ [ ]
    **for** $s = 1$ to $S$ **do**                                 ▷ Start gradient tracking
        $\hat{H}_k \leftarrow RH_{nearest}[s]$
        d = dist($\hat{H}_q^{(s)}, \hat{H}_k^{(s)}$)                      ▷ $d \in \mathbb{R}^b$
        $D_{nearest}[s] = \frac{1}{b} \sum_b d$
    **end for**
    $f = \frac{1}{S} \sum_s D_{nearest}[s]$)                                 ▷ Final loss
    update_(R, $\nabla_R f$ )      ▷ Gradient descent, orthogonality ensured by special parameterization
    **if** $k > p$ and $k \bmod t = 0$ **then**
        $I =$ mutual_info($R, c, B$)                      ▷ $I \in \mathbb{R}^{S \times S}$
        **if** any $\frac{I_{[s_1][s_2]}}{d_{s_1} + d_{s_2}} > \tau$ **then**
            Sort by $\frac{I_{[s_1][s_2]}}{d_{s_1} + d_{s_2}}$ and select top $S/8$ non-intersecting pairs $> \tau$
            Update configuration c (and R if necessary, to keep the correspondence between c and
sub-matrices of R)
        **else**
            **break**
        **end if**
    **end if**
  **end for**
  **return** R, c

---

$\epsilon(n)$ is twice the distance from $x_n$ to its $k$th neighbor, const. is a constant if N and k are fixed. Meanwhile, the total correlation (Watanabe, 1960) or multi-information (Studenỳ & Vejnarová, 1998) is defined as

$$C(X_1, \ldots, X_S) = \sum_{s=1}^{S} H(X_s) - H(X_1, \ldots, X_S) \tag{6}$$

which generalizes mutual information's binary case to multiple random variables and measures the redundancy and dependence among them. Therefore, when we minimize the distances in Eq.4, the entropy of individual subspaces decreases[6], and because the last term of Eq.6 is constant as the entropy of the joint subspace is invariant under orthogonal transformation, we actually minimize the total correlation.

## B.2 Algorithm Details

Here we supplement Sec. 3 and Algorithm 1 with some important details and explanations. The MI is measured by KSG estimator (Kraskov et al., 2004). We use the first algorithm presented in the paper (Estimator $I^{(1)}(X, Y)$) and normalize the variance of each subspace before estimation, by dividing subspace activations by the square root of the sum of the variance of each dimension in the subspace. Note that MI is invariant under this transformation in theory but it helps reduce errors in practice (Kraskov et al., 2004). We measure pair-wise MI for subspaces, and then divide them by the sum of dimensions of subspace pairs. We use subspace dimension as a very rough estimate of entropy, which forms the upper-bound of MI. So the intuition behind normalizing MI with dimensionality is to roughly estimate the ratio of entropy removed after the other variable is observed. We set a threshold for normalized MI, we select and sort all pairs whose normalized MI is above the threshold. We then iterate over these pairs, selecting pairs that do not intersect with selected pairs. For example, for subspace a, b, c, if (a, b) is selected, (b, c) will not be selected. We merge at most $S/8$ pairs, where $S$ is the number of subspaces. Note that merging does not change the weights of $\mathbf{R}$ (or just reordering), merging affects the training as we now search the nearest neighbor in the joint space and optimize the distance in the joint space. When we merge subspaces, the optimizer states (for the whole $\mathbf{R}$ are also re-initialized.

Note that although we have experimented with some alternatives, these specific details are not necessarily optimal, each part of the method can be studied and optimized more extensively in the future.

Lastly, to clarify which hyperparameters we refer to throughout the rest of the paper. We assign them the following names: the initial dimensionality for subspaces is referred as unit size (i.e. $d_1, \cdots d_S$ in Algorithm 1 which are equal at the beginning). We search the nearest neighbor among $N$ activations (including the point itself, so only $N-1$ candidates), we refer this as search number (i.e. equal to $n \times m$ in Algorithm 1). The orthogonal matrix is first trained for some steps, during this stage no merging happens, we call it merging start delay. After this stage, we measure MI and merge subspaces every $t$ step, we call $t$ merging interval. Lastly, the normalized MI threshold $\tau$ is called merging threshold.

## C More Experiments in Toy Setting

### C.1 Toy Model Feature Geometry

In this section, we extend the experiment in Sec 2 with more different configurations. We keep model the same, while changing the feature group structure and model dimensions to bring a more comprehensive view. We use group=(20, 20) to refer to 2 groups, each group has 20 features, other cases follow this pattern. All experiments in toy setting is done with batch size of 128, training steps of 10000, Adam optimizer, and linear learning rate scheduler which decrease learning rate from 0.003 to 0.0003. The trained model is tested on 12800 samples. Overall, we see strong orthogonal structure in all cases.

---

[6]In our preliminary experiments, we found including the $\log(\cdot)$ in Eq.5 as part of the training objective does not help.

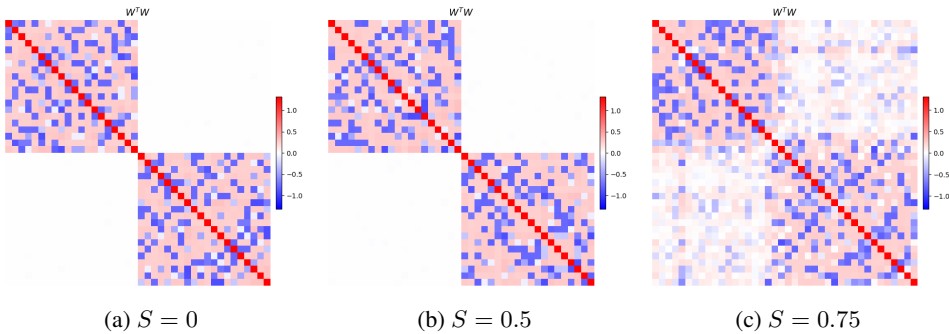

(a) $S = 0$   (b) $S = 0.5$   (c) $S = 0.75$

Figure 7: $\mathbf{W}^T\mathbf{W}$, $z = 40$, $d = 12$, group=(20, 20)

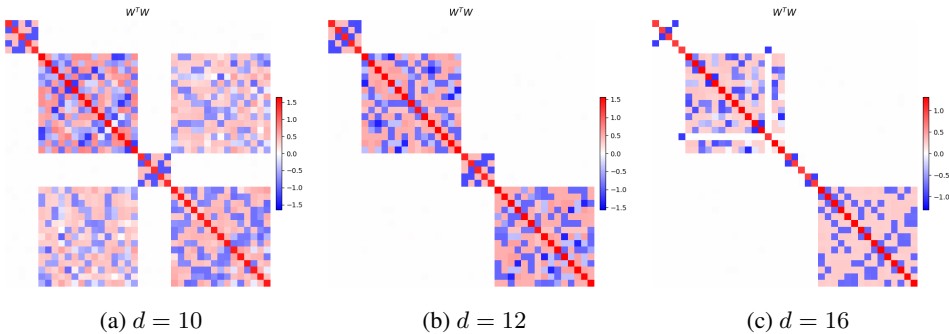

(a) $d = 10$   (b) $d = 12$   (c) $d = 16$

Figure 8: $\mathbf{W}^T\mathbf{W}$, $z = 40$, group=(5, 15, 5, 15), $S = 0.25$

**Group Sparsity**   We refer the probability for a whole feature group being not activating as group sparsity $S$. In Sec 2 we have experimented with $S = 0.25$. We keep other parameters the same ($z = 40$, $d = 12$, group=(20, 20)), but set $S = 0.0$ (Fig. 7a) $S = 0.5$ (Fig. 7b) and $S = 0.75$ (Fig. 7c). We train the model until convergence and final FVU is 0.028, 0.028 and 0.035 respectively. From the figures we can see as the $S$ increases, features in different groups are less likely to co-occur, thus more similar to mutual exclusive condition, which encourages superposition. So we see weaker orthogonality (bottom left and top right submatrices are nonzero, though the absolute values are smaller) when $S = 0.75$.

**More Groups and Different Hidden Dimension**   We also experiment with group=(5, 15, 5, 15), $z = 40$, $S = 0.25$, and $d = 10, 12, 16$, corresponding to 3 subfigures in Fig. 8. Final FVU are 0.166, 0.066, 0.016 for $d = 10, 12, 16$ respectively. We can still see similar orthogonal structure. When $d = 10$, the dimensionality is not enough to allocate orthogonal subspaces to each group, and the features from 2nd and 4th groups are superposed (as their frequency is lower than those from the other two groups), and the model cannot reconstruct $x$ very well. When $d = 16$, as the dimensionality constraint becomes more relieved, orthogonality also occurs within groups.

**More Features**   We increase the total number of features and experiment with 2 configurations. (a) $d = 32$, $z = 160$, $S = 0.25$, group=(80, 80), Fig. 9a, final FVU is 0.025. (b) $d = 64$, $z = 400$, $S = 0.25$, group=(40, 80, 120, 160), Fig. 9b, final FVU is 0.033. Again, the orthogonal structure is very obvious.

## C.2   Toy Model Subspace Partitioning

In this section, we describe more experiments of NDM in toy setting. We apply it for toy models trained in previous section (Sec. C.1) whose orthogonal structure is clear.

**Experimental Details**   For toy setting, we use a slightly different version from Algorithm 1. Specifically, we find it is not necessary to search over a huge amount of subspace activations, so we

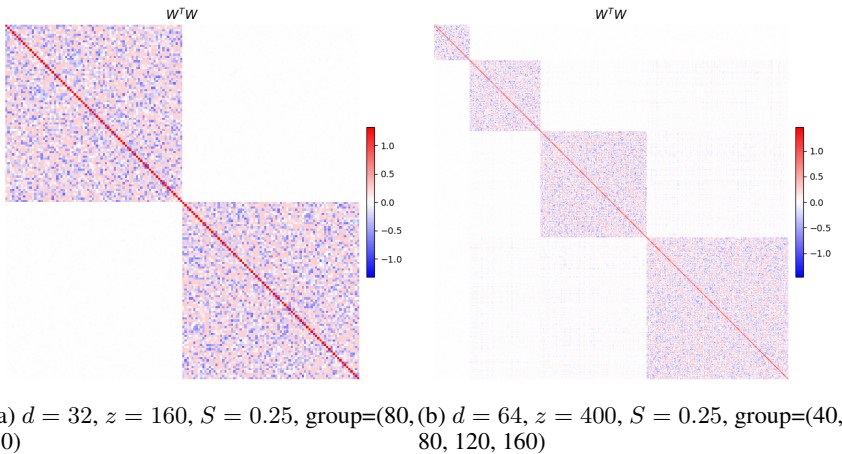

(a) $d = 32$, $z = 160$, $S = 0.25$, group=(80, 80)

(b) $d = 64$, $z = 400$, $S = 0.25$, group=(40, 80, 120, 160)

Figure 9: $\mathbf{W}^T\mathbf{W}$

search for the nearest neighbor inside the batch (Eq. 3) thus simplify the algorithm. In addition, we find $\mathbf{R}$ usually get stuck at sub-optimal point. When observing $\mathbf{RW}$, we usually find one subspace is computed from two feature group, and this is usually accompanied with higher overall training loss, and more training steps and changing hyperparameters in Adam optimizer do not seem to help. We apply the following heuristics: we take the subspace with the largest loss and measure MI between this and all other subspaces, and multiply the MI values with loss of all other subspaces, then divide the resulting values (always $> 0$) by their sum as the probability by which we randomly choose the other subspace. This process selects two large loss subspaces which encodes similar information. We multiply their corresponding rows in $\mathbf{R}$ with a random orthogonal matrix ($\in \mathbb{R}^{(d_1+d_2)\times(d_1+d_2)}$), thus re-initialize the weights for these two subspaces by mixing these two subspaces, while the whole matrix $\mathbf{R}$ remains orthogonal. We find that this technique helps $R$ achieve lower training loss and produce a more ideal $\mathbf{RW}$. In other words, we have a better chance to find global minimum. But we also find this becomes less effective when number of features and dimension become larger. This also calls for more study on this atypical optimization problem, i.e., with orthogonality constraint and ignorance of distance change to all but the nearest neighbor when computing gradients.

For reproducibility we report the hyperparameters used as follows: We use euclidean distance as the optimization objective. We find 1 - cosine similarity leads to worse results, we also find taking the square or logarithm of the distance does not help. We use 2 as the unit size. We use search number $N = 4$. We find small search number is crucial, it can be much smaller than the number of features, but we are unclear about why small search number helps. For re-initialization of largest loss, we evaluate the loss (distance to nearest neighbor) on a much larger search number, $N = 1024$, which we find more accurate for selecting the correct subspace (subspace that contains more than 1 feature group). We use batch size of 128, learning rate of 0.001, and Adam optimizer. As we described previously, the training step is not fixed. Merge start delay is 60k steps, during this stage two subspaces are randomly re-initialized every 3k step. The optimizer states for the whole matrix are also re-initialized. After the 60k steps, we measure MI and merge subspaces with merge interval of 3k steps, and merge threshold of 0.04. Importantly, we use the same set of hyperparameters for all experiments in toy setting, regardless of the varying number of features and activation dimension.

**Experimental Results** We train $\mathbf{R}$ for toy models whose orthogonal structure is clear. Concretely, we present 4 experiments here, corresponding to toy models visualized in Fig. 1a, Fig. 8b, Fig. 9a and Fig. 9b. The results are shown in Fig. 10, 11, 12, 13. We can see the proposed method is quite effective. Before merging, each subspace gradually learns to encode only one feature group. For example, in Fig. 10c, each 2D subspace in $\hat{h}$ is computed either from the first or the second feature group (e.g., the first 2 rows only have nonzero entries in first 20 columns, corresponding to the first group). In Fig. 11, at the beginning the third subspace encodes all 4 groups (a), then after 3k steps it only encodes the 1st and 3rd group (b), then later it encodes only the 1st group. We can clearly see how NDM training objective leads $\mathbf{R}$ to evolve towards the desirable direction. Loss

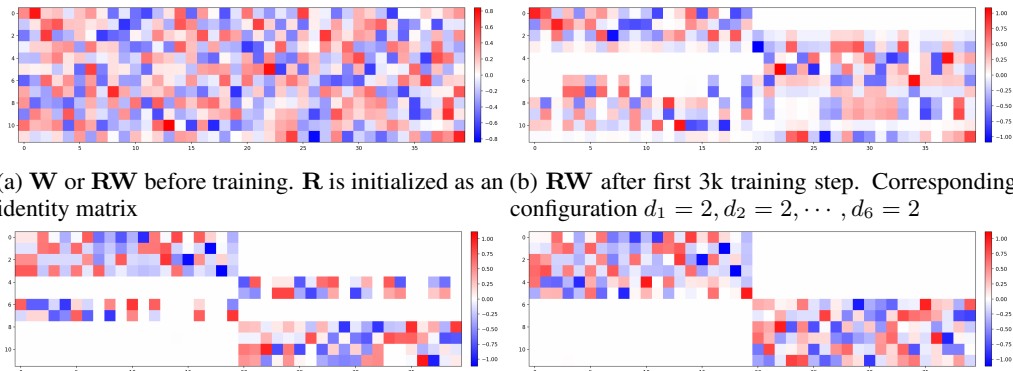

(a) **W** or **RW** before training. **R** is initialized as an identity matrix

(b) **RW** after first 3k training step. Corresponding configuration $d_1 = 2, d_2 = 2, \cdots, d_6 = 2$

(c) **RW** after 60k training step before merging. Corresponding configuration $d_1 = 2, d_2 = 2, \cdots, d_6 = 2$

(d) **RW** after entire training procedure completes. Corresponding configuration $d_1 = 6, d_2 = 6$

Figure 10: **RW** at different stages of training. The toy model is the one from Fig. 1a, where $d = 12$ $z = 40$, group=(20, 20), $S = 0.25$

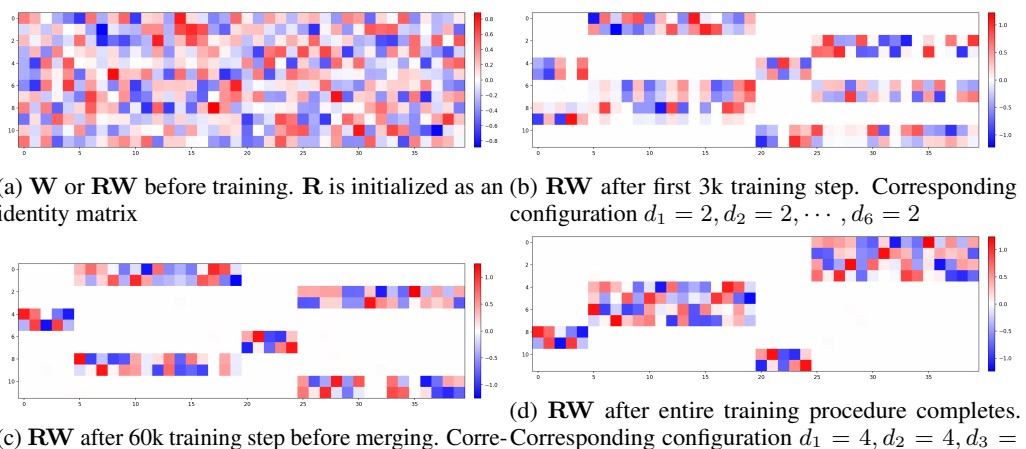

(a) **W** or **RW** before training. **R** is initialized as an identity matrix

(b) **RW** after first 3k training step. Corresponding configuration $d_1 = 2, d_2 = 2, \cdots, d_6 = 2$

(c) **RW** after 60k training step before merging. Corresponding configuration $d_1 = 2, d_2 = 2, \cdots, d_6 = 2$

(d) **RW** after entire training procedure completes. Corresponding configuration $d_1 = 4, d_2 = 4, d_3 = 2, d_4 = 2$

Figure 11: **RW** at different stages of training. The toy model is the one from Fig. 8b, where $d = 12$ $z = 40$, group=(5, 15, 5, 15), $S = 0.25$

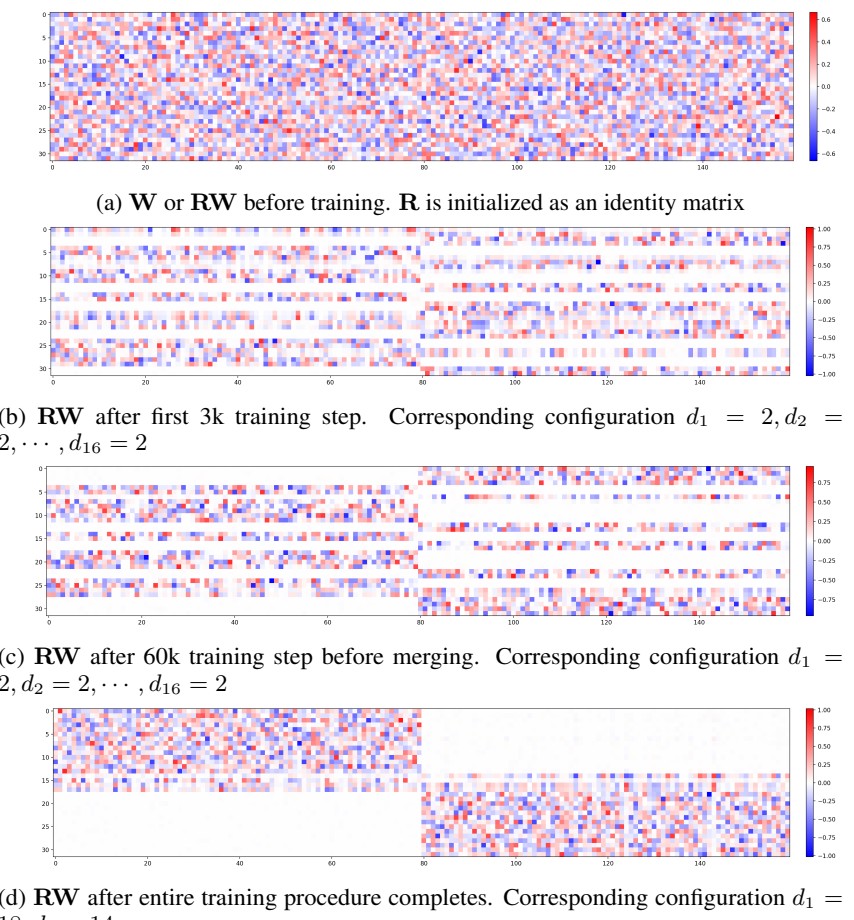

(a) **W** or **RW** before training. **R** is initialized as an identity matrix

(b) **RW** after first 3k training step. Corresponding configuration $d_1 = 2, d_2 = 2, \cdots, d_{16} = 2$

(c) **RW** after 60k training step before merging. Corresponding configuration $d_1 = 2, d_2 = 2, \cdots, d_{16} = 2$

(d) **RW** after entire training procedure completes. Corresponding configuration $d_1 = 18, d_2 = 14$

Figure 12: **RW** at different stages of training. The toy model is the one from Fig. 9a, where $d = 32$ $z = 160$, group=(80, 80), $S = 0.25$

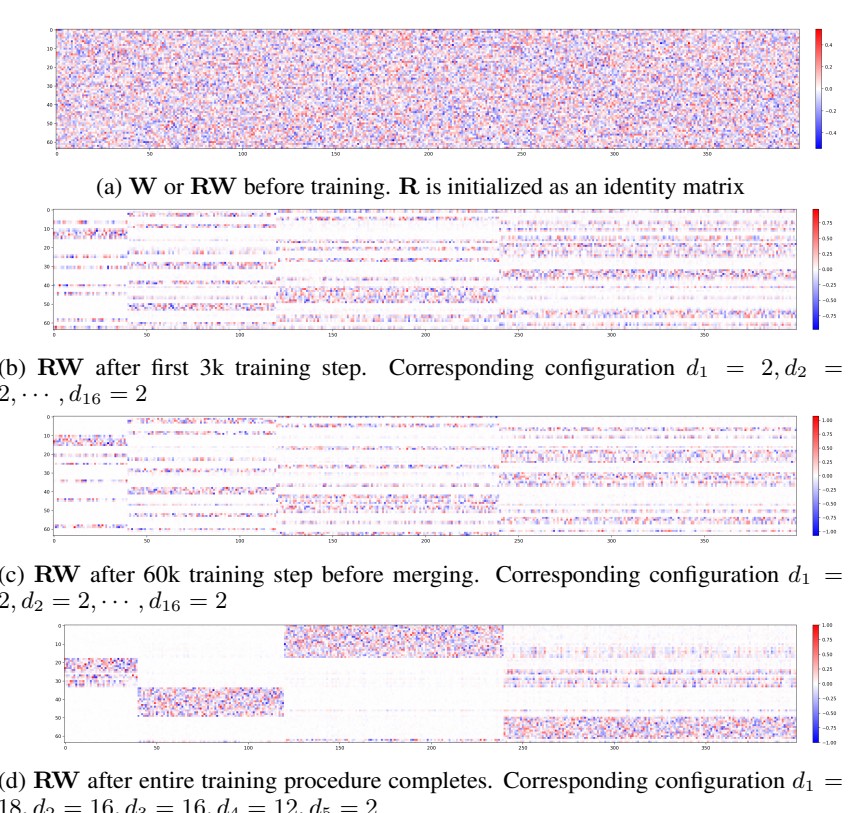

(a) $\mathbf{W}$ or $\mathbf{RW}$ before training. $\mathbf{R}$ is initialized as an identity matrix

(b) $\mathbf{RW}$ after first 3k training step. Corresponding configuration $d_1 = 2, d_2 = 2, \cdots, d_{16} = 2$

(c) $\mathbf{RW}$ after 60k training step before merging. Corresponding configuration $d_1 = 2, d_2 = 2, \cdots, d_{16} = 2$

(d) $\mathbf{RW}$ after entire training procedure completes. Corresponding configuration $d_1 = 18, d_2 = 16, d_3 = 16, d_4 = 12, d_5 = 2$

Figure 13: $\mathbf{RW}$ at different stages of training. The toy model is the one from Fig. 9b, where $d = 32$ $z = 400$, group=(40, 80, 120, 160), $S = 0.25$

values also confirm the alignment between neighbor distances and "purity" of subspace (encoding one group instead of a mixture of groups). We denote the loss when search number is 4 as $l_{N=4}$, and similarly $l_{N=1024}$. We take the mean of loss across subspaces instead of sum when reporting them. For example, Fig. 10b's loss values are: $l_{N=4} = 0.212$ and $l_{N=1024} = 0.011$. After more training when subspaces are "pure" (subfigure c), loss values are: $l_{N=4} = 0.194$ and $l_{N=1024} = 0.004$. Another example, Fig. 11 loss values are: $l_{N=4} = 0.340$ and $l_{N=1024} = 0.016$ for subfigure b, and $l_{N=4} = 0.322$ and $l_{N=1024} = 0.004$ for subfigure c. Importantly, we also see that $\mathbf{R}$ sometimes does not find global minimum. In Fig. 12 and 13, we see some sub-optimal subspaces, such as the 3rd subspace from the last in Fig. 12c, which is the reason for later imperfection of final $\mathbf{R}$. Moreover, we also see that the proposed MI-based merging works well. It correctly merges the subspaces encoding the same group. For example, in Fig. 11, normalized MI between 1st and 5th subspace is 0.55, and between 2nd and 6th is 0.49, normalized MI between all other pairs is below 0.03. In Fig. 10 and 11, we get the correct number of subspace and their dimension. In Fig. 12 and 13, the final configuration is imperfect but close to the ground truth. As mentioned before, we generally find the imperfection results from $\mathbf{R}$ being unable to find global minimum or disentangle feature groups for a few subspaces, the MI estimate usually works well.

In summary, we find the training objective of NDM indeed encourages subspaces to encode only one feature group. Together with MI based merging, the final subspace partition is close to the groundtruth, with each subspace encoding one feature group. While training objective aligns well with our goal, the orthogonal matrix sometimes gets stuck at local minimum of this objective, resulting in sub-optimal partition. Nevertheless, it might not be necessary to achieve perfect partition to make itself useful in practice. In subfigure b of Fig. 10-13, we observe that these sub-optimal points already have a good deal of disentanglement, which could potentially be very useful when using these subspaces as analyzing unit instead of random subspaces.

lastly, more experiments can be done in toy setting, we have not done extensive experiments exploring the effect of varying unit size and search number, much more feature groups, and many other aspects. More rigorous study can be done in this ideal testbed, but we would like to focus more on real world language models, in order to address more urgent concerns about whether the orthogonal structure even exists in real models.

# D    QUANTITATIVE EXPERIMENTS ON LANGUAGE MODELS

## D.1    SUBSPACE PATCHING

Here we describe our method in detail. Formally, suppose $\boldsymbol{h}_{\text{cln}}$ and $\boldsymbol{h}_{\text{crp}}$ are the activations from clean and corrupt inputs respectively, at the same layer and token position. We patch subspace $s$ as follows:

$$\hat{\boldsymbol{h}}_{\text{crp}} = \mathbf{R}\boldsymbol{h}_{\text{crp}} \tag{7}$$

$$\hat{\boldsymbol{h}}_{\text{cln}} = \mathbf{R}\boldsymbol{h}_{\text{cln}} \tag{8}$$

$$\hat{\boldsymbol{h}}_{\text{crp}} = \begin{bmatrix} \hat{\boldsymbol{h}}_{\text{crp}}^{(1)} \\ \vdots \\ \hat{\boldsymbol{h}}_{\text{crp}}^{(s)} \\ \vdots \\ \hat{\boldsymbol{h}}_{\text{crp}}^{(S)} \end{bmatrix}, \quad \hat{\boldsymbol{h}}_{\text{cln}} = \begin{bmatrix} \hat{\boldsymbol{h}}_{\text{cln}}^{(1)} \\ \vdots \\ \hat{\boldsymbol{h}}_{\text{cln}}^{(s)} \\ \vdots \\ \hat{\boldsymbol{h}}_{\text{cln}}^{(S)} \end{bmatrix} \tag{9}$$

$$\hat{\bar{\boldsymbol{h}}} = \begin{bmatrix} \hat{\boldsymbol{h}}_{\text{cln}}^{(1)} \\ \vdots \\ \hat{\boldsymbol{h}}_{\text{crp}}^{(s)} \\ \vdots \\ \hat{\boldsymbol{h}}_{\text{cln}}^{(S)} \end{bmatrix} \tag{10}$$

$$\bar{\boldsymbol{h}} = \mathbf{R}^T \hat{\bar{\boldsymbol{h}}} \tag{11}$$

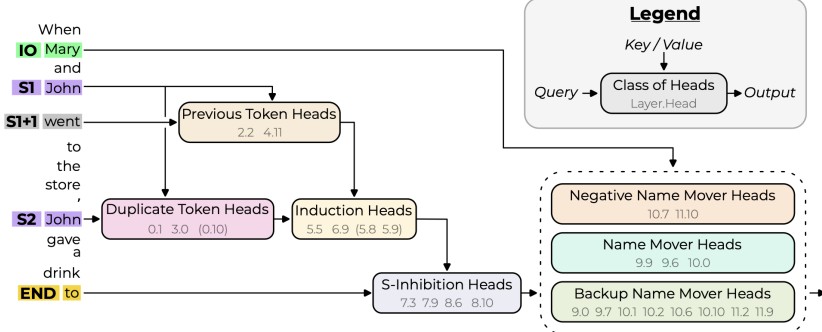

Figure 14: IOI circuit, Fig. 2 from (Wang et al., 2022). Attention heads in this figure is denoted as `layer_index.head_index`. E.g., The second previous token head, 4.11, refers to layer 4 and attention head 11 in that layer. Note that indices start from 0.

after patching, the model uses $\bar{h}$ to finish the rest of the forward pass on the clean input. To measure the patching effect, we first run the model on clean input, recording the original logits or probability, which we refer to as clean run. Then we run the model on corrupted input, to save the corrupted activations $h_{\mathrm{crp}}$, which we refer to as corrupted run. Finally we run the model on clean input, with certain subspace activation being patched with the value saved in corrupted run, and record the patched logits or probability, which we refer to as patched run. We compare the model output from clean run and patched run to measure the subspace patching effect.

## D.2 GPT-2 SMALL TEST SUITE

**Tests based on IOI circuit** The Indirect Object Identification (IOI) task consists of sentences like: "When Mary and John went to the store, John gave a drink to _", which should be completed with "Mary". (Wang et al., 2022) found a circuit in GPT-2 Small responsible for solving this task (Fig. 14). Besides computational graph, they also reveal the information moved between the nodes of the graph.

To understand our intervention, we briefly describe the information flow inside the circuit: At token S1+1 (see Fig. 14 for the notation), the model attends the previous token and moves the name from it, which will be used as key in later induction heads. *Right after layer 4, the model finishes moving previous token information, and this information is not yet used by downstream components.* At token S2, the induction heads attend to S1+1, moving the relative position of S1 (i.e., it occurs before or after IO, or equivalently, S1+1 is "and" or "went"). Meanwhile, Duplicate token heads attend to S1 from S2, moving S name and relative position of S1. Therefore, at S2's residual stream, *right after layer 6 the model finishes moving position information of S1, and it is not yet moved by any downstream component.* Then at END token, S-Inhibition heads attend to S2 and move the position information of S1 and as well as the S name. *Right after layer 8, the model finishes moving position information of S1 and information of S name, and they are not yet used by any name mover heads.* Lastly, name mover heads avoid attending S1 by using those information and attend to IO to move IO name. Note that the claims about the moved information is supported by strong causal evidence from (Wang et al., 2022), and is confirmed again with a different method in (Huang et al., 2024b).

Importantly, we emphasize 3 key places in the circuit in our description (see the italics). Specifically, $h_{\mathrm{S1+1}}^{4,\mathrm{post}}$, $h_{\mathrm{S2}}^{6,\mathrm{post}}$, $h_{\mathrm{END}}^{8,\mathrm{post}}$. They are ideal for causal intervention because in those places we know the existing information, and all heads that gather the information are before the place and all heads that use the information are after the place. In contrast, if we intervene on $h_{\mathrm{S1+1}}^{3,\mathrm{post}}$ to change previous token information, the changed information might be overwritten by head 4.11, resulting in negligible effect even if we intervene on the right subspace. This is also the reason why we do not pick the layer after duplicate token heads, because similar information will be added later by induction heads. Moreover, if we intervene on $h_{\mathrm{S1+1}}^{5,\mathrm{post}}$ to change previous token information, it will only affect induction head 6.9, while the original information was already moved by heads in layer 5. We do not select $h_{\mathrm{END}}^{11,\mathrm{post}}$

because we suspect there is a big "output subspace" after the last layer and our method would not be very interesting there.

Based on the information used in the circuit, we design two kinds of counterfactual inputs. **(1) Position Swap**. We swap the position of the IO and S1 in the sub-clause at the beginning. For example, clean input: "When Mary and John went to the store, John gave a drink to"; corrupted input: "When John and Mary went to the store, John gave a drink to". **(2) Name Swap**. We swap the names of IO and S (both S1 and S2). For example, clean input: "When Mary and John went to the store, John gave a drink to"; corrupted input: "When John and Mary went to the store, Mary gave a drink to".

Finally, we design following tests:

1. Do subspace patching in residual stream $h_{\text{S1+1}}^{4,\text{post}}$ and $h_{\text{IO+1}}^{4,\text{post}}$ (IO+1 means the token following IO), using Position Swap. If the information about previous token is successfully changed, then in the patched run, induction heads would attend IO+1 instead of S1+1. Therefore, the induction heads would move the position of IO instead of S1, and pass the changed information to later components and cause name mover heads to avoid attending IO.

2. Do subspace patching in residual stream $h_{\text{S2}}^{6,\text{post}}$ using Position Swap. If patching in the right subspace, corrupted activations should contain a different position of S1, thus cause the model to change towards an opposite prediction, similar to the explanation above.

3. Do subspace patching in residual stream $h_{\text{END}}^{8,\text{post}}$ using Position Swap. This is to change the position of S1 in the residual stream of END.

4. Do subspace patching in residual stream $h_{\text{END}}^{8,\text{post}}$ using Name Swap. This is to change the S name in the residual stream of END.

For all 4 tests in IOI circuit, we measure the drop in logit difference, we denote it as $\Delta LD$. So $\Delta LD = (\text{logit}_{\text{cln}}^{\text{IO}} - \text{logit}_{\text{cln}}^{\text{S}}) - (\text{logit}_{\text{pch}}^{\text{IO}} - \text{logit}_{\text{pch}}^{\text{S}})$, where $\text{logit}_{\text{cln}}^{\text{IO}}$ is the logit value for IO name on the clean run, and $\text{logit}_{\text{pch}}^{\text{IO}}$ is the logit value for IO name on the patched run, and similarly for S. We use 1000 examples, which are clean inputs, and we construct counterfactual inputs on-the-fly. In these examples, the order of IO and S1 in the sub-clause is uniformly random. The names are picked randomly from a list of common one-token names. The template ("When ... went to the store, ... gave a drink to", "Then, ... had a long argument, and afterwards ... said to", etc.) is also picked randomly from a pre-defined list. The final reported effect $\Delta LD$ is averaged over the 1000 samples.

**Test based on Greater-than circuit**   The Greater-than task (Hanna et al., 2023) consists of sentences in the form "The `<noun>` lasted from the year `XXYY` to the year `XX`", such as "The war lasted from the year 1732 to the year 17", the model should assign higher probability to years greater than 32, thus performing the simple greater-than math operation. (Hanna et al., 2023) revealed a circuit in GPT-2 Small responsible for solving this task. Briefly speaking, some attention heads between layer 5 to layer 9 (both inclusive) move `YY` to the last token's residual stream, and MLP layer 8 to 11 (both inclusive) specify which years are greater than `YY`. Therefore, *right after layer 9, the model finishes moving* `YY` *to the last token*, so we intervene at $h_{\text{END}}^{9,\text{post}}$ to change the `YY`, where END denotes the last token.

Regarding counterfactual inputs, we use the same as the original paper: the `YY` in clean input is replaced by 01. For example, clean input: "The war lasted from the year 1732 to the year 17"; corrupted input: "The war lasted from the year 1701 to the year 17".

We do one test based on Greater-than circuit (continue the indexing from IOI tests):

5. Do subspace patching in residual stream $h_{\text{END}}^{9,\text{post}}$, using 01 counterfactual input. If the information `YY` is successfully replaced with 01, the sum of the probability of years greater than `YY` will decrease.

Unlike IOI, we do not compare logits of two tokens as there are many possible answers. Instead, we compute the drop in sum of the probability of valid answers, we denote it as $\Delta P$. In our running example where `YY`=32, it means $\Delta P = P_{\text{cln}}(\text{next token} \in \{33, 34, \cdots, 99\}) - P_{\text{pch}}(\text{next token} \in \{33, 34, \cdots, 99\})$, where $P_{\text{cln}}$ and $P_{\text{pch}}$ denotes probability in clean and patched run respectively.

Similar to IOI, the inputs for this task is also constructed from random nouns, years and random templates, while restricting years such that they are always 2-token long. The final reported effect $\Delta P$ is averaged over the 1000 samples.

**Measuring target information inequality**    Therefore, we have 5 tests in total for GPT-2 Small, and 4 layers are involved. To test a training configuration of NDM, we apply the same procedure and hyperparameters to train 4 orthogonal matrices, for post MLP residual stream at layer 4, 6, 8, 9 respectively. Then apply the 5 aforementioned tests to each of the subspaces found in corresponding layer.

Each test changes one *type* of information. Analogously it means a fixed "variable", while the "variable" can take many different values. To measure if the target information (the "variable" we are interested in) only lives in one subspace, we quantify the inequality of patching effect across subspaces by calculating Gini coefficient. Same as before, $s$ is subspace index, for the patching effect $\Delta_s$ (can be either $\Delta LD_s$ or $\Delta P_s$), Gini coefficient is

$$G = \frac{\sum_{s_1=1}^{S} \sum_{s_2=1}^{S} |\Delta_{s_1} - \Delta_{s_2}|}{2S \sum_{s=1}^{S} \Delta_s} \tag{12}$$

A Gini coefficient of 0 means the effect is evenly distributed across all subspaces. If the effect is nonzero for only one subspace and zero elsewhere, the Gini coefficient will be close to 1.0 (the more subspaces there are, the closer it gets). If the effect is concentrated in about half of the subspaces, the Gini coefficient will fall somewhere in between. So by using Gini coefficient, the inequality of effects across subspaces or the inequality of the amount of target information across subspaces is summarized as one value, which makes our evaluation easier.

Importantly, there is one trivial way to get high value in this metric while not matching our expectation: there can be one big subspace and many tiny subspaces (e.g., one dimensional), the big one would presumably encode the target information, but as well as many other types of information. Therefore, we also calculate Gini coefficient after dividing the patching effect by the dimension of the subspace, which computes the "density" of target information in each subspace, or "amount of target information per dimension". Moreover, there can be another trivial way to get high value in the second metric: one can use PCA to concentrate variance or compress information into a few subspaces, thus these subspaces would have high "density" for all kinds of information. This again goes against our expectation, as we want subspaces to have high "density" for one or few kinds of information and low "density" for all others. Therefore, the third metric is to calculate Gini coefficient after dividing the patching effect by the sum of the variance of each axis in the subspace. We refer to this sum of variance as $\text{Var}_s$. We report all 3 variants (original effect, normalized by dimension, normalized by variance) in the results. Ideally, a good subspace partition should achieve high values in all 3 metrics.

## D.3    NDM Training Details on GPT-2 Test Suite

We use the NDM algorithm described in Sec. 3 (with more details provided in App. B and Algorithm 1).

For reproducibility, we report the hyperparameters used as follows: We use euclidean distance as the optimization objective. We use 32 as the unit size. We try different search numbers $N = \{1 \times 2^{14}, 5 \times 2^{14}, 25 \times 2^{14}\}$. We do not do random re-initialization during the pre-merging stage like we did for the toy setting; preliminary results show this is not helpful for language models. We use batch size 128, learning rate 0.0003, and Adam optimizer. We try different merging intervals, 4k and 8k, and set the merging start delay as 2.5 times the merging interval. Longer merging intervals mean more training steps in general. We set 100k as the maximum training steps, which is usually not reached. We also try different merging threshold $\{0.02, 0.03, 0.04, 0.05\}$. We use MiniPile (Kaddour, 2023) as the data to run GPT-2 Small when collecting activations, instead of the OpenWebText dataset which is similar to GPT-2 pretraining data. Therefore, we do hyperparameter search over all the combinations of different search numbers, merging interval, and merging threshold, while other hyperparameters are always the same. For each hyperparameter configuration, we train 4 orthogonal matrices for post-MLP residual stream at layer 4, 6, 8, 9 respectively, as they are needed in the test suite.

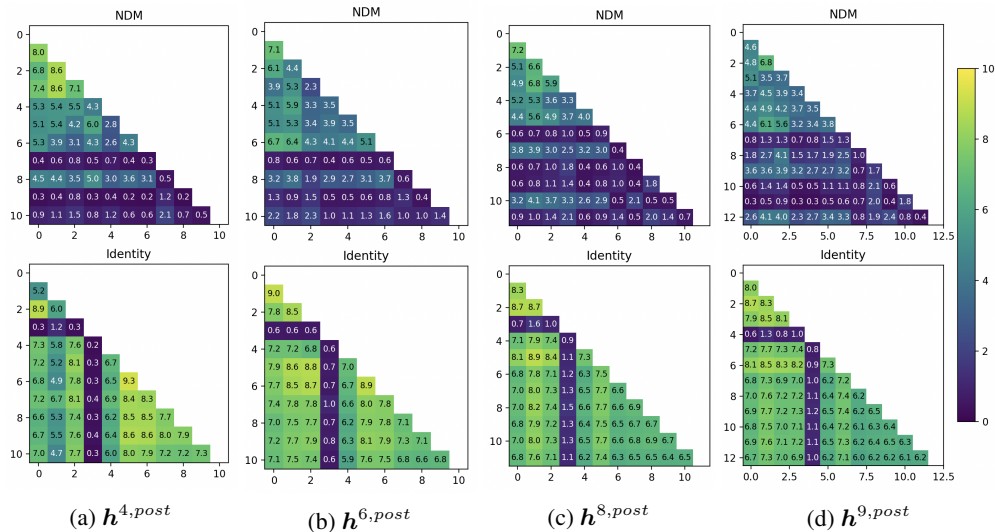

(a) $\boldsymbol{h}^{4,post}$  (b) $\boldsymbol{h}^{6,post}$  (c) $\boldsymbol{h}^{8,post}$  (d) $\boldsymbol{h}^{9,post}$

Figure 15: MI between each pair of subspaces. Top row: subspace partition given by NDM. Bottom row: subspace partition using the identity matrix and the dimensionality configuration given by NDM. Columns correspond to 4 layers in GPT-2 Small. We use the same MI estimation method as the one used during training. The MI values are not normalized. Bigger values in the top left corner are because subspaces are sorted in descending order according to dimensionality. We can see NDM produces more independent subspaces.

| index $s$ | 0 | 1 | 2 | 3 | 4 | 5 | 6 | 7 | 8 | 9 | 10 | 11 | 12 |
|---|---|---|---|---|---|---|---|---|---|---|---|---|---|
| $d_s$ | 128 | 96 | 96 | 64 | 64 | 64 | 64 | 32 | 32 | 32 | 32 | 32 | 32 |
| $\text{Var}_s$ $(\times 10^3)$ | 2.1 | 2.7 | 1.3 | 0.6 | 0.8 | 1.4 | 1.7 | 1.0 | 1.0 | 0.4 | 4.0 | 4.8 | 0.3 |
| $\Delta P_s$ $(\times 10^{-2})$ | 5.4 | 0.7 | 1.8 | 1.9 | 29.7 | 0.6 | 0.6 | 0.3 | 0.6 | 0.8 | 1.2 | 0.7 | 0.7 |
| $\frac{\Delta P_s}{d_s}$ $(\times 10^{-4})$ | 4.2 | 0.8 | 1.9 | 3.0 | 46.4 | 1.0 | 0.9 | 0.9 | 2.0 | 2.4 | 3.9 | 2.0 | 2.3 |
| $\frac{\Delta P_s}{\text{Var}_s}$ $(\times 10^{-5})$ | 2.6 | 0.3 | 1.4 | 2.9 | 39.6 | 0.5 | 0.3 | 0.3 | 0.6 | 2.2 | 0.3 | 0.1 | 2.2 |

Table 2: Test 5 raw data for NDM results shown in main paper (Table 1). Gini coefficient for row $\Delta P_s$ is 0.72, for row $\frac{\Delta P_s}{d_s}$ is 0.67, and for row $\frac{\Delta P_s}{\text{Var}_s}$ is 0.78.

As shown in later section, the best configuration is search number = $25 \times 2^{14}$}, merging interval = 8k, merging threshold = 0.04. In Fig. 15, we show the MI after training, compared with the identity matrix, which is used to initialize $\mathbf{R}$.

In later section, we show ablation studies, and some of the hyperparameter choices will be justified by those results.

### D.4 REPRESENTING PATCHING EFFECT DISTRIBUTION WITH GINI COEFFICIENT

We do not show the patching effect for each individual subspace, as there would be too many numbers, but rather summarize the degree of concentration using Gini coefficient. To provide a sense of how high the Gini coefficient should be for the partition to be considered good, we show some raw data in Table 2 - 4. We can see that, when the Gini coefficient is above 0.6 (Table 2), the distribution of the effect is quite satisfactory, as there is a single value much greater than all others. When Gini coefficient is around 0.5 (third and fifth row in Table 3), we do see some degree of concentration, but effect is concentrated in quite a few subspaces. Lastly, if Gini coefficient is lower than 0.3 (Table 4), we do not see any significant degree of concentration. Therefore, we can say that Gini coefficient should be greater than 0.6 to be considered "good", and the higher the better. In other sections of the paper, we only report Gini coefficients for the GPT-2 Small test suite.

| index $s$ | 0 | 1 | 2 | 3 | 4 | 5 | 6 | 7 | 8 | 9 | 10 |
|---|---|---|---|---|---|---|---|---|---|---|---|
| $d_s$ | 192 | 128 | 64 | 64 | 64 | 64 | 64 | 32 | 32 | 32 | 32 |
| $\text{Var}_s$ ($\times 10^3$) | 0.3 | 0.3 | 0.2 | 0.2 | 0.3 | 0.3 | 0.4 | 0.3 | 0.3 | 0.4 | 10.7 |
| $\Delta LD_s$ ($\times 10^{-1}$) | 14.8 | 11.6 | 7.6 | 4.2 | 3.1 | 2.5 | 2.9 | 1.4 | 1.6 | 0.9 | 2.2 |
| $\frac{\Delta LD_s}{d_s}$ ($\times 10^{-3}$) | 7.7 | 9.1 | 11.9 | 6.6 | 4.8 | 3.9 | 4.6 | 4.4 | 5.0 | 2.9 | 6.8 |
| $\frac{\Delta LD_s}{\text{Var}_s}$ ($\times 10^{-4}$) | 49.5 | 37.6 | 38.5 | 18.1 | 11.0 | 7.5 | 7.1 | 5.5 | 5.2 | 2.2 | 0.2 |

Table 3: Test 2 raw data for PCA 1 baseline results shown in main paper (Table 1). Gini coefficient for row $\Delta LD_s$ is 0.46, for row $\frac{\Delta LD_s}{d_s}$ is 0.22, and for row $\frac{\Delta LD_s}{\text{Var}_s}$ is 0.52.

| index $s$ | 0 | 1 | 2 | 3 | 4 | 5 | 6 | 7 | 8 | 9 | 10 |
|---|---|---|---|---|---|---|---|---|---|---|---|
| $d_s$ | 128 | 128 | 96 | 96 | 64 | 64 | 64 | 32 | 32 | 32 | 32 |
| $\text{Var}_s$ ($\times 10^3$) | 0.7 | 1.0 | 0.5 | 7.9 | 0.4 | 0.2 | 0.2 | 0.1 | 0.1 | 0.1 | 0.1 |
| $\Delta LD_s$ ($\times 10^{-1}$) | 3.9 | 3.7 | 2.3 | 2.5 | 1.6 | 1.7 | 1.7 | 0.7 | 0.8 | 0.8 | 0.7 |
| $\frac{\Delta LD_s}{d_s}$ ($\times 10^{-3}$) | 3.0 | 2.9 | 2.4 | 2.6 | 2.4 | 2.6 | 2.7 | 2.2 | 2.4 | 2.4 | 2.2 |
| $\frac{\Delta LD_s}{\text{Var}_s}$ ($\times 10^{-4}$) | 5.5 | 3.6 | 4.7 | 0.3 | 4.0 | 8.1 | 9.1 | 5.2 | 8.0 | 7.7 | 7.0 |

Table 4: Test 1 raw data for identity baseline results shown in main paper (Table 1). Gini coefficient for row $\Delta LD_s$ is 0.33, for row $\frac{\Delta LD_s}{d_s}$ is 0.05, and for row $\frac{\Delta LD_s}{\text{Var}_s}$ is 0.23.

## D.5 ABLATION STUDY ON GPT-2 TEST SUITE

We also show hyperparameter searching results in Table 5, in order to provide information on how each factor affects the training result. First of all, we can see it is better to have larger merging interval, in other words, more training steps. Due to compute limits, we do not further experiment with more training steps, which would probably produce better results. Moreover, we show the same data in Fig. 16, using search number as the variable on x-axis. We can see that larger search number generally produces better results, though depending on merging threshold. However, considering both search number and merging interval, when given fixed compute budget, it is a trade-off between searching among more activations per step and performing more training steps. Determining which side we should spend more compute on is an important question and should be explored more extensively in future work.

We also consider other alternatives for other hyperparameters. In Table 6 we compare different distance metric, under different condition. We use good hyperparameters for merging threshold (0.04 or 0.05) and merging interval (8000) based on our previous finding. When using cosine similarity as distance metric, losses (average distances) from different subspaces are weighted by the subspace dimensions. We found it important, because in some preliminary experiments where there is no such weights, the orthogonal matrix prioritizes maximizing cosine similarity for small subspaces while the big subspaces are somewhat ignored. From the results we can see that cosine similarity produces similar or worse results, which is consistent with our choice in toy setting.

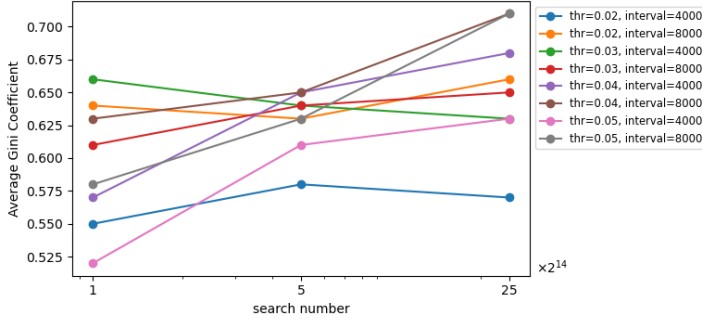

Figure 16: Average Gini coefficient on GPT2 test suite, as a function of search number, under different conditions. Same data as Table 5. X-axis is in log-scale.

| search number ($\times 2^{14}$) | merging threshold | merging interval | number of subspaces | avg coefficient |
|---|---|---|---|---|
| 1 | 0.02 | 4000 | 6,8,6,5 | 0.55 |
| 1 | 0.02 | 8000 | 9,9,9,9 | 0.64 |
| 1 | 0.03 | 4000 | 9,11,11,10 | 0.66 |
| 1 | 0.03 | 8000 | 9,11,11,10 | 0.61 |
| 1 | 0.04 | 4000 | 12,13,12,14 | 0.57 |
| 1 | 0.04 | 8000 | 12,12,13,14 | 0.63 |
| 1 | 0.05 | 4000 | 14,16,17,19 | 0.52 |
| 1 | 0.05 | 8000 | 13, 17,18,18 | 0.58 |
| 5 | 0.02 | 4000 | 5,8,7,6 | 0.58 |
| 5 | 0.02 | 8000 | 9,9,9,9 | 0.63 |
| 5 | 0.03 | 4000 | 8,8,9,10 | 0.64 |
| 5 | 0.03 | 8000 | 9,9,9,9 | 0.64 |
| 5 | 0.04 | 4000 | 11, 12, 14, 14 | 0.65 |
| 5 | 0.04 | 8000 | 10, 12, 13, 13 | 0.65 |
| 5 | 0.05 | 4000 | 12, 15, 16, 17 | 0.61 |
| 5 | 0.05 | 8000 | 12, 15, 16, 16 | 0.63 |
| 25 | 0.02 | 4000 | 5, 6, 5, 6 | 0.57 |
| 25 | 0.02 | 8000 | 9,9,9,9 | 0.66 |
| 25 | 0.03 | 4000 | 7, 9, 8, 10 | 0.63 |
| 25 | 0.03 | 8000 | 9, 9, 9, 10 | 0.65 |
| 25 | 0.04 | 4000 | 10, 10, 12, 13 | 0.68 |
| 25 | 0.04 | 8000 | 11, 11, 12, 13 | 0.71 |
| 25 | 0.05 | 4000 | 10, 13, 15, 16 | 0.63 |
| 25 | 0.05 | 8000 | 12, 15, 14, 15 | 0.71 |

Table 5: Combinations of different search number, merging threshold, and merging interval, and their resulting number of subspaces in the 4 layers used in the test (from low to high) and resulting average Gini coefficient on GPT-2 test suite. The one shown in Table 1 is the third row from the last, which is slightly better than the last row if we show more precision. Other hyperparameters are fixed, and described in D.3.

| search number ($\times 2^{14}$) | merging threshold | avg coefficient | |
|---|---|---|---|
| | | dist=1-cosine | dist=euclidean |
| 1 | 0.04 | 0.62 | 0.63 |
| 1 | 0.05 | 0.58 | 0.58 |
| 5 | 0.04 | 0.64 | 0.65 |
| 5 | 0.05 | 0.64 | 0.63 |
| 25 | 0.04 | 0.65 | 0.71 |
| 25 | 0.05 | 0.68 | 0.71 |

Table 6: Average Gini coefficient on GPT2 test suite. We set merging interval as 8000. The last column comes from Table 5. Compared to euclidean distance, using cosine similarity produces in similar or slightly worse results

| search number ($\times 2^{14}$) | merge threshold | avg coefficient | |
|---|---|---|---|
| | | data=OpenWebText | data=MiniPile |
| 1 | 0.04 | 0.54 | 0.63 |
| 1 | 0.05 | 0.54 | 0.58 |
| 5 | 0.04 | 0.59 | 0.65 |
| 5 | 0.05 | 0.58 | 0.63 |
| 25 | 0.04 | 0.61 | 0.71 |
| 25 | 0.05 | 0.60 | 0.71 |

Table 7: Average Gini coefficient on GPT2 test suite. We set merging interval as 8000. The last column comes from Table 5. Compared to MiniPile, using OpenWebText produces worse results. Closer look shows that, instead of degrading uniformly across all tests, using OpenWebText only hurts performance on test 2 - 5.

In Table 7 we compare different data sources, i.e., the text data on which we run GPT-2 Small to collect activations. Again, we use merging threshold of 0.04 or 0.05, and merging interval of 8000. We can see that OpenWebText produces worse results in all cases, some of them are much worse. After checking Gini coefficient for individual tests, we found that using OpenWebText only hurts performance on test 2 - 5. We speculate that it is probably due to greater diversity in MiniPile.

A key hyperparameter is unit size, in the paper we normally use attention head dimension divided by 2. Though we think this is a very important aspect, we haven't done extensive testing on different unit sizes as the search space is big and our compute budget is small. Moreover, the interaction of unit size and search number is also worth exploring, big search number might be less helpful when subspace dimension is small, thus we could spend compute on more training steps. More importantly, there could be many other changes, one could set search number depending on the dimension of the subspace instead of using the same search number for all subspaces throughout the whole training process, one could also do merging until a target number of subspaces is reached, instead of based on fixed threshold. In sum, we think there are many possibilities, we have only explored a small part of them, there is a lot of potential improvements for NDM.

### D.6    Knowledge Conflict Experiment

Unlike the original purpose of RAVEL dataset, we do not aim to find separate subspaces for different types of attributes. Because we find many attributes given in RAVEL have high MI, e.g., the continent and country of a city, we think these attributes would not "naturally" been placed in separate subspaces, even if so, our method might merge them because of high MI. In contrast, we think there might be separate circuits routing context and parametric knowledge, and subspaces involved in these two circuits might be more separable.

We filter entities to make sure language models have parametric knowledge about them. Specifically, for each entity in RAVEL prize winner split, we prompt the model to complete "A. Michael Spence won the Nobel Prize in Economics. [target entity] won the Nobel Prize in", and only keep entities where model successfully predicts the ground truth. Moreover, we remove entities whose ground truth answer has more than one token. This includes "Linus Pauling" and "Marie Curie". This process produces 552 entities for Qwen2.5-1.5B and 719 for Gemma-2-2B after filtering, among total 927 entities in RAVEL's prize winner data split.

We only use Nobel prize winner data in RAVEL, because it satisfies several conditions: 1) Language models do not completely lean to one side. When there is no intervention, Original model choice should be ideally equally distributed between the context answer and parametric answer. Nobel prize winner data has a good distribution for both Qwen2.5-1.5B and Gemma-2-2B with this respect, make it possible to observe effect on both sides. 2) It should not be too hard for 2B models, such that we have enough instances to give a reliable result. 3) Answers are single tokens for the ease of implementation (not a real problem).

We patch subspace activations at all token positions together. In order to keep a simple one-to-one relationship between patching source and destination tokens, the made-up entity always has the same number of tokens as the target entity. We do so by prompting GPT-4o to generate random names that resemble real names, but are not associated with any well-known person. The length of generated

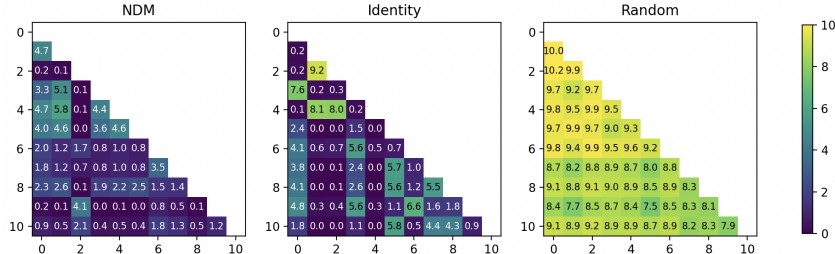

Figure 17: MI between each pair of subspaces. Activation site: $h^{11,mid}$. NDM: subspaces given by NDM. Identity: subspaces given by the identity matrix and the dimensionality configuration from NDM. Random: subspaces given by the identity matrix and the dimensionality configuration from NDM. We use the same MI estimation method as the one used during training. The MI values are not normalized. We can see NDM produces more independent subspaces.

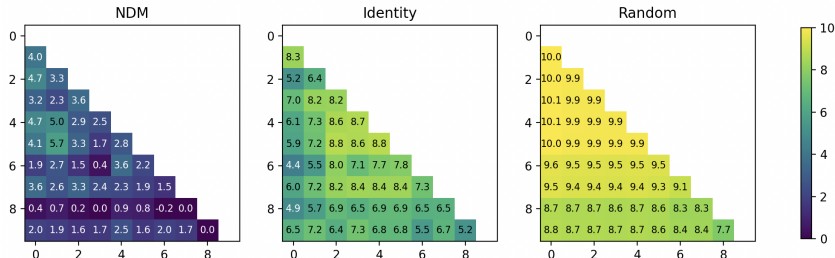

Figure 18: MI between each pair of subspaces. Activation site: $h^{9,mid}$. See the previous Fig. 17 for explanation.

names varies, ensuring that each target entity has at least one made-up name of the same number of tokens.

Hyperparameters for NDM are the same as those for GPT-2 (App. D.3), except for the following ones: We use 64 as the unit size for Qwen2.5-1.5B and 72 for Gemma-2-2B. We use $N = 25 \times 2^{14}$ as the search number. We set 150k as maximum training steps. We try different merging threshold $\{0.015, 0.02, 0.03, 0.04\}$ for Qwen and add one additional value 0.01 for Gemma. We try all these combinations and choose the best one. For Qwen2.5-1.5B, it is threshold = 0.02, merging interval = 8000; FOr Gemma-2-2B, it is threshold = 0.01, merging interval = 4000; We apply NDM to $h^{11,mid}$ of Qwen2.5-1.5B (totally 28 layers) and $h^{9,mid}$ of Gemma-2-2B (totally 26 layers). We choose the layer according to preliminary experiments where we patch the whole activation at the last token position of target entity.

After training the estimated MI decreases significantly, see Fig. 17 and 18. Interestingly, as we can see, the original basis of the representation space of Qwen2.5-1.5B is quite special, simply partition on standard basis can produce many independent subspaces.

We show baseline results in Fig. 19 and 20.

**Some caveats for interpreting results** We say the two mechanisms is mediated by different subspaces, instead of saying these subspaces encode contextual/parametric knowledge, as we do not know the underlying circuit, encoded information could also be others that support information routing, e.g., the value of target entity (as a key for retrieval), the fact that the entity is repeated in the context (may trigger certain operation). They are different types of information and may be encoded in separate subspace, so effect does not have to be concentrated in one subspace. Moreover, it is also understandable if certain subspace is used in both mechanisms, thus having effect on both sides. So even if a subspace partition is perfect, it does not necessarily mean *all* subspaces only have effect on one side, but finding such subspaces is nevertheless a good sign. In addition, it is not necessarily the case that the two mechanism are using different subspaces in all layers. In early layers the current

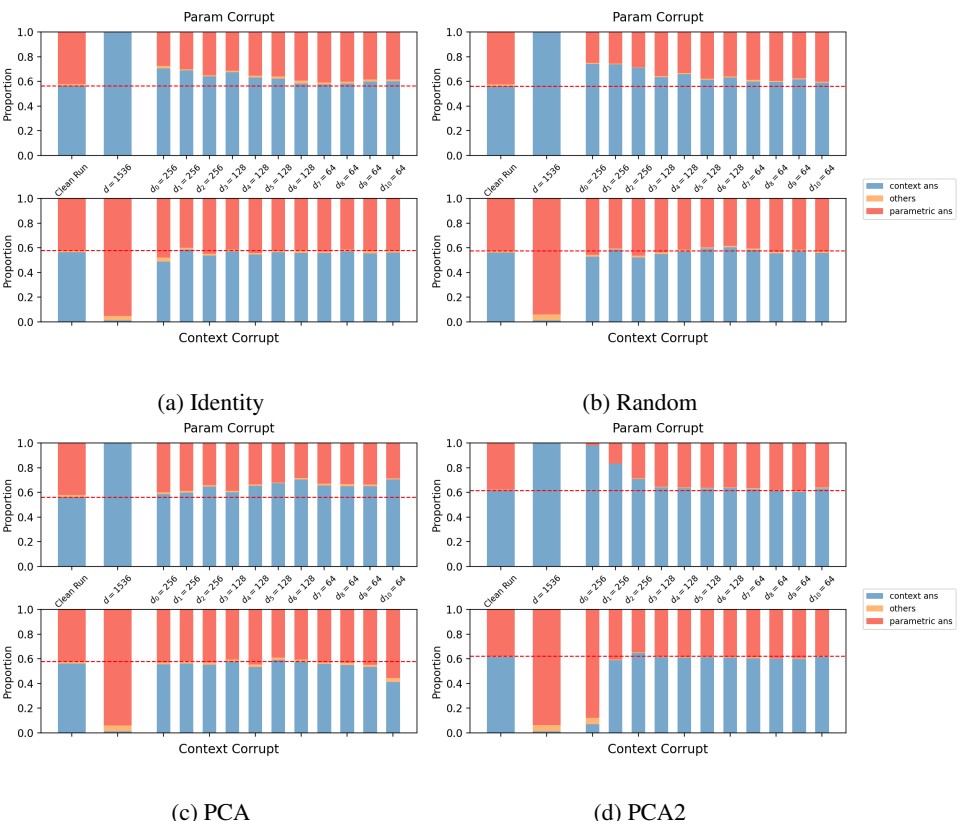

Figure 19: Subspace patching effect. Partition given by an identity matrix, a random orthogonal matrix, eigenvectors whose corresponding eigenvalues are in ascending order, eigenvectors whose corresponding eigenvalues are in descending order, on $h^{11,mid}$ of Qwen2.5-1.5B. This figure supplements the main paper Fig. 5 (the right one) with baselines. We can see the effect under two types of counterfactual inputs tend to correlate.

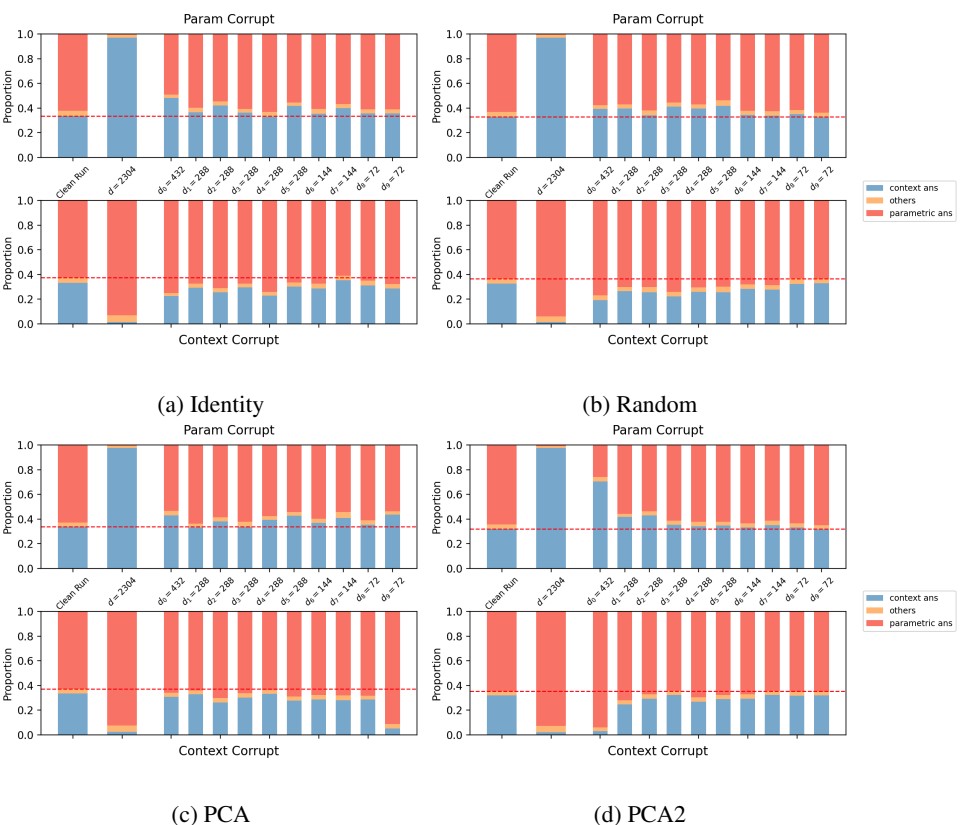

Figure 20: Subspace patching effect. Partition given by an identity matrix, a random matrix, eigenvectors whose corresponding eigenvalues are in ascending order, eigenvectors whose corresponding eigenvalues are in descending order, on $h^{9,mid}$ of Gemma-2-2B. This figure supplements the main paper Fig. 5 (the right one) with baselines. We can see the effect under two types of counterfactual inputs tend to correlate.

token subspace can affect both mechanism, while in later layers, the conflict may have been resolved and the final result is encoded in one subspace. The middle layer is likely the place where retrieval is already happened but one type of knowledge is not yet overwritten by the other.

# E  QUALITATIVE ANALYSIS DETAILS

## E.1  BUILDING ACTIVATION DATABASE

As we mentioned in main paper, we apply InversionView to subspaces and simply do vector search to find preimages. Here are the details: (1) The text data includes 2000 random sequences in MiniPile, 1000 IOI prompts and 1000 Greater-than prompts. We made IOI and Greater-than prompts using the same procedure as we did for clean inputs in patching experiments. We run GPT-2 Small on these text, producing 1178970 activations for each activation site (i.e., post MLP residual stream at layer 4, 6, 8, 9). Each activation has a corresponding token position from which it is collected. We maintain their correspondence such that we can always retrieve the input tokens that result in certain activation (i.e., all tokens in its previous context). (2) For a subspace partition trained for an activation site, i.e., orthogonal matrix $\mathbf{R}$ and configuration $c$, we make $S$ (number of subspaces) databases by using FAISS (Douze et al., 2024) index. Activations in whole space are converted into subspace activations (multiplied by $\mathbf{R}$ and split according to $c$) and are added to each database respectively. Therefore, we make separate databases for each layer (activation site) and each subspace. (3) Then for any interested subspace activation (query activation), we search for other activations in corresponding subspace database whose cosine similarity is above the threshold. We then show the inputs corresponding to query activation and searching results.

## E.2  INPUT ATTRIBUTION DETAILS

As language model's representations are contextualized, sometimes activations encode information far from the token position it corresponds to, i.e., long-range dependency. This makes interpretation difficult, as we need to search commonality in a large context. We thus make use of input attribution method to highlight important tokens. We use integrated gradients with 10 interpolations. Specifically, given an query subspace activation and a sequence of tokens, we first obtain the initial residual stream activations from these tokens, and multiply them with $\alpha = 0.1, 0.2, \ldots, 1.0$ to create 10 interpolations. We continue the forward pass to get subspace activations. Importantly, we freeze all attention weights to the values they take when running on clean inputs ($\alpha = 1.0$) during the forward pass. We finally compute the cosine similarity between resulting subspace activation and the given query subspace activation. The gradient (not via attention weights) of the cosine similarity with respect to each interpolations are averaged and multiplied with residual stream activations to obtain the final token-level attribution score. The scores are normalized by the sum of scores of all tokens in the same input sequence before showing in the web application. So the scores sum to 1. But they usually spread somewhat evenly across many tokens, resulting in invisible highlights, so we also provide options in the web application to scale these scores together. Because it requires time to compute attribution scores, we pre-compute them for 50 random query activations and show them in a separate page in the application.

The reason for fixing attention weights is that, we empirically find it produces more plausible attribution. The idea behind it is to emphasize what is moved by the attention circuit, while ignoring how this circuit is formed. For example, for a induction head on input "A B ... A", it would only highlight "B" though all 3 tokens play a role. The same intuition is also used in other works (Ameisen et al., 2025). Importantly, we find the effectiveness of our attribution method is limited, as the scores are usually too concentrated at the most recent token, readers should be cautious when using them. This also highlights importance of better input attribution, as in larger and more capable models long-range dependency might be more prevalent.

## E.3  MORE EXAMPLES AND DETAILED FINDINGS

**Examples supplementing Fig. 3**  Fig. 21 - 24 correspond to 4 subfigures in the main paper Fig. 3, in each of them we fix the subspace and vary inputs to observe the consistency.

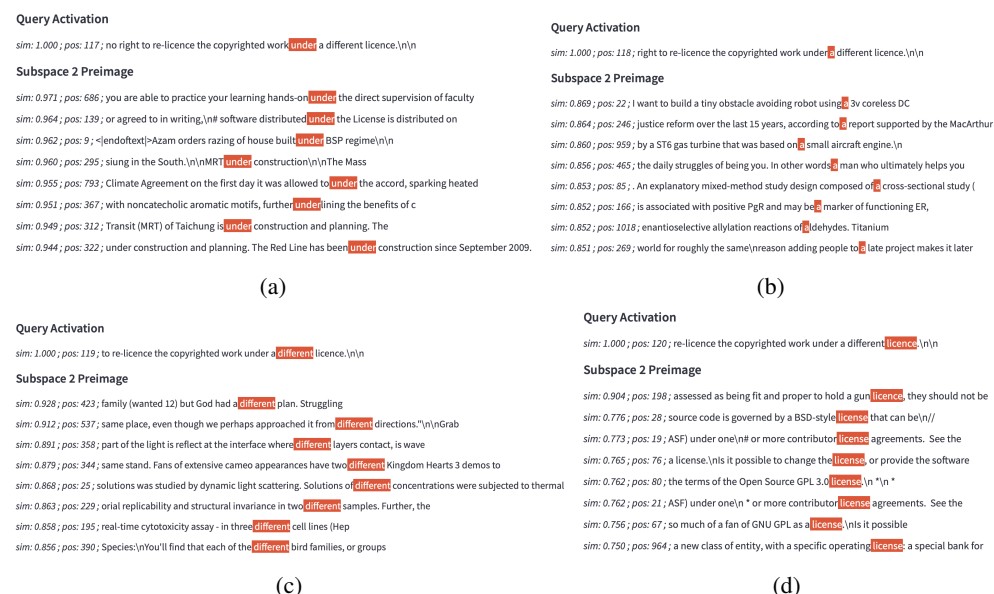

Figure 21: Subspace 2 at $\boldsymbol{h}^{4,post}$: Preimages of the residual stream activation corresponding to a continuous span of tokens ("under", "a", "different", "licence") in the same subspace after the 5th layers in GPT-2 Small. Interpretation of encoded information: (a) the current token "under". (b) the current token "a". (c) the current token "different". (d) the current token "license". But there are not enough samples above cosine similarity of 0.85, we lower the threshold to 0.75 to check more samples. It is probably because it encodes more than just the current token.

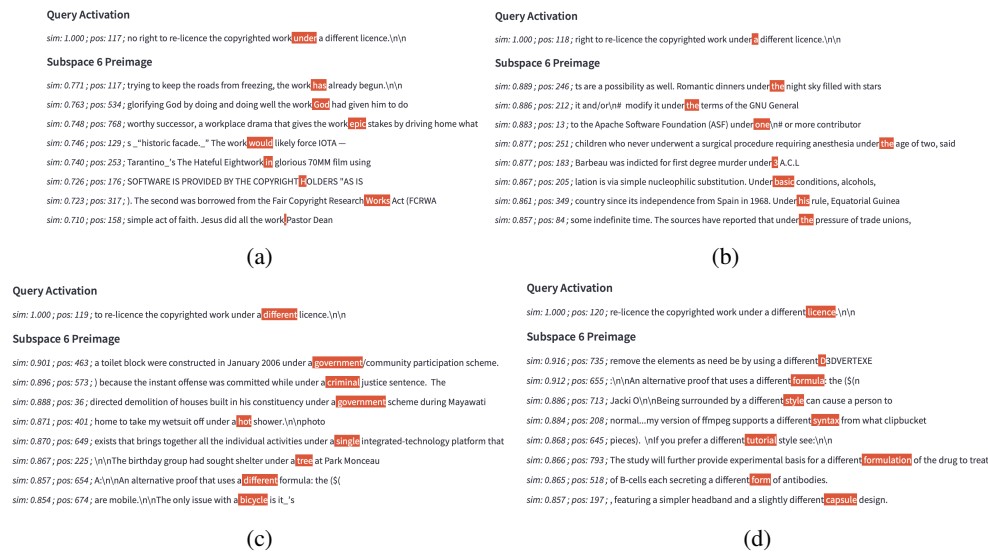

Figure 22: Subspace 6 at $\boldsymbol{h}^{4,post}$: Preimages of the residual stream activation corresponding to a continuous span of tokens ("under", "a", "different", "licence") in the same subspace after the 5th layers in GPT-2 Small. Interpretation of encoded information: (a) the previous two tokens "copyrighted" and "work". We have to lower the threshold to 0.7 to have enough samples inside preimage. In those samples, we can see some contain "work" and others contain "COPYRIGHT" in previous tokens. We think that both tokens are encoded, that is why having only one of them cannot produce high similarity. (b) the previous token "under". (c) the previous token "a". (d) the previous token "different". This subspace is the one with the largest patching effect in test 1. Its un-normalized effect is 3.60, much greater than the second largest value 0.03.

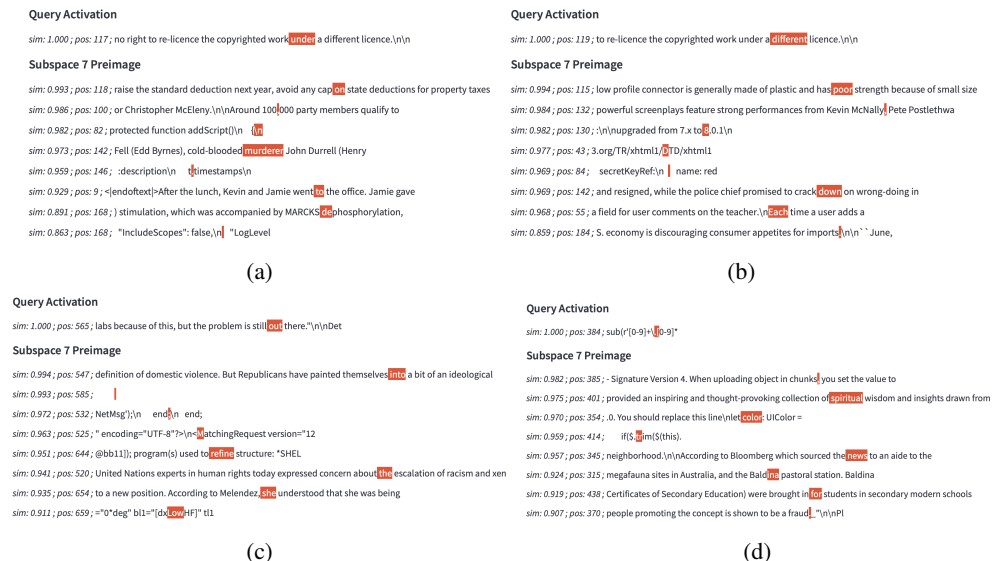

Figure 23: Subspace 7 at $h^{4,post}$: Preimages of the residual stream activation corresponding to two tokens adjacent to the one in Fig. 3(c) ("under", "different"), and two tokens in very different positions (at position 565 and 384) in the same subspace after the 5th layers in GPT-2 Small. Interpretation of encoded information: (a) current position is around 117. (b) current position is around 119. (c) current position is around 565. (d) current position is around 384. For this subspace, one can increase the threshold and obtain samples of very similar positions as shown in Figure 31

.

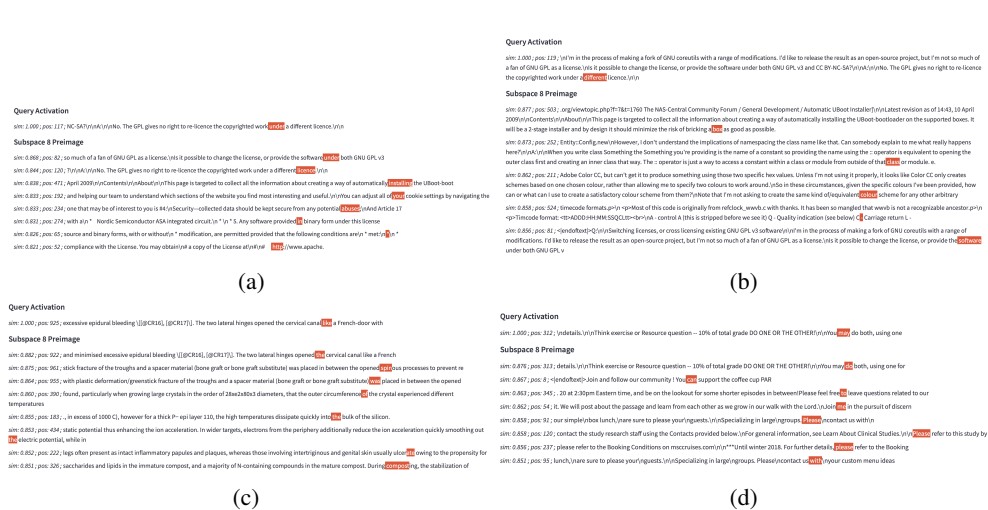

Figure 24: Subspace 8 at $h^{4,post}$: Preimages of the residual stream activation corresponding to two tokens adjacent to the one in Fig. 3(c) ("under", "different"), and two tokens in very different context in the same subspace after the 5th layers in GPT-2 Small. Interpretation of encoded information: (a) context is about software license/permission. (b) discussing software development. (c) describing detailed medical or physical processes in technical contexts. (d) online advertisements and announcements, more concretely, suggesting readers to do something. We can see as (a) and (b) are close, the topic information is also similar, while it is very different from (c) and (d)

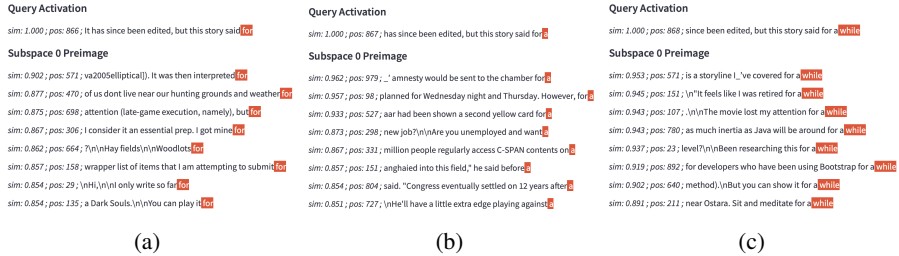

(a)  (b)  (c)

Figure 25: Subspace 0 at $h^{4,post}$: Preimages of the residual stream activation corresponding to three tokens in a continuous span. While encoded information for (a) and (b) are the current token ("for", "a"), the encoded information for (c) is "for a while" instead of just "while" (we confirm this by checking more samples and varying threshold). So the high-level interpretation of the subspace is not current token, but more like current semantic unit. This also shows that sometimes several recent tokens are merged in representation, but sometimes they are not, indicating some kind of "internal tokenization".

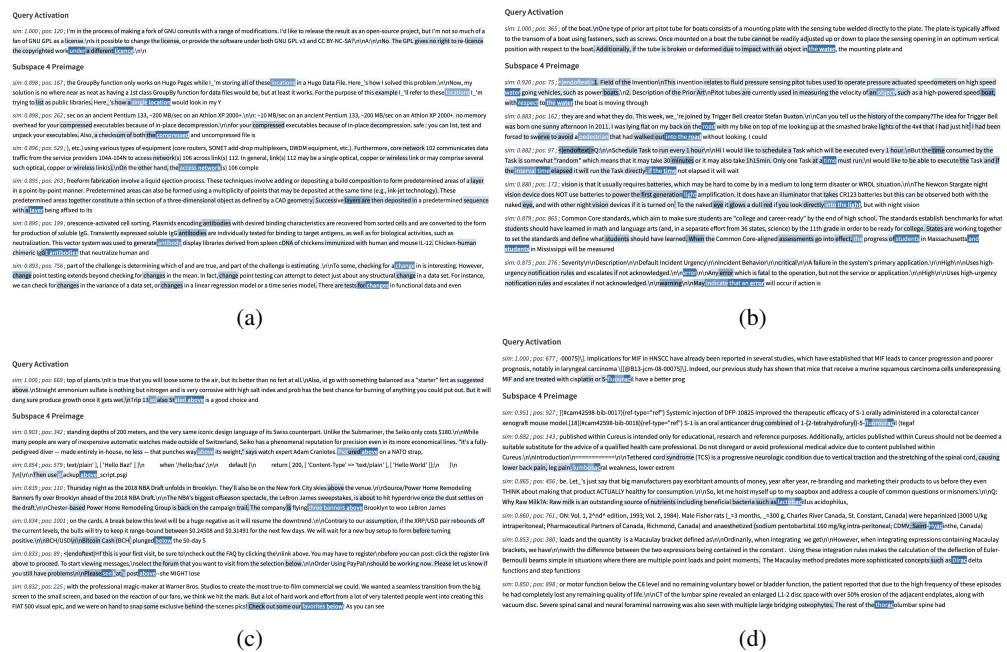

(a)  (b)

(c)  (d)

Figure 26: Subspace 4 at $h^{4,post}$: Preimages of the residual stream activation. Different from other preimage figures, we show input attribution scores here, because the encoded information depends on long-range context, we need these hints to help us find commonality. The blue color represent the attribution score (see Sec. E.2 for details), they are amplified by a factor of 20 for visibility. The token where the activation is taken from is marked by bold and underscored font. Encoded information is: (a) the current token is repeated multiple times. (b) The current token is repeated multiple times. Note that we check the entire previous context of the query activation, and there are indeed more "water" tokens there. We don't show it due to space limit. (c) The current token "above" (if we only consider similarity > 0.85) (d) The current token "ac". We checked the entire context and found the token "ac" is not repeated. Overall, we found the following pattern: when the current token is relatively less frequent and is repeated, this subspace clusters together activations corresponding to different tokens, otherwise it clusters activations according to current token. The signal might be important to activate induction heads in later layers.

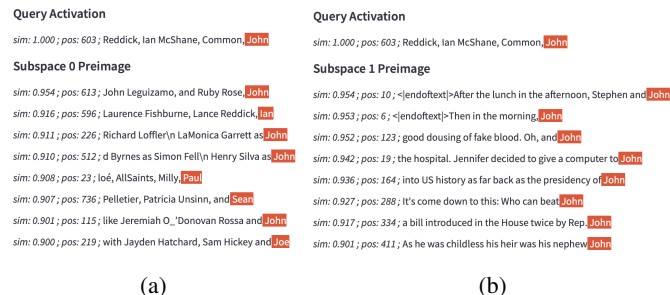

Figure 27: Subspace 0 and subspace 1 at $h^{4,post}$: Preimages of the residual stream activation corresponding to the same input. Both subspace 0 and 1 in this case encode local context, but they are slightly different: subspace 0 encodes a more coarse-grained information for current token (i.e., a first name) and encodes a slightly longer context (i.e., there are more than one name in context); subspace 1 encode shorter context (only the current token) but more fine-grained information (i.e., exactly "John").

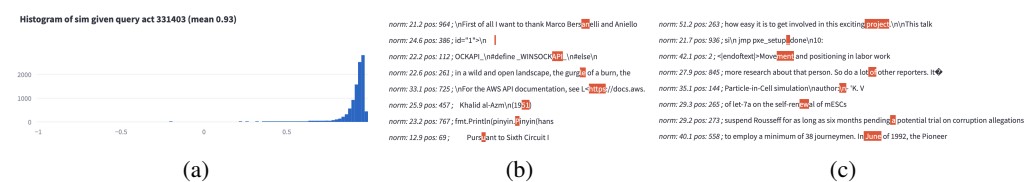

Figure 28: Subspace 9 at $h^{4,post}$: (a) we show the histogram of cosine similarity between a activation and all other activations in the subspace. We can see all activations in the subspace are in similar direction. (b) (c) we show some examples inputs and the L2 norm of their corresponding activations in this subspace, we can see that while direction is similar, the norm can vary between 10-50.

**Subspace interpretation is consistent while being dependent on the input**    When we say the subspace's interpretation or function is consistent, it does not mean it must align exactly with any human concept. For example, as shown in Fig. 25, subspace 0 encodes only the current token in first two subfigures, but it encodes 2 more previous token in the last subfigure. This does not contradict with our claim of consistency, as the 3 tokens form a frequent phrase, and can be regarded as one.

More importantly, as shown in Fig. 26, we found subspace interpretation can depend on input in an interesting and different way: the subspace encodes the occurrence of repetition if the current token is relatively less common and repeated, otherwise it "falls back" to encode the current token. This behavior is understandable, certain meaningful pattern does not always occur, the subspace might have an "default/empty state". When it occurs, it overwrites the information in the subspace. We also think the phenomenon for subspace 4 in Fig. 26 might be because the subspace partition is not done very well, so that it mixes "repetition subspace" with "current token subspace".

**Several subspaces encode local context, while being slightly different**    We found that subspace 0, 1, 2, 3, 5 encode short-range local context, most of the time they only encode information about most recent 1-2 tokens. But they are slightly different. Fig. 27 shows an example, where subspace 0 encodes a more coarse-grained information for current token but a slightly longer context, and in contrast subspace 1 encodes a more fine-grained information for current token but shorter context. Among these subspaces, we found subspace 2 encodes mostly only the exact current token.

**Some subspaces are hard to explain**    We also find the remaining two subspaces 9 and 10 hard to explain. For subspace 9, we find that, somewhat surprisingly, almost all activations are in a similar direction, as shown in Fig. 28. We also find the norm of activations in this subspace varies, the main variance in this subspace is from this direction. If this is a single feature, what is the pattern that affect the magnitude in this direction? However, we do not have very confident conclusion for this question. We find that the norm is almost always small if the current token is in the middle of a multi-token word (e.g., Fig. 28(b)'s bottom row) and very big $\approx 2000$ for "end of sequence"

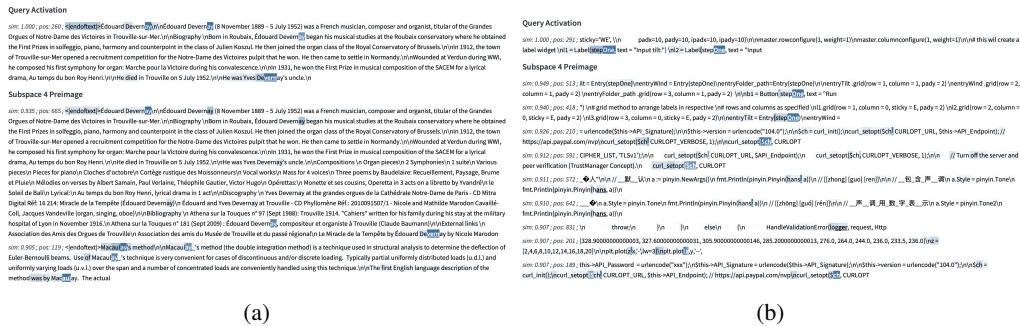

(a)                                                                                          (b)

Figure 29: Subspace 4 at $h^{6,post}$: The subspace is the one with the highest patching effect in test 2 described in Sec. 5.1, so it should encode output of induction heads. (a) "ay" is moved by induction mechanism. We show the entire context for each input. (b) "," is moved by induction mechanism. We check the entire context and confirm that the first, second sample's context indeed contain a ("One", ",") combination, and the third from the last indeed contains a ("er", ",") combination. We scale attribution scores by 10 or so. Importantly, we *pick* these two on purpose, because in many case the induction mechanism is not "activated". In those cases the information is more about the current token. In many other cases there are simply not enough samples in preimage due to limited size of our activation database.

(EOS) token. One point we can believe is that, the information encoded in this direction is quite independent with the information in other subspaces. Interestingly, subspace 9 in $h^{8,post}$ also shows this pattern, i.e., all activations point to similar direction and big norm for EOS. But subspace in that layer is the one with the highest effect in test 3 described in Sec. 5.1: it has an un-normalized effect of 2.31 ($d_9 = 32$), the second and third largest is 0.78 ($d_2 = 96$) and 0.30. This means such an subspace indeed encodes certain meaningful information. In IOI domain it is the position of the subject (outputs from S-Inhibition heads), but it might be part of a more general mechanism, like a signal to not attend to certain part of context. That explains why it is so hard to interpret.

Coming back to layer 4, for subspace 10, we also see many activations in the same direction, but not as often as subspace 9. For both 9 and 10, we do not have a clear conclusion for the information encoded by them. But we believe they are worth more extensive study, as it might reveal some unexpected features used by the model.

**Summary of subspace interpretation for $h^{4,post}$**. So far we have described every subspace we found for one layer. Here is a brief overview:

- Subspace 0 ($d_0 = 128$), 1 ($d_1 = 128$), 2 ($d_2 = 96$), 3 ($d_3 = 96$), 5 ($d_5 = 64$): short-range local context, usually only most recent few tokens. They encode slightly different information.

- Subspace 4 ($d_4 = 64$): Repetition in the context.

- Subspace 6 ($d_6 = 64$): Previous token, almost excluding the current token.

- Subspace 7 ($d_7 = 32$): Position information.

- Subspace 8 ($d_8 = 32$): Longer-range context, topic or the main described object.

- Subspace 9 ($d_9 = 32$), 10 ($d_{10} = 32$): No clear conclusion.

**Subspaces in later layers** After exhaustively exploring subspaces in the residual stream after layer 4 (the 5th layer), we move on to take a brief look at subspaces in later layers. We show some preimages in Fig. 29 and Fig. 30, which correspond to two subspaces with the highest patching effect in quantitative evaluation. The web application includes results for all 4 layers (4, 6, 8, 9) used in quantitative evaluation, we encourage readers to check our web application.

To check interpretability of subspaces in later layers, we also try to interpret every subspace of layer 9, and summarize our findings as follows:

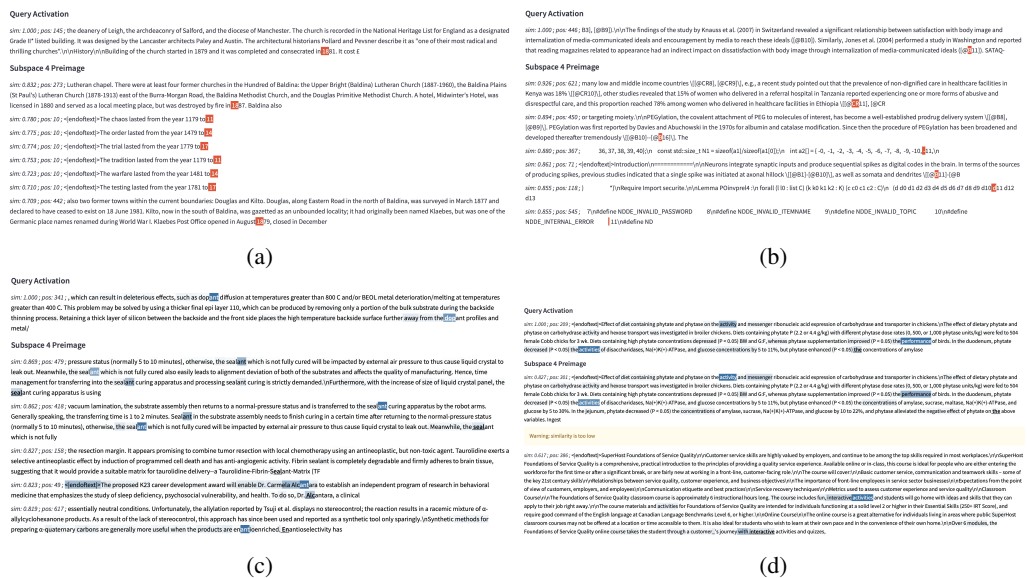

Figure 30: Subspace 4 at $h^{9,post}$: The subspace is the one with the highest patching effect in test 5 described in Sec. 5.1, the effect of all subspaces is shown in Table 2. So it should encode the starting year information in Greater-than prompts. (a) Encoded information is (approximately) "79". We can see how the same mechanism in Greater-than prompts is being used in more natural text. Note that we lower the threshold to 0.7 in order to include more samples. We can see though having the same starting year "79", the four Great-than prompts are not close enough to query activation, so some other context is also slightly encoded in this subspace. (b) Encoded information is "10". Here we can see how this circuit is useful in a broader domain. (c) Encoded information is "ant". We can see this subspace is essentially a induction output subspace. (d) It seems that the encoded information is a mixture of "activities" and "performance", having only one of them (like the last sample) is not enough to get close to the query activation. So the induction circuit is triggered quite often, it can be used to predict the noun after "the".

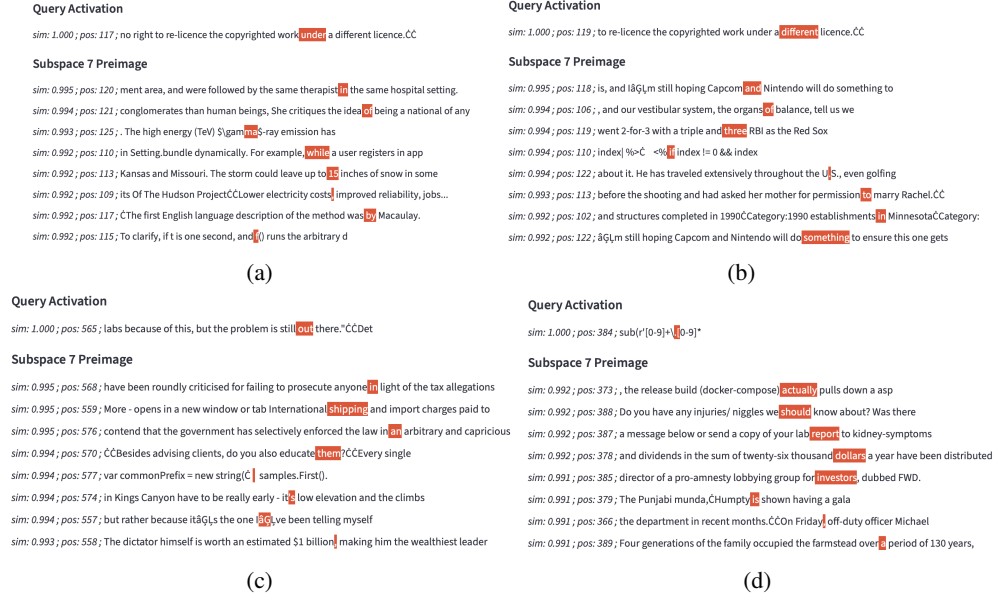

Figure 31: Subspace 7 at $h^{4,post}$: Same examples as Figure 23, but the threshold for cosine similarity is increased to 0.99, we obtain samples of very similar positions. Remaining error could be because the optimization is not optimal, or the model has forgotten the information a bit at this layer.

- Subspace 0 ($d_0 = 128$), 1 ($d_1 = 96$), 2 ($d_2 = 96$), 3 ($d_3 = 64$), 8 ($d_8 = 32$): local context. Compared to layer 4, they tend to cover a longer span of context, such as a sub-clause. They encode slightly different information.

- Subspace 4 ($d_4 = 64$): Induction output.

- Subspace 5 ($d_5 = 64$): Longer-range context, topic or the main described object.

- Subspace 6 ($d_6 = 32$): Certain kind of syntactic role of the current token, not necessarily aligned with human definition, such as whether the current token is the subject, or starting/ending of a sub-clause/sentence.

- Subspace 7 ($d_7 = 32$): Position information.

- Subspace 9 ($d_9 = 32$), 10 ($d_{10} = 32$), 11 ($d_{11} = 32$), 12 ($d_{12} = 32$): No clear conclusion. Subspace 10 shows similar pattern as subspace 9 of layer 4 residual stream $h^{4,post}$.

Again, we see some subspaces that are hard to interpret, they tend to be quite independent from other subspaces (Fig. 15(d)).

### E.4 Subspaces for Parametric Knowledge Routing in Qwen2.5-1.5B

In Fig. 5(a), we see that there are two subspaces, 2 and 6, playing important roles for parametric knowledge routing. We apply the same qualitative analysis to see what is the information encoded in them, and if they differ in some way. Like in quantitative tests, we test and remove entities that model does not have knowledge for. We randomly sample 250 entity pairs from them, each is used to construct a 2-shot example. We then run Qwen2.5-1.5B on these examples, together with 2000 text sequences from MiniPile, and collect activations to build the database. Fig. 32 shows some examples. As we can see, they indeed differ. Though they both contain retrieved parametric knowledge, subspace 2 encodes the knowledge about the current n-gram, while subspace 6 encodes knowledge that does not depend on current token. We therefore have the following hypothesis: when enough tokens are observed and the name is identifiable, MLP retrieves knowledge and places it into subspace 2, this knowledge is then emitted by attention heads to many other tokens' residual stream (or in other words, attended by other tokens), this information arrives at certain subspace similar to subspace 6, but at later layers. This behavior is probably distributed and repeated across multiple middle layers, so subspace 6 in this layer receives information from certain subspaces in earlier layer that resemble subspace 2 of this layer. So the information transfer is not done all at once. To verify this hypothesis, more rigorous experiments should be done. We encourage readers to check more examples. Due to its large size and various limits we did not manage to deploy the database for Qwen2.5-1.5B on our web application. But we released all trained orthogonal matrices together with the code. Readers can follow the instructions in our repository and easily reproduce the database.

## F Comparison to Sparse Autoencoders (SAEs)

Due to the popularity of SAEs, in this section, we explain the difference and connections between them and our method. Readers who are familiar with SAEs might find this section particularly helpful.

### F.1 FAQ

**(1)** *Why not compare with SAEs?*

**Quantitative evaluation** In previous quantitative evaluations, we patch each subspace and see its average effect on the output. SAEs do not provide such subspaces directly. If we have to make a comparison, we can change the evaluation method to apply an analogous evaluation to SAEs. One way is to patch the basis elements that SAEs provide, the features. This means that we replace a feature's activation value with its value on the corrupted input. For example, in the clean input on the token after "John", a feature representing *"previous token is John"* is highly activated – while, in the same position in the corrupted input, a feature representing *"previous token is Mary"* is highly activated. If we patch the "prev-Mary" feature, the new activation would say *"previous token is both John and Mary"*. If we patch the "prev-John" feature, the new activation has no information

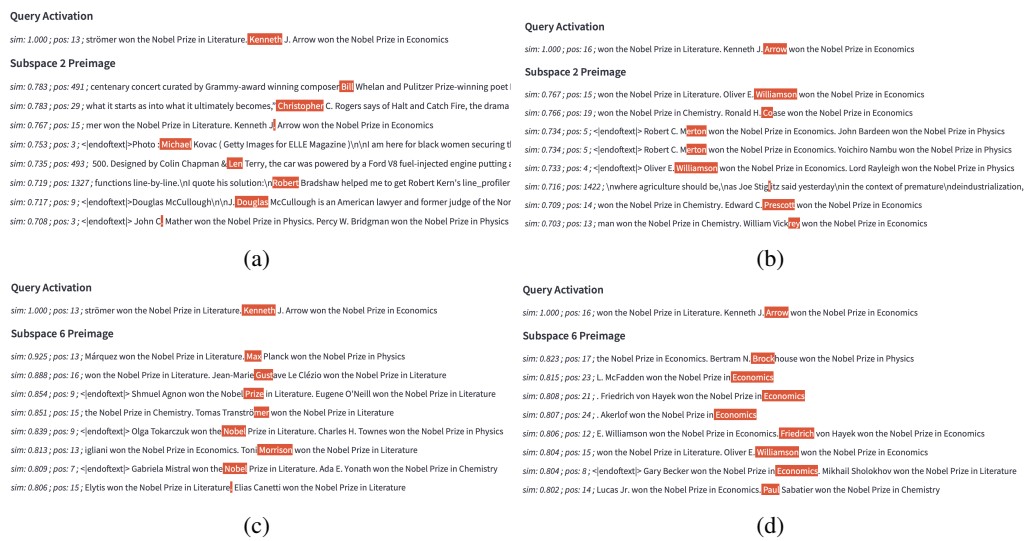

(a)

(b)

(c)

(d)

Figure 32: All statements shown are true, there is no made-up entity. **(a) & (b)**: Subspace 2 at $h^{11,mid}$ in Qwen2.5-1.5B. The subspace is the one with the highest patching effect under `Param Corrupt` in shown in Fig. 5(a). So it should play an important role for routing parametric knowledge. Encoded information shown in (a) is simply "first name" or "partial person name", while information shown in (b) is the knowledge retrieved from model's parametric memory, "the person is economist". Therefore, we hypothesize that *this subspace receives knowledge retrieved from MLP layer*. When only part of the name is observed, the only "knowledge" is that "this is a first name", when enough tokens are observed (not necessarily the full name) to identify a memorized person, the subspace is filled with retrieved knowledge, like the field of work. Note that for this subspace we lower the threshold of cosine similarity to 0.7 in order to include enough samples. So it seems to also encode other kinds of information. Better training might help disentangle them. **(c) & (d)**: Subspace 6 at $h^{11,mid}$ in Qwen2.5-1.5B. This is the one with the second highest effect under `Param Corrupt` in shown in Fig. 5(a). That means it also plays a non-trivial role for parametric knowledge. Encoded information shown in (c) is "Literature in the context", in (d) is "Economics in the context". From these two preiamges, we hypothesize that *this subspace receives knowledge broadcasted from the most recent recognized entity*. At first name token position, the current name "Kenneth" is not enough to identified any memorized person, so this subspace contains the previous entity's knowledge "Literature", other normal tokens such as "Prize", "Nobel", "." also receive this knowledge, and similarly for (d). Note that for this subspace we sometimes do not find clear interpretation for individual preimage. Possible reason could be that sometimes current entity is "half recognized" and its knowledge is mixed with previous entity.

about previous token. In other words, we need to add a feature while removing another one (and it's hard to select which one since there are multiple active features), which requires understanding of feature dependency that SAEs do not provide ("prev-Mary" and "prev-John" are mutually exclusive). More importantly, patching the "prev-John" feature would not have any real effect on inputs where the names are not John. In other words, conceptually, one would expect such patching to only have effect on a few specific inputs, and only to confuse the model. In contrast, patching subspace can be thought of patching meaningful feature *groups*. Moreover, applying such patching to SAEs requires running evaluation on each of the features, which is too computationally expensive due to the large amount of features.

Because of these fundamental difference, we do not compare with SAEs in the main paper. Nevertheless, readers might still wonder how would SAEs perform under the same evaluation used in the paper. For this, a better way is to derive subspaces from SAE features. We take this up in Section F.2.

**Qualitative evaluation**    Our qualitative evaluation has two aspects: (i) interpretability of preimages for subspaces, (ii) consistency of the type of information encoded in a subspace across inputs (thus, interpretability of the overall subspace). Regarding (i), we think that a single SAE feature would have interpretability comparable to a single subspace preimage, as we know that they are interpretable most of the time but also some features do not have clear and easy interpretation. However, regarding (ii), this is not applicable to SAEs, as they provide no clear structure between features; rather, to find encodings of the same type of information requires inspecting each feature individually. In contrast, information in the same subspace found by NDM is of the same "type" across inputs most of the time. Inspecting some of the them can give an idea of the whole feature group, significantly reducing the amount of interpretation work needed.

**(2)** *What about training SAEs within subspaces?*

Conceptually, if one trained a top-1 SAE (top-k SAE with k=1) within each subspace, we expect that the features would be the cluster centers in the subspace. Inspecting a feature means to find inputs whose activation in the subspace is close to the feature. This is roughly the same as inspecting preimages in InversionView, with query activations taking the role of the "feature". Thus, InversionView can conceptually be thought of as a "parameter-free" top-1 SAE.

F.2    QUANTITATIVE EXPERIMENTS ON GPT2 TEST SUITE

In this section, we check whether we can obtain some meaningful subspaces from SAE features and evaluate them with our metrics.

**Settings**    We use the method introduced in Engels et al. (2024) (Sec. 4), where the authors cluster feature vectors and find subspaces. Specifically, the method has three steps:

1. Cluster features by their pairwise cosine similarity. There are two clustering method, (a) `Graph`. It connects each feature with its top-k neighbors whose similarity is above a threshold, and form clusters by recursively searching for connected features. (b) `Spectral`. Spectral clustering.

2. For each cluster, run the SAE on model activations. When reconstructing activations, they only use features inside the cluster (i.e., activation value for other features are always zero), we refer them as cluster-specific reconstructions. They remove zero reconstructions from this collection, so that at least one feature in the cluster is activated.

3. Find the multi-dimensional subspaces spanned by these cluster-specific reconstructions. They apply PCA on these reconstructions and inspect the points in the subspaces spanned by pairs of principle components with large eigenvalues.

In our case, we aim to *extract subspaces* from clusters of features. Thus we slightly change their method for our purpose:

1. Cluster features. Same as above.
2. Sort clusters either by the average cosine similarity between features inside the cluster, or by the cluster size (number of features) in descending order.

For each cluster (in this order), apply the next three steps. Overall, we iteratively build a matrix $C \in \mathbb{R}^{d \times m}$ of orthonormal where $m$ is the number of dimensions selected so far. The columns of $C$ are orthonormal columns, representing a basis for the subspaces constructed so far. for each cluster, we add another set of columns representing the subspace for that cluster.

(a) Obtain cluster-specific reconstructions, using the same method as Step 2 above in the original method. Denote the reconstructions as $X \in \mathbb{R}^{d \times n}$, where $n$ is the number of data points and $d$ is model dimension.

(b) Remove variance on previous selected directions (skip if no direction has been selected so far). Denote the directions selected so far as $C \in \mathbb{R}^{d \times m}$, where $m$ is the number of selected directions. We then take

$$\hat{X} \leftarrow X - CC^T X \tag{13}$$

which removes variance on the directions selected so far, or – equivalently – projects on the orthogonal complement of the subspaces constructed so far.

(c) Apply SVD on $\hat{X}$. We select all left singular vectors (i.e., the same same as the eigenvectors in PCA) whose corresponding singular values are above a threshold. The threshold is set by multiplying the maximum singular value with a hyperparameter `rtol`. The selected singular vectors, or directions, form a **new subspace** corresponding to this feature cluster. They are added as the new columns of $C$.

3. We stop once $C$ becomes a full basis, i.e., $C \in \mathbb{R}^{d \times d}$, or the clusters are exhausted. In the latter case, we supplement $C$ with remaining directions to make it an orthogonal matrix.

In sum, we follow the original method from Engels et al. (2024) as much as we can, while adapting it to construct a full orthogonal decomposition of the space. We test this method on the GPT-2 test suite, with many different hyperparameter configurations. Each configuration is a combination of the following hyperparameters: In step 1, use `Graph` or `Spectral`. For the former, there are two hyperparameters: the `k` used in defining top-k neighbors and the threshold `cutoff`. For the latter, `num` specifies how many cluster there should be. In step 2, sort the feature in two ways, `avg-sim` or `size`. In Step 5, `rtol` that filters out unimportant directions. We search `k` among [2, 3, 4, 5, 6, 8, 16, 32], search `cutoff` among [0.1, 0.11, 0.13, 0.14, 0.16, 0.18, 0.21, 0.23, 0.26, 0.3, 0.34, 0.38, 0.43, 0.48, 0.5, 0.55, 0.62, 0.7, 0.78, 0.89, 1.0], search `num` among [10, 25, 50, 100], search `rtol` among [0.01, 0.03, 0.1]. For `graph`, the resulting number of clusters varies a lot. We remove those configurations whose number of clusters in any layer is out of the range [5, 200]. This results in 16 configurations for `graph`. When extracting subspaces, we ignore clusters which contain $\leq 3$ features (as the target information we are testing is not likely to be encoded in 3D or smaller subspaces), and further restrict the number of subspace to be inside [5, 50] in order to speed up the evaluation. Unexpectedly, none of the configuration using `graph` produced a partition satisfying the constraint. We find that this is mainly because it produces very uneven-sized clusters. There are usually a few big clusters, and many small clusters each of which has a few features. Some configurations that use `spectral` are end up being filtered out.

**Results** The results are shown in Table 8. We can see there are two best-performing configurations whose average Gini coefficient is 0.56; we put the first row in the main paper, Table 1. It shows that extracting subspaces from SAE feature clusters is significantly better than other baseline methods (0.21-0.38 in Table 1), probably producing somewhat meaningful partition. But there is still a significant gap between it and NDM's result (0.71). In addition, we can see that many configurations are somewhat successful in test 1, e.g., the third to last row (Spectral, num=100, avg-sim, rtol=0.03), but not very successful in other tests.

## G  ADDITIONAL DISCUSSION

### G.1  WHY IS INVERSIONVIEW MORE SUITABLE IN SUBSPACES

Compared to whole representation space, we argue that InversionView (Huang et al., 2024b) is more suitable to be applied to subspaces. For example, subspace 2 and subspace 7 in Fig. 3 encode current token and current position respectively. If we look for the preimage in the joint space of subspace

| Cluster config | Extract config | # subspaces | test 1 | | | test 2 | | | test 3 | | | test 4 | | | test 5 | | | Avg |
|---|---|---|---|---|---|---|---|---|---|---|---|---|---|---|---|---|---|---|
| | | | - | $d_s$ | $\mathrm{Var}_s$ | - | $d_s$ | $\mathrm{Var}_s$ | - | $d_s$ | $\mathrm{Var}_s$ | - | $d_s$ | $\mathrm{Var}_s$ | - | $d_s$ | $\mathrm{Var}_s$ | |
| Spectral, num=10 | avg-sim, rtol=0.1 | 7,8,9,7 | 0.60 | 0.44 | 0.60 | 0.61 | 0.41 | 0.67 | 0.73 | 0.39 | 0.54 | 0.74 | 0.31 | 0.54 | 0.71 | 0.56 | 0.61 | **0.56** |
| Spectral, num=10 | size, rtol=0.1 | 5,7,8,8 | 0.70 | 0.61 | 0.35 | 0.40 | 0.32 | 0.56 | 0.70 | 0.30 | 0.49 | 0.64 | 0.22 | 0.44 | 0.68 | 0.50 | 0.46 | 0.49 |
| Spectral, num=25 | avg-sim, rtol=0.03 | 6,7,7,7 | 0.78 | 0.72 | 0.72 | 0.42 | 0.16 | 0.36 | 0.48 | 0.24 | 0.33 | 0.35 | 0.20 | 0.30 | 0.48 | 0.33 | 0.41 | 0.42 |
| Spectral, num=25 | avg-sim, rtol=0.1 | 10,12,12,13 | 0.52 | 0.60 | 0.53 | 0.50 | 0.26 | 0.48 | 0.60 | 0.43 | 0.54 | 0.60 | 0.30 | 0.51 | 0.59 | 0.36 | 0.48 | 0.49 |
| Spectral, num=25 | size, rtol=0.1 | 9,9,8,7 | 0.81 | 0.67 | 0.36 | 0.53 | 0.48 | 0.52 | 0.55 | 0.38 | 0.33 | 0.49 | 0.44 | 0.42 | 0.59 | 0.51 | 0.42 | 0.50 |
| Spectral, num=50 | avg-sim, rtol=0.01 | 5,6,6,6 | 0.65 | 0.67 | 0.71 | 0.37 | 0.15 | 0.42 | 0.13 | 0.11 | 0.35 | 0.18 | 0.14 | 0.35 | 0.27 | 0.15 | 0.32 | 0.33 |
| Spectral, num=50 | avg-sim, rtol=0.03 | 7,8,9,8 | 0.77 | 0.69 | 0.73 | 0.45 | 0.21 | 0.41 | 0.43 | 0.19 | 0.36 | 0.39 | 0.18 | 0.34 | 0.43 | 0.25 | 0.46 | 0.42 |
| Spectral, num=50 | avg-sim, rtol=0.1 | 15,18,21,19 | 0.85 | 0.80 | 0.53 | 0.55 | 0.31 | 0.44 | 0.52 | 0.29 | 0.36 | 0.46 | 0.27 | 0.42 | 0.49 | 0.26 | 0.49 | 0.47 |
| Spectral, num=50 | size, rtol=0.1 | 7,8,8,7 | 0.63 | 0.62 | 0.68 | 0.45 | 0.29 | 0.35 | 0.52 | 0.43 | 0.49 | 0.46 | 0.35 | 0.38 | 0.64 | 0.48 | 0.50 | 0.48 |
| Spectral, num=100 | avg-sim, rtol=0.01 | 9,8,8,9 | 0.70 | 0.56 | 0.72 | 0.28 | 0.17 | 0.36 | 0.21 | 0.09 | 0.27 | 0.19 | 0.11 | 0.26 | 0.31 | 0.14 | 0.30 | 0.31 |
| Spectral, num=100 | avg-sim, rtol=0.03 | 13,12,11,12 | 0.84 | 0.70 | 0.81 | 0.45 | 0.21 | 0.47 | 0.34 | 0.16 | 0.32 | 0.33 | 0.18 | 0.33 | 0.34 | 0.18 | 0.36 | 0.40 |
| Spectral, num=100 | avg-sim, rtol=0.1 | 24,26,26,25 | 0.86 | 0.74 | 0.54 | 0.51 | 0.33 | 0.47 | 0.48 | 0.32 | 0.43 | 0.41 | 0.24 | 0.42 | 0.38 | 0.21 | 0.41 | 0.45 |
| Spectral, num=100 | size, rtol=0.1 | 9,12,12,11 | 0.76 | 0.74 | 0.66 | 0.54 | 0.49 | 0.57 | 0.63 | 0.59 | 0.60 | 0.51 | 0.33 | 0.29 | 0.69 | 0.59 | 0.40 | **0.56** |

Table 8: Results on GPT-2 Small test suite, higher is better. Refer to Table 1 for explanation of the metrics. We include hyperparameters and the number of subspaces produced for each layer used in the tests.

2 and 7, the inputs will need to have the same token at the same position to be inside the preimage. Thus when the query activation contains multiple pieces of information, or encode multiple aspects of the input, the samples in preimage should be the same in all these aspects, which makes it hard to find such samples. Even if we can find enough, they might simply have the same recent tokens, which does not give much insights.

Importantly, some subspaces might be dominating the joint space while some others are ignored. If the norm of the activations in subspace A is much bigger than those in subspace B, the cosine similarity in the joint space of A and B will mainly depend on the cosine similarity in subspace A. In this case, the information read out in preimage of the joint space is only the information encoded in subspace A. In some cases this is fine, as the information in subspace A is much more prominent than the one in subspace B, and InversionView gives the right overall picture. But in some cases, e.g., in the residual stream of transformers, small subspaces might dominate certain attention heads, as attention heads usually operate in small subspaces (we do not think this only applies to attention heads).

## G.2 ON THE RELIABILITY OF SUBSPACE ACTIVATION PATCHING

Our quantitative experiments on language models relies on subspace activation patching. (Makelov et al., 2023) found cases where subspace activation patching produces illusory conclusion. Subspace intervention might change the model's output by activating a dormant pathway which does not have effect in normal computation, as illustrated in Fig. 33. Here we argue that our experiments are still reliable because of following reasons: (1) Our method finds subspace partition that reduces MI or dependency between subspaces, instead of partition that maximize patching effect. The partition shown in Fig. 33 that causes illusion has very high MI between subspaces, thus our method would not give such partition. In other words, we apply subspace patching on subspaces found by NDM, because of relatively good independency, so activations from different subspaces can be composed more or less freely, without causing out-of-distribution activations. (2) (Makelov et al., 2023) recommend doing subspace patching in activation bottlenecks, especially residual stream, to avoid such illusion. This is exactly what we did in experiments (3) We also provide validation by examining the subspaces qualitatively.

## G.3 ABOUT FUTURE DIRECTIONS

So far we have provided various evidence supporting our claim that we can decompose representation space into smaller and meaningful subspaces, without any human supervision. These subspaces, being orthogonal to each other, can serve as new basic units or mediators for mechanistic interpretability. Compared to neurons, they capture the distributed nature of neural representations. Compared to sparse features, they provide more structure. Each subspace can be thought of as a group of infinitely many features sharing a certain higher-level concept, by interpreting some of them we can understand the whole group. Moreover, it enables us to understand the model by directly analyzing the model's weight matrices. For example, we can understand an attention head by checking which subspace its query, key, and value matrices read from and which subspace it writes to (Elhage et al., 2021). By doing so, we build connections between subspaces across layers, resulting in subspace circuits. We believe subspaces could be more "fundamental" than model components as they are designed to be as

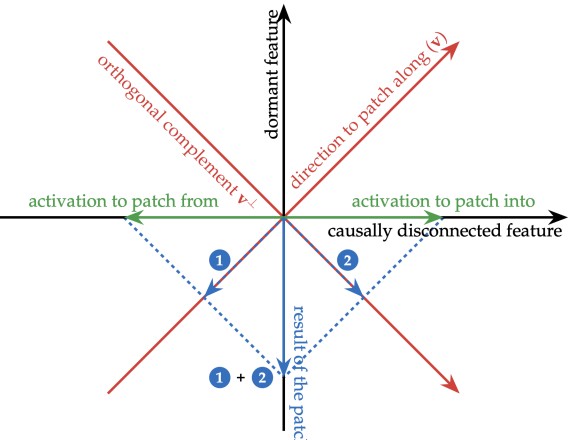

Figure 33: An example illustrating subspace activation patching illusion. Fig. 1 from (Makelov et al., 2023). The 2D space is partitioned into two 1D space (red arrrows). Green arrows represent activations in the whole space. Subspace patching modify their projection along the red $v$ direction, resulting a new activation (blue arrow). In this setting, suppose x-axis has variance but no downstream causal effect, while y-axis has no variance when running model on normal data but has causal effect if there is any nonzero value. In this setting, resulting new activation (blue arrow) is out-of-distribution and will have causal effect, suggesting $v$ is a meaningful direction. However, this 2D space does not play a meaningful role in model's computation (suppose this 2D space is a subspace of a bigger space). In our case this does not raise a severe concern. When applying NDM in this 2D space, partitioning like red arrows in this figure is not likely to happen, because the resulting two 1D subspaces have very high MI. Partitioning like the x and y-axis is the optimal choice, thus resulting two unimportant 1D space.

independent as possible. Moreover, subspace circuits would, like real transformers, operate in the same manner over the whole input sequence (uniform across token positions) as well as over different input sequences. Such circuits could be predecessors of source-code-like explanation of the model, since variables in RASP language (Weiss et al., 2021) can fundamentally be treated as subspaces of residual stream. To support this ambitious plan, the current version of subspace partition method might not be good enough. We believe there is plenty of room for further improvement. For example, allowing more flexible searching over dimension configurations (rather than having all dimensions as multiples of a fixed number). We also think a hierarchical structure of subspaces could also be helpful. For example, subspace A and B are very independent (e.g., current token and position), and A can be further partitioned into A1 and A2, which are less independent (e.g., the plain token and retrieved knowledge about it). More importantly, there could be potentially different ways achieving the same goal (e.g., by minimax approaches, see App. G.4). We also call for efforts from industry as this unsupervised method should benefit from more data and compute.

### G.4    POSSIBLE ALTERNATIVE APPROACHES

In this section, we describe some alternative approaches in order to inspire future research.

**Split-based approach**    The high-level idea is as follows:

1. partition the space into two subspaces. We search over n possible configurations (for example, when d=100, we can have $[d_1 = 25, d_2 = 75]$, $[d_1 = 50, d_2 = 50]$, $[d_1 = 75, d_2 = 25]$), each configuration has a dedicated orthogonal matrix. It turns out that initialization matters a lot and standard basis is kind of special (Fig. 15), though they are symmetric, empirically we found $[d_1 = 25, d_2 = 75]$ and $[d_1 = 75, d_2 = 25]$ produces quite different results. After certain amount of steps, measure (normalized) MI for each configuration, select the one with the lowest MI.

| index $s$ | 0 | 1 | 2 | 3 | 4 | 5 | 6 | 7 | 8 | 9 | 10 | 11 |
|---|---|---|---|---|---|---|---|---|---|---|---|---|
| $d_s$ | 36 | 54 | 54 | 36 | 12 | 144 | 108 | 36 | 18 | 54 | 54 | 162 |
| $\text{Var}_s$ $(\times 10^3)$ | 0.4 | 1.0 | 0.5 | 0.4 | 0.8 | 2.7 | 8.3 | 0.6 | 0.4 | 0.6 | 0.6 | 5.8 |
| $\Delta P_s$ $(\times 10^{-2})$ | 0.3 | 1.3 | 1.1 | 0.3 | 0.3 | 2.8 | 2.3 | 0.6 | 0.4 | 1.9 | 0.6 | 40.7 |
| $\frac{\Delta P_s}{d_s}$ $(\times 10^{-4})$ | 0.9 | 2.5 | 2.0 | 0.8 | 2.1 | 1.9 | 2.2 | 1.6 | 2.3 | 3.5 | 1.1 | 25.1 |
| $\frac{\Delta P_s}{\text{Var}_s}$ $(\times 10^{-5})$ | 0.9 | 1.3 | 2.2 | 0.7 | 0.3 | 1.1 | 0.3 | 0.9 | 1.1 | 3.2 | 1.0 | 7.0 |

Table 9: Test 5 raw results for split-based NDM. Gini coefficient for row $\Delta P_s$ is 0.78, for row $\frac{\Delta P_s}{d_s}$ is 0.55, and for row $\frac{\Delta P_s}{\text{Var}_s}$ is 0.47.

2. Repeat step 1 for each subspaces already obtained, while restricting the rotation inside the already obtained subspace. For example, assume we have learned a matrix $\mathbf{R}^{\text{prev}}$ and configuration $[d_1^{\text{prev}} = 75, d_2^{\text{prev}} = 25]$, we keep (previous) subspace 1 and 2 fixed. For subspace 1, we search over configurations $[d_1 = 18, d_2 = 57]$, $[d_1 = 36, d_2 = 39]$, $[d_1 = 54, d_2 = 21]$, each associated with a separate $\mathbf{R}_1, \mathbf{R}_2, \mathbf{R}_3 \in \mathbb{R}^{75 \times 75}$. Similarly do this for subspace 2. Assume in this iteration, after training $[d_1 = 36, d_2 = 39]$ has lowest MI for subspace 1, we then merge its orthogonal matrix $\mathbf{R}_2$ into $\mathbf{R}^{\text{prev}}$ by multiplying it with corresponding rows in $\mathbf{R}^{\text{prev}}$: $\mathbf{R}_{[0:75]}^{\text{prev}} \leftarrow \mathbf{R}_2 \mathbf{R}_{[0:75]}^{\text{prev}}$. On the other hand, suppose none of the configuration for subspace 2 is lower than threshold, then we do not update that subspace. After this iteration, we have a new $\mathbf{R}^{\text{prev}}$, and new configuration $[d_1^{\text{prev}} = 36, d_2^{\text{prev}} = 39, d_3^{\text{prev}} = 25]$. Then keep repeating this loop until no subspace is separable (all configurations of all subspaces have MI greater than the threshold), or a pre-defined maximum iteration is reached.

This process increases the number of subspaces exponentially, if all subspace partition attempts are successful (MI below threshold). However, a disadvantage compared to merging-based approach in the main paper is increased compute cost, because we search over n configurations, training n $\mathbf{R}$s and select only one. An advantage is that it gives a tree structure, we can understand subspaces of different granularity. Our code link also contains code for this variant.

We did some preliminary experiments to test this approach, the results are not better than merging-based approach, while it takes longer to train. For example, Table 9 shows a partition learned for layer 9 and tested on Test 5, it runs 4 outer iteration (each iteration splits subspaces), and each time 3 configuration are tested for each subspace. Search number is $5 \times 2^{14}$, totally 40k steps.

However, our negative results do no rule out its usefulness, it could be more computationally efficient if there's more flexible and efficient way to determine subspace dimensions, and might be suitable in some cases.

**Minimax approach** Mutual information can also be estimated by neural networks (Belghazi et al., 2018). The Mutual Information Neural Estimator (MINE) is as follows:

Let $\mathcal{F} = \{T_\theta\}_{\theta \in \Theta}$ be the set of functions parametrized by a neural network. MINE is defined as (we simplify the notation to avoid some details, please refer to (Belghazi et al., 2018) for the true definition):

$$I(X; Z) = \sup_{\theta \in \Theta} \mathbb{E}_{\mathbb{P}_{XZ}}[T_\theta] - \log\left(\mathbb{E}_{\mathbb{P}_X \otimes \mathbb{P}_Z}[e^{T_\theta}]\right). \tag{14}$$

Simply speaking, $\mathbb{P}_{XZ}$ means sampling vectors $\boldsymbol{x}$ and $\boldsymbol{z}$ from joint distribution, and $\mathbb{P}_X \otimes \mathbb{P}_Z$ means sampling them from their respective marginal distribution. The neural network $T_\theta$ takes both $\boldsymbol{x}$ and $\boldsymbol{z}$ as input and output a scalar, it is optimized with gradient ascent to approach the supremum.

Like split-based approach, we train an orthogonal matrix $\mathbb{R}$ to partition the space into two subspaces, given a fixed dimension configuration $d_1, d_2$. Assume X and Z represent random variables in the two subspaces. In order to find independent subspaces, we can optimize $\mathbf{R}$ to lower MI as follows:

$$\min_{\mathbf{R}} I(X; Z) = \min_{\mathbf{R}} \max_{\theta \in \Theta} \mathbb{E}_{\mathbb{P}_{XZ}}[T_\theta] - \log\left(\mathbb{E}_{\mathbb{P}_X \otimes \mathbb{P}_Z}[e^{T_\theta}]\right). \tag{15}$$

Empirically, $\mathbb{P}_{XZ}$ means that subspace activations are projected from the same activations, i.e., each pair $x$ and $z$ results from rotating and splitting a single activation vector into two parts. On the other hand, $\mathbb{P}_X \otimes \mathbb{P}_Z$ means each pair results from projecting two independently sampled activation vectors.

The intuition is that, if X and Z have high mutual information, knowing one can predict the other, the neural network can easily discriminate whether $x$ and $z$ are projected from the same activation, thus output high value when they "match" and low value when they do not, resulting high estimate of $I(X;Z)$. $\mathbf{R}$ rotates and reflects the space such that two subspace are as independent as possible, such that the neural network cannot easily discriminate, resulting low MI estimate. This is similar to Generative Adversarial Networks (Goodfellow et al., 2020).

However, empirically we do not find this approach is effective. We estimate MI with KSG estimator (so that comparable to our previous approaches) to evaluate performance during training, and found this does not produce significant decrease in MI.

In addition, based on the same intuition, we try another simpler approach: train two neural networks to reconstruct one subspace's activations based on the other's, and train $\mathbf{R}$ to make it hard to reconstruct.

$$\max_{\mathbf{R}} \min_{T_1, T_2} \mathbb{E}_{\mathbb{P}_{XZ}}[\mathrm{MSE}(x, T_1(z)) + \mathrm{MSE}(T_2(x), z)]. \tag{16}$$

where MSE denotes mean squared error, and $T_1, T_2$ are separate neural networks of different input-output dimension. We find this approach is more effective than the one based on MINE, but resulting final MI is still much higher than split-based NDM (and harder to train). Again, our negative results do not rule out its potential, the results might be because of the difficulties in optimization rather than training objective.

## H  COMPUTE USAGE

We run all experiments on H100s. We report the sources used when using $25 \times 2^{14}$ as search number (largest option in our experiments). Train a subspace partition for one activation site in GPT-2 Small takes around 2 hours on a single H100 (stops at around 80k steps). Train a partition for one activation site in Qwen2.5-1.5B takes around 8 hours on a single H100 (stops at around 80k steps). For Gemma-2-2B, it takes around 8 hours on a single H100 (stops at around 50k steps). Experiments for subspace patching run quickly.

## I  LLMS USAGE

We used LLMs to polish the writing and rephrase some statements to improve the readability, while ensuring that the content continues to convey our intended meaning accurately. We also used them to identify related work and to generate code for producing figures.

