# OpenReview forum: "Decomposing Representation Space into Interpretable Subspaces with Unsupervised Learning"
_ICLR.cc/2026/Conference — ICLR 2026 Poster_

### Official Review · Reviewer_bKFF · 2025-10-25

**Soundness:** 3
**Presentation:** 3
**Contribution:** 2
**Rating:** 4
**Confidence:** 4

**Summary:**

This paper introduces Neighbor Distance Minimization (NDM) as a method for performing unsupervised identification of interpretable subspaces within the latent spaces of neural models. NDM operates under the hypothesis that groups of mutually exclusive features form subspaces, and the features within these subspaces exist within superposition. Through experiments on GPT-2, the authors demonstrate that these subspaces behave similarly to the “variables” used by the model. This approach provides a new avenue for interpretability research within and between subspaces.

**Strengths:**

1. Motivation and Construction: The motivation of NDM and its subsequent demonstration on the toy models of superposition setting provides a clear intuition as to why we’d expect it to work on larger models and how we can interpret its results.
2. Novel Perspective: The developed perspective offers an alternative avenue for future research in the field of interpretability, specifically by considering the interaction between or within subspaces.

**Weaknesses:**

1. Lack of Details on the Computational Aspect of NDM: There is a lack of information regarding the computational burden imposed by using NDM. Indeed, 2B parameter models are not large-scale by current standards. This raises questions about the practicality of these methods in real-world settings.
2. Single Domain Evaluation: NDM is only evaluated on text models. It would also be important to evaluate the effectiveness of exploring image models.
3. Comparison to other Interpretability techniques: Although the intuitive differences between NDM and other interpretability methods, such as sparse autoencoders, are provided, no practical comparison is made.
4. Limited Granularity: By construction, NDM can only identify as many subspaces as there are dimensions in the latent space. In fact, it is much less than that, as the subspaces often have relatively large dimensions. Therefore, it is unclear how granular these “variables” corresponding to subspaces are, and it is likely that additional interpretability techniques would have to be applied to the individual subspaces.

**Questions:**

1. What are the computational requirements for applying NDM?
2. How sensitive is NDM to the provided N model activations?
3. Have you explored an iterative application of NDM? Namely, re-applying NDM to the identified subspaces?

---

> ### Author Response · Authors · 2025-11-17
> **(1/1)**
>
> Thanks for the close reading and the feedback!
>
> ## Reply to Weaknesses
>
> >    Lack of Details on the Computational Aspect of NDM: There is a lack of information regarding the computational burden imposed by using NDM. Indeed, 2B parameter models are not large-scale by current standards. This raises questions about the practicality of these methods in real-world settings.
>
> In the second to last Section in Appendix (Appendix G in old ver. or Appendix H in new ver.), we report the compute usage of our experiments. Even though we search among a huge number of activations for the neighbor, there is only a single trained matrix. For example, it takes 2 hours on one H100 to train a partition for a layer in GPT-2, and 8 hours for Gemma-2-2B. While it is likely possible to get better results with more GPU hours, we think the current results are interesting enough. Thus, the compute usage is roughly at the same level as sparse autoencoders (SAEs).
>
> We hadn't referred to that section in the main paper, sorry about that, we have updated the paper accordingly. Thanks for pointing it out.
>
> >    Single Domain Evaluation: NDM is only evaluated on text models. It would also be important to evaluate the effectiveness of exploring image models.
>
> Indeed, we only evaluated on the text domain. Given the substantial effort required to develop this method and 4 different evaluations, we haven't tested it on image models, so that remains for future work. Still, we are optimistic about the applicability on those models, as the method itself does not rely on domain-specific assumptions.
>
> >    Comparison to other Interpretability techniques: Although the intuitive differences between NDM and other interpretability methods, such as sparse autoencoders, are provided, no practical comparison is made.
>
> Regarding SAEs, kindly refer to the global comment for discussion and new experiments, where we report an SAE-based comparison for our quantitative evaluation.
>
>
> >    Limited Granularity: By construction, NDM can only identify as many subspaces as there are dimensions in the latent space. In fact, it is much less than that, as the subspaces often have relatively large dimensions. Therefore, it is unclear how granular these “variables” corresponding to subspaces are, and it is likely that additional interpretability techniques would have to be applied to the individual subspaces.
>
> We agree that the subspaces we currently find are usually big ones, and it will be interesting to find more fine-grained ones. But we also would like to emphasize that some "variables" require big subspaces by their nature, because they can take a larger number of "values". In the evaluations, many types of target information, such as the current token or the induction mechanism output, can take a large number of values, and we thus expect them to be encoded in large subspaces.
>
> ## Reply to Questions:
>
> >    What are the computational requirements for applying NDM?
>
>
> Please see our reply to the first weakness.
>
>
> >    How sensitive is NDM to the provided N model activations?
>
>
> In general, we find that large N is usually better; we show this in Figure 15 in the Appendix. We can also see that performance does not degrade too much even when we reduce N to just 1/25 of its maximum value used in experiments.
>
>
> >    Have you explored an iterative application of NDM? Namely, re-applying NDM to the identified subspaces?
>
> That's an interesting idea. We haven't tried it. It might be a way to get more fine-grained partitions. Thanks for pointing it out.

---

> > ### Comment · Reviewer_bKFF · 2025-11-25
> >
> > I thank the authors for their response, which included a comparison to SAEs and addressed some of my questions. However, I still have concerns regarding its practical applicability. Training a single subspace partition appears to be rather expensive, and so fitting the various hyperparameters seems costly but necessary, as there is a large variability in subsequent performance (e.g., in Table 5, where even at the maximum search number, Gini coefficients vary from 0.57 to 0.71). Similarly, although numerous different experiments have been conducted, it does not appear that many have been repeated over different random initialisations to obtain error bars for the results. Therefore, I am inclined to maintain my score of leaning towards rejection, as I am unsure how robust the conclusions from NDM are.

---

> ### Author Response · Authors · 2025-11-27
>
> Thanks for your response and checking the paper again. We agree that training could be expensive, and some hyperparameter searching is needed. However, the widely used SAE approach faces similar challenges (even worse in terms of the number of parameters to be trained), which we think may be inherent to unsupervised methods.
>
> We also would like to emphasize that Gini coefficients varying from 0.57 to 0.71 does not necessarily mean that the quality or interpretability of the subspaces varies a lot. The partition whose coefficient is 0.57 can be as meaningful as the one with 0.71. The reason is that the hyperparameters influence the granularity of information, and some information could be separated into different subspaces or not, depending on the desired granularity of the decomposition (e.g., input token and the knowledge about the token). We expect that the Gini score will be highest when the granularity matches the one required by the evaluation. Hence, variability in the Gini score can just reflect mismatch in granularity of information partition to the evaluation, not that some decompositions are inherently better than others. Under a specific evaluation metric, one would have higher score than the other, though they both make sense. Hence, we consider different levels of granularity by changing the hyperparameters, and report the best ones for both NDM and SAE baselines. This can be viewed as the score obtained when the decomposition granularity is best matched to the evaluation suite.

---

### Official Review · Reviewer_MPxT · 2025-10-27

**Soundness:** 2
**Presentation:** 3
**Contribution:** 3
**Rating:** 4
**Confidence:** 3

**Summary:**

The paper begins from the premise that mutual exclusiveness could be a fundamental condition for superposition. Building on this idea, it proposes that data may contain groups of mutually exclusive features; for example, features encoding categorical variables such as different subjects, where only one is active at a time (L. 110). Motivated by this observation, the paper introduces an unsupervised method, termed NDM, designed to identify subspaces of mutually exclusive features. The method does so by finding subspaces in which data points are projected to similar locations. The paper demonstrates the effectiveness of NDM both on a synthetic (toy) model and on representations from large language models.

**Strengths:**

- I really appreciate the core idea and intuition behind the argument that mutual exclusiveness may be a fundamental condition for superposition, as well as the proposed method built on this insight.
- The results on the toy model effectively illustrate and complement the central narrative.
- The paper does a good job explaining the intuition behind the approach and is, for the most part, well written and thorough in its exposition.
- Overall, I found this to be one of the more enjoyable papers to read, with a clear take-home message.
- While the evaluation still leaves considerable room for improvement, the paper does at least include some empirical assessment of the proposed approach.

**Weaknesses:**

- I did not find the qualitative examples in Figure 2 particularly convincing. The positions shown in panel (c) still exhibit a fairly large range, and for the other examples, it seems plausible that sparse autoencoders could identify similar concepts. Given the narrative developed in the paper, I would have expected the qualitative examples to focus more on illustrating the idea of distinct variables, rather than on specific concepts that might also be captured by SAEs.
- In Table 1, it is unclear why no comparison to sparse autoencoders (SAEs) is included. A theoretical justification would be sufficient if there is a solid reason for omitting such a comparison. Additionally, it would be helpful to include an analysis of how the number of subspaces affects the results. For instance, if only a single subspace were used, the condition that the high-level "variable" lies within the same subspace (L. 261) would trivially hold.
- The method relies on the mutual information (MI) threshold, but it is not clear how this threshold should be selected in practice. Although the paper includes ablation studies on this parameter, the specific effect of the threshold remains unclear, as the quantitative experiments are relatively small in scale.

**Questions:**

- In Eq. (1), why is $h = W x x'$ instead of $h = W x$, as used in the experimental setup of Elhage et al. (2022)? It is possible that the equation is incorrectly formatted and that $x'$ is intended to appear on the right-hand side?
- How to choose the MI threshold and how sensitive is the method to that threshold?
- How would sparse autoencoders (SAEs) perform under the proposed evaluation setup, both in quantitative metrics and in qualitative analyses?
- According to the manuscript, superposition occurs when mutual exclusiveness holds, and the proposed method aims to identify subspaces of mutually exclusive features. However, it is not clear why we should expect these subspaces to be inherently interpretable if superposition still occurs within them. Wouldn’t we, in many cases, need an additional method to disentangle or remove the remaining superposition? Furthermore, how can we identify situations in which such additional disentanglement would be necessary?

Due to the open questions regarding the evaluation and the distinction from sparse autoencoders (SAEs), I would currently lean toward rejecting the paper. However, I believe the work has potential, and if my concerns are addressed convincingly and no major issues are raised by other reviewers, I would be happy to reconsider and improve my score.

---

> ### Author Response · Authors · 2025-11-17
> **(1/2)**
>
> Thanks for your deep engagement and the close reading!
>
> ## Reply to Weaknesses
> > I did not find the qualitative examples in Figure 2 particularly convincing. The positions shown in panel (c) still exhibit a fairly large range, and for the other examples, it seems plausible that sparse autoencoders could identify similar concepts. Given the narrative developed in the paper, I would have expected the qualitative examples to focus more on illustrating the idea of distinct variables, rather than on specific concepts that might also be captured by SAEs.
>
> Thanks for this feedback. We'd like to make two points:
>
> First, regarding panel (c) in Figure 2, in our web application, you can select "7-32" as the subspace on the left toolbar (and keep "x4.post"), and increase the threshold to 0.98 or even higher. We also added a new Figure 30 in the paper showing the case when threshold is 0.99. You can see that the positions are in a more concentrated range (remaining error could be because the optimization is not optimal, or the model has forgotten the information at this layer). The larger range in panel (c) is because we use a threshold of 0.85 for all subspaces, including the position subspace. Thus, it seems that position information is encoded more continuously, without a strong discrete clustering pattern like some other subspaces.
>
> Second, for other examples in Figure 2, yes, we agree SAEs are likely to identify similar concepts. Our key point here is that related concepts are in the same subspace. So actually Figure 20-23 would be more important qualitative examples to illustrate this idea. Each of them shows the **same subspace** on different inputs,  and the concept in a subspace is *consistent* while different, like a "variable" taking different "values". On the other hand, Figure 2 takes one panel from each of Figure 20-23, showing **different subspaces**. It aims to show what kind of "variables" we can have. Due to space limit, we can only choose one Figure to put in the main paper. Do you think it would be better if we choose one figure from Figure 20-23 instead of extracting a panel from each of them? Would love to hear your opinion! We also have more figures like 20-23 showing various examples in the Appendix E.3, E.4.
>
>
> > In Table 1, it is unclear why no comparison to sparse autoencoders (SAEs) is included. A theoretical justification would be sufficient if there is a solid reason for omitting such a comparison. Additionally, it would be helpful to include an analysis of how the number of subspaces affects the results. For instance, if only a single subspace were used, the condition that the high-level "variable" lies within the same subspace (L. 261) would trivially hold.
>
> Regarding SAEs, kindly refer to the global comment for discussion and new experiments.
>
> Regarding the effect of the number of subspaces, thanks for pointing this out -- we updated the paper to include the number in Table 5, along with the the final results. Indeed, there could be some trivial cases making the high-level "variable" lie within the same subspace. For example, there could be a big subspace that takes most of the dimension or most of the variance, together with many tiny subspaces. That's why we also include columns $d_s$ and $Var_s$, where the effects are normalized by subspace dimension and variance. These trivial cases would show low Gini coefficient in these columns. Intuitively, these columns measure whether there exists a subpace containing the target information (the high-level "variable") without including too much other information.
>
> > The method relies on the mutual information (MI) threshold, but it is not clear how this threshold should be selected in practice. Although the paper includes ablation studies on this parameter, the specific effect of the threshold remains unclear, as the quantitative experiments are relatively small in scale.
>
> Indeed, we select the MI threshold based on empirical results. Conceptually, some information can be encoded in one or multiple subspaces. For example, an entity name and the knowledge about the entity might be in one or two subspace; either seems like a reasonable partition. The MI between them can be strong (when one completely determine the other) or relatively weak (e.g., knowing the profession cannot determine the person).  The MI threshold determines the granularity of subspaces, and human users can set the threshold based on the granularity they want, e.g., whether they want to see these two kind of information separated or together.

---

> ### Author Response · Authors · 2025-11-17
> **(2/2)**
>
> ## Reply to Questions
>
> > In Eq. (1), why is h=Wxx' instead of h=Wx, as used in the experimental setup of Elhage et al. (2022)? It is possible that the equation is incorrectly formatted and that  is intended to appear on the right-hand side?
>
> Thanks a lot for pointing out. Yes, this is a misformating. The correct version is
> $$
> h=Wx
> $$
> and then
> $$
> x^\prime = ReLU(W^Th+b)
> $$
> where $x$ is the ground truth vector and $x^\prime$ is trained to reconstruct it.
> The symbols were misaligned after we put them in one line to meet the page limit. We are very sorry for the confusion. We have corrected it and uploaded the new version.
>
> > How to choose the MI threshold and how sensitive is the method to that threshold?
>
> In our experiments, the estimated MI between subspaces varies a lot. For example, for GPT-2 layer 4, MI between equally-sized subspaces (before any merging) vary from 0.003 to 0.138. So there are some subspaces pairs that almost always get merged, and some are always separated. Nevertheless, the MI threshold indeed plays a role as there are also borderline cases, and the threshold determiens granularity of the decomposition in those cases.  We believe that the granularity needs to be chosen depending on the desired use case and evaluation; hence, we empirically  test a few and select the best. Importantly, Table 5 in the Appendix shows results with different hyperparameters. We can see that, even though the results vary between thresholds, all of them substantially surpass the baselines.
>
> > How would sparse autoencoders (SAEs) perform under the proposed evaluation setup, both in quantitative metrics and in qualitative analyses?
>
> Regarding SAEs, kindly refer to the global comment for discussion and new experiments.
>
>
> > According to the manuscript, superposition occurs when mutual exclusiveness holds, and the proposed method aims to identify subspaces of mutually exclusive features. However, it is not clear why we should expect these subspaces to be inherently interpretable if superposition still occurs within them. Wouldn’t we, in many cases, need an additional method to disentangle or remove the remaining superposition? Furthermore, how can we identify situations in which such additional disentanglement would be necessary?
>
> Yes, superposition is expected to occur within subspaces. Nonetheless, if there is superposition, InversionView can still effectively decode the information. Even if there is superposition inside a subspace, there can be many non-orthogonal but still distinct clusters, from which InversionView can effectively decode. As long as two clusters are separated at some angle, they will be distinguished by InversionView even if they are far from orthogonal.
>
> We also remark that SAEs are also commonly thought to be a solution to the superposition problem. In principle, training SAEs within subspaces could be another approach, which we expect might lead to similar results as appling InversionView. We also briefly comment in  "*What about training SAEs within subspaces?*" in our global comment.
>
>
> Nevertheless, we agree that in some cases we might need other additional techniques for interpretation, since some features/preimages may be hard to interpret by only looking at input strings.

---

> > ### Comment · Reviewer_MPxT · 2025-11-27
> >
> > I thank the authors for their solid rebuttal. Most of my concerns have been addressed to a satisfactory degree, prompting me to raise my score from 4 to 6. However, in line with Reviewer bKFF, I still have some remaining reservations, for example, the seemingly strong reliance on hyperparameters, which prevented me from increasing my score further. That said, I believe this is a nice paper with contributions that could be of interest to the ICLR community.

---

> > > ### Author Response · Authors · 2025-11-27
> > >
> > > Thank you for raising the score, we appreciate your support. For concerns about hyperparameters, kindly refer to our newest reply to reviewer bKFF.

---

### Official Review · Reviewer_EMRg · 2025-11-01

**Soundness:** 2
**Presentation:** 4
**Contribution:** 3
**Rating:** 6
**Confidence:** 5

**Summary:**

A core objective in interpretability research is to understand content and geometry of activations. The authors propose a novel, original idea to learn a set of linear subspaces, where each subspace holds mutually exclusive features encoded via superposition. Different subspaces are orthogonal thus independent. The authors propose a method to learn these subspaces unsupervised from activation data alone, by learning a rotation matrix (similar to DAS) such that data points within each subspace have minimal distance. They provide intuitive understanding, show their approach in toy models and prove its applicability in small and medium language models. Specifically, they show for some known tasks that all features of interest lie in the same subspace, and they show that subspaces are interpretable (ie monosemantic).

The paper is well-written, proposes an original idea, and elegantly combines mathematical intuition, toy models, and translation to LLMs. My main concern is that some assumptions/hypotheses were not empirically validated. For example, it wasn't shown that features within a subspace are mutually exclusive, or that the orthogonality requirement is faithful of real activation space geometry. They critique SAEs but don't use them as baselines.

**Strengths:**

- great presentation, paper is well-written and I found it easy to follow. The paper provides both intuitive understanding and mathematical precision.
- the paper elegantly combines mathematical intuition, validation in toy models, and translation to LLMs
- the paper posits an interesting, novel idea for an important problem rather than an incremental improvement
- I do find the author's work interesting from a slightly different perspective that wasn't as highlighted as it could: Feature independence. When steering with SAEs, editing individual features results in an OOD reconstruction because many SAE features co-occur which is a real problem.

**Weaknesses:**

Major:
1. "Mutual exclusiveness" doesn't seem like a more fundamental condition, or much different from sparsity at all. Elhage 2022 say that when features are sparse, they can be encoded via superposition. Sparse features are already "almost mutually exclusive" but the "almost" seems important as strong guarantees are hard to make for neural networks.
The authors write that superposition wouldn't work if 2 or more features are active but this is false. Yes, interference and error would grow, but only by a tiny amount in practice. Best proof are SAEs that work well with e.g. 20 features active.
In fact, feature co-occurrence is baked into superposition theory: A prediction of their theory and toy models is that features that are mutually exclusive should make heavy use of superposition, while features that often co-occur, should be better separated, ie encoded as orthogonal directions (at least more orthogonal than mutually exclusive ones). So no new theory is needed to predict geometry of feature groups so I don't understand how "mutual exclusiveness and feature groups" are fundamentally different from "sparsity and superposition". It's good to have methods to decompose this, but it follows from superposition theory, and not new theory/hypothesis has to be invented.
2. The authors assume that features within a subspace are mutually exclusive but you never test this (in fact, you don't even extract features within a subspace).
3. The authors for some tasks that most task-relevant features land in a single subspace but they don't show that this subspace only contains those features, aka is interpretable and monosemantic. It could very well be the case that completely unrelated information is stored within the same subspace as many unrelated features are mutually exclusive.
4. The number of subspaces is strictly bound by the dimensionality of activation space as they must be exactly orthogonal. This might limit the method's potential and interpretability. As we see with SAEs, representation space can be well-approximated with tens of Millions of features and having only few hundred feature groups available at most might be quite limiting, and many different features might be packed into the same subspace (because different features are often mutually exclusive as well). This would limit interpretability. My concern is that in reality, the independence-between-feature-groups assumption doesn't hold 100% and we would be better off with relaxing this restriction a little to allow more feature groups to exist and be more interpretable. A possible approach here could be to use MOLTs (Lindsey, Anthropic 2025).
5. The authors heavily criticize superposition and SAEs but they never directly compare against SAEs although direct comparisons should be possible. They state that other baseline comparisons aren't possible because their subspaces are learned unsupservised from activations, but SAEs are as well. In fact, I think that their method doesn't disagree with superposition and sparsity at all.


Other things that would improve the paper:
- Insight into LLM computation. This paper mainly proposes a new method and validates it but there's no new mechanistic insight about LLM computation. It would improve the paper a lot if the authors could prove by example that this method can discover things that other methods like SAE, DAS, etc can't
- More experiments with LLMs. Qualitative examples in Figure 2 lack rigor or quantification. IOI/greater than results don't measure logit difference recovered when only patching an individual subspace, more evidence that subspaces are monosemantic and interpretable, etc.

**Questions:**

1. Many things are mutually exclusive. For example, if the text is about software licenses, it's not about a fiction novel or a mental health consultation. Features from all these different contexts could be squished into a single subspace. Especially since the number of subspaces is extremely limited. Do you observe this in practice? Doesn't this imply that mutual exclusiveness != interpretability? Doesn't this hurt interpretability of those subspaces and the applicability of this method a lot?

---

> ### Author Response · Authors · 2025-11-17
> **(1/3)**
>
> Thanks for your deep engagement and careful consideration of the points raised in our paper.
>
> ## Reply to Weaknesses
> >    "Mutual exclusiveness" doesn't seem like a more fundamental condition, or much different from sparsity at all. Elhage 2022 say that when features are sparse, they can be encoded via superposition. Sparse features are already "almost mutually exclusive" but the "almost" seems important as strong guarantees are hard to make for neural networks. The authors write that superposition wouldn't work if 2 or more features are active but this is false. Yes, interference and error would grow, but only by a tiny amount in practice. Best proof are SAEs that work well with e.g. 20 features active. In fact, feature co-occurrence is baked into superposition theory: A prediction of their theory and toy models is that features that are mutually exclusive should make heavy use of superposition, while features that often co-occur, should be better separated, ie encoded as orthogonal directions (at least more orthogonal than mutually exclusive ones). So no new theory is needed to predict geometry of feature groups so I don't understand how "mutual exclusiveness and feature groups" are fundamentally different from "sparsity and superposition". It's good to have methods to decompose this, but it follows from superposition theory, and not new theory/hypothesis has to be invented.
>
> Thanks for thinking about it closely. We would like to emphasize that we do not say superposition wouldn't work if 2 or more features are active. We certainly do not mean SAEs would not work if more than 1 feature is active. We mean that, *given 2 or more specific features*, if they often co-occur, superposing *these features* would not work well -- incentivizing the model to represent them more orthogonally. On the other hand, there are some concepts that are mutually exclusive in the real world, and the model can make use of this fact and superpose their features. We then call this a *feature group*. An activation can of course activate many features by picking one feature from each group. Your reasoning in the second half of this point agrees well with the paper. We don't think we have contradictary opinions.
>
> Moreover, we do not say that "mutual exclusiveness and feature groups" is a new theory over "sparsity and superposition". We do not intend to invent new theory, but rather make use of current theory and emphasize implications that are not emphasized by previous work. The difference is that the concept "sparsity" treats features uniformly, and does not straightforwardly indicate any different degree of superposition between features. We would like to emphasize a non-uniform view under which some sets of features can be more orthogonal/superposed than others because of non-uniform dependency between features.
>
> Sorry for the confusion, we have added these clarification to the main paper.
>
>
> >    The authors assume that features within a subspace are mutually exclusive but you never test this (in fact, you don't even extract features within a subspace).
>
> Regarding mutual exclusiveness: In the qualitative examples we show in the paper, the features are indeed mutual exclusive (Figure 20-23). For example, in Figure 20, "current token is 'under'" is mutually exclusive with "current token is 'a'", they cannot happen simultaneously.
>
> Regarding extracting features within a subspace: We comment on this in the section "*What about training SAEs within subsapces?*" in our global reply, where we explain that InversionView allows us to observe features.

---

> ### Author Response · Authors · 2025-11-17
> **(2/3)**
>
> >    The authors for some tasks that most task-relevant features land in a single subspace but they don't show that this subspace only contains those features, aka is interpretable and monosemantic. It could very well be the case that completely unrelated information is stored within the same subspace as many unrelated features are mutually exclusive.
>
> We agree with the concern, we should have pointed the reader in the main paper to the qualitative example of the subspaces with highest effect. We have updated accordingly. In the original version, we have mentioned them in the Appendix, e.g., the second to last sentence in caption of Figure 21, the first sentences in caption of Figure 28 and 29.
>
>
> For example, the decomposition given by NDM in test 1 have high values in all 3 columns (0.89, 0.89, 0.9 for "-": unnormalized, "$d_s$": normalized by dimension, "$Var_s$": normalized by variance respectively). Figure 20 corresponds to subspace 6 in layer 4, where patching effect in Test 1 peaks. We can see strong monosemanticity. As checking only 4 examples can't really confirm it, we also encourage you to check more examples in our web application. The decomposition in Tests 2 and 5 also show relatively high values. Figure 28 corresponds to subspace 4 in layer 6, which is where patching effect in Test 2 peaks. Figure 29 correspond to subspace 4 in layer 9, where patching effect Test 5 peaks. These two subspaces encode the output of induction mechanism, so they show a somewhat mixed meaning, when induction is possible (the current token occurs in previous context), they encode the next token after the current token's previous occurrence. When it is not possible, they often encode recent context. We also emphasized this point in Appendix E.3. For Test 3, patching effect peaks in subspace 9 in layer 8 peaks; here, we found that though its meaning is clear in IOI input domain, the general role is not clear (we discussed this in Line 1938 of the old version). For Test 4, the values are low, around 0.5 in two columns. We do not find its meaning very clear in general.
>
> Moreover, our metric also takes monosemanticity into account, as we measure inequality between *normalized* effects (columns $d_s$ and $Var_s$), if a single subspace also contains much other information, it will unavoidably occupy more dimensions and variance of the whole space, and this will be punished by normalization term.
>
> Furthermore, we'd like to argue that, if features are mutually exclusive, they are not truly unrelated, as mutual exclusiveness itself indicates there is some logical or real-world relation between these features.
>
>
> >    The number of subspaces is strictly bound by the dimensionality of activation space as they must be exactly orthogonal. This might limit the method's potential and interpretability. As we see with SAEs, representation space can be well-approximated with tens of Millions of features and having only few hundred feature groups available at most might be quite limiting, and many different features might be packed into the same subspace (because different features are often mutually exclusive as well). This would limit interpretability. My concern is that in reality, the independence-between-feature-groups assumption doesn't hold 100% and we would be better off with relaxing this restriction a little to allow more feature groups to exist and be more interpretable. A possible approach here could be to use MOLTs (Lindsey, Anthropic 2025).
>
>
> Yes, we agree the strict orthogonality between subspaces is a constraint. We also agree with the point that independence does not hold 100%. Empirically, to make things work, managable, and optimizable, we need some constraints, and found orthogonality to work well. Also, as we showed, our method makes subspaces as independent as possible, and works without requiring 100% underlying independency. And thanks for mentioning possible approaches, we have been following this line of research as well and will consider it.

---

> ### Author Response · Authors · 2025-11-17
> **(3/3)**
>
> >    The authors heavily criticize superposition and SAEs but they never directly compare against SAEs although direct comparisons should be possible. They state that other baseline comparisons aren't possible because their subspaces are learned unsupservised from activations, but SAEs are as well. In fact, I think that their method doesn't disagree with superposition and sparsity at all.
>
> To clarify, we do not criticize superposition itself, but rather try to discover more structure for superposition. Our arguments rely on the same conceptual framework (i.e., assuming there exist underlying true features) as superposition and SAEs papers.
>
>
> Regarding direct comparison with SAEs, please refer to the global reply, where we describe why meaningful direct comparison is not possible, and show a solution to adapt SAEs to our evaluation and the results.
>
> We fully agree that our method doesn't fundamentally disagree with superposition and sparsity, we are only trying to emphasize different aspects. We slightly changed the tone of the paper in the new version to avoid such misunderstanding.
>
>
>
> ### Other things that would improve the paper:
>
> >    Insight into LLM computation. This paper mainly proposes a new method and validates it but there's no new mechanistic insight about LLM computation. It would improve the paper a lot if the authors could prove by example that this method can discover things that other methods like SAE, DAS, etc can't
>
> Thanks for the suggestion, we agree. We leave this as future work. Nevertheless, the paper indeed include some interesting findings about LLM computation. The major things is that this decomposition is even possible, i.e., LLM do have "natural" meaningful subspaces, which is different from DAS. There are also some small findings scattered in Appendix, e.g., moving knowledge by attention is not done at once, this process is repeated and distributed across layers (Appendix E.4).
>
> >    More experiments with LLMs. Qualitative examples in Figure 2 lack rigor or quantification. IOI/greater than results don't measure logit difference recovered when only patching an individual subspace, more evidence that subspaces are monosemantic and interpretable, etc.
>
> Thanks for the suggestion. We agree that additional experiments could further enrich the study. At the same time, we have aimed to include a broad and complementary set of experiments in the current version, with each addressing limitations of the others. This already makes the paper quite dense, and we have had to move a substantial amount of material to the appendix to stay within the space constraints.
>
>
> ## Reply to Questions:
>
> >    Many things are mutually exclusive. For example, if the text is about software licenses, it's not about a fiction novel or a mental health consultation. Features from all these different contexts could be squished into a single subspace. Especially since the number of subspaces is extremely limited. Do you observe this in practice? Doesn't this imply that mutual exclusiveness != interpretability? Doesn't this hurt interpretability of those subspaces and the applicability of this method a lot?
>
> Yes, we do observe this in practice, e.g., each of Figure 20-23 show some mutually exclusive concepts within the same subspace. In your example, such strong mutual exclusiveness actually enhances the interpretability of the subspace, making it possible to summarize the subspace as describing the "topic" of the text. The fact that the subspace can encode different topics depending on the topic in the input is exactly what makes it useful and act like a "variable". Importantly, we argue that real and strict mutual exclusiveness implies that concepts are somehow related, allowing us to extract the high-level commonality linking them.

---

### Official Review · Reviewer_rZmX · 2025-11-03

**Soundness:** 3
**Presentation:** 4
**Contribution:** 3
**Rating:** 6
**Confidence:** 2

**Summary:**

The paper describes and evaluates an unsupervised method — neighbor distance minimization — for decomposing representation space into interpretable subspaces.

**Strengths:**

**Well-written with clear explication:** The paper presents complex concepts with clear language, helpful intuitions, and useful examples.

**Vast amount of work:** The paper, combined with the appendices, reflects a vast amount of experimental work.

**Weaknesses:**

**Evaluation**: As the authors say, “The key question is NDM’s applicability to real-world neural models.” The evaluation relies on the intuition that “when processing inputs, key intermediate results should ideally lie in a single subspace.” On the one hand, this idea is appealing, partially because it provides a fairly clear evaluation criterion for methods for decomposing representation space into interpretable subspaces. On the other hand, this criterion could both under under-determine and over-determine useful results. That is, essentially useless decompositions could satisfy this criterion, and useful decompositions could violate it. Ultimately, a more convincing evaluation would use an end-to-end criterion (i.e., one that shows that an MI pipeline produces more useful results when it includes NDM rather than some other subspace discovery approach). That’s a very tall order, but one that is far less error-prone than this paper’s current evaluation criteria. This reflects a more general problem with current MI research: whole MI pipelines cannot be easily created because we don’t have all the components, and candidate components cannot easily be created because we don’t have the pipeline.

**Questions:**

In Section 5.1, you say that “The results of NDM using the best hyperparameters are shown in Table 1…”  This raises the possibility of overfitting because only the “best” hyperparameters are shown. How were the best hyperparameters selected, and is this method reasonable in practice?

---

> ### Author Response · Authors · 2025-11-17
> **(1/2)**
>
> Thanks for your thoughtful feedback!
>
> ## Reply to Weaknesses
> > Evaluation: As the authors say, “The key question is NDM’s applicability to real-world neural models.” The evaluation relies on the intuition that “when processing inputs, key intermediate results should ideally lie in a single subspace.” On the one hand, this idea is appealing, partially because it provides a fairly clear evaluation criterion for methods for decomposing representation space into interpretable subspaces. On the other hand, this criterion could both under under-determine and over-determine useful results. That is, essentially useless decompositions could satisfy this criterion, and useful decompositions could violate it. Ultimately, a more convincing evaluation would use an end-to-end criterion (i.e., one that shows that an MI pipeline produces more useful results when it includes NDM rather than some other subspace discovery approach). That’s a very tall order, but one that is far less error-prone than this paper’s current evaluation criteria. This reflects a more general problem with current MI research: whole MI pipelines cannot be easily created because we don’t have all the components, and candidate components cannot easily be created because we don’t have the pipeline.
>
> First of all, in the reply, we use MI to refer to Mutual Information, instead of referring to Mechanistic Interpretability, which (we believe) is the meaning of MI in your review.
>
> We agree that useful decompositions might sometimes violate our basic criterion, because of a mismatch in granularity. For example, an entity name and the knowledge about the entity might be in one or two subspace; either way would be a reasonable and useful partition. The evaluation might test whether they are contained in one subspace. However, we think this is largely mitigated by testing different hyperparameters (e.g., MI threshold), which partition the space with different granularities, and selecting the best one.
>
> On the other hand, to be honest, we do not agree that a useless decomposition could satisfy this criterion. In the evaluation, we also measure inequality between normalized effects (columns $d_s$ and $Var_s$). If a single subspace contains *all* the key intermediate variables, it will unavoidably occupy most of the dimension and variance of the whole space, resulting low inequality of normalized effect. If a decomposition perfectly satisfy the evaluation metric, i.e., very high inequality between subspaces with or without normalization, it means that the tested key intermediate variable lies in a single subspace, while other variables or information do not. This decomposition thus would perfectly disentangle different types of information. We can further analyze which downstream components operate in this subspace to understand their function.  For example, Figure 20 shows qualitative examples for a subspace with high value in all 3 columns (0.89, 0.89, 0.90 for - unnormalized, $d_s$ normalized by dimension, and $Var_s$ normalized variance respectively). We also encourage you to use our web application to check more examples and see if they also contain other information.
>
> We agree that an end-to-end criterion would be more ideal. If NDM played a crucial role in a full Mech Interp pipeline, that would be a very strong result and indeed a tall order. We also share your view on this general challenge for Mech Interp research. We believe that, by creating more fundamentally different and interesting methods, we can contribute to the longer-term search for an ideal full Mech Interp pipeline.

---

> ### Author Response · Authors · 2025-11-17
> **(2/2)**
>
> ## Reply to Questions
> > In Section 5.1, you say that “The results of NDM using the best hyperparameters are shown in Table 1…” This raises the possibility of overfitting because only the “best” hyperparameters are shown. How were the best hyperparameters selected, and is this method reasonable in practice?
>
>
> We don't think the concept of overfitting applies here: If we obtain a good decomposition by selecting the best among multiple ones, that means the method is useful. If you are thinking about sensitivity to hyerparameters, we have sections in Appendix D.5 showing results under different hyperparameters; the one shown in Table 1 is the best among them. As you can see, even though different hyperparameters produce different results, NDM performs significantly better than the baselines across them.

---

### Author Response · Authors · 2025-11-17
**Global Reply Regarding SAEs (1/2)**

We thank all reviewers for their thoughtful reviews and constructive feedback!

We notice that some concerns are raised by multiple reviewers, especially about sparse autoencoders (SAEs). We summarize and address them in this global comment, so this is a shared reply for all reviewers. We have updated the paper and added these new discussions and experiments (Appendix F).

> ### Why not compare with SAEs?

**In quantitative evaluation**, we patch each subspace and compute its average effect on the output. SAEs do not provide such subspaces directly. Comparing NDM with SAEs thus requires a method for applying an analogous evaluation to SAEs. One way is to patch the basis elements that SAEs provide, the features. This means that we replace a feature's activation value with its value on the corrupted input. For example, in the clean input on the token after "John", a feature representing *"previous token is John"* is highly activated -- while, in the same position in the corrupted input, a feature representing *"previous token is Mary"* is highly activated. If we patch the "prev-Mary" feature, the new activation would say *"previous token is both John and Mary"*. If we patch the "prev-John" feature, the new activation has no information about previous token. In other words, we need to add a feature while removing another one (and it's hard to select which one since there are multiple active features), which requires understanding of feature dependency that SAEs do not provide ("prev-Mary" and "prev-John" are mutually exclusive). More importantly, patching the "prev-John" feature would not have any real effect on inputs where the names are not John. In other words, conceptually, one would expect such patching to only have effect on a few specific inputs, and only to confuse the model. In contrast, patching subspace can be thought of patching meaningful feature *groups*. Moreover, applying such patching to SAEs requires running evaluation on each of the features, which is too computationally expensive due to the large amount of features.

Because of these fundamental differences, we did not compare with SAEs in the original submission. Nevertheless, multiple reviewers are wondering how SAEs would perform under the same evaluation used in the paper. To test this, we derived subspaces from SAE features. We use the method introduced in [1] (Sec. 4), where the authors cluster feature vectors and then, for each cluster, find multi-dimensional subspaces spanned by activations reconstructed only with features in this cluster. Specifically, they inspect pairs of principal components of the cluster-specific reconstructions and manually determine the most interesting pairs. We adapt this by producing one subspace for each cluster, spanned by the principal components with the largest eigenvalues. To keep the subspaces orthogonal to each other, we first sort the clusters by their average cosine similarity or their size in descending order, then iteratively project each cluster on the orthogonal complement of the already-selected dimensions before finding the eigenvectors.

In sum, we follow the original method from [1] as much as we can, while adapting it to construct a full decomposition of the space. The details are in the updated version of the paper (Appendix F.2). We test this method on the GPT-2 test suite, with many different hyperparameter configurations. The full results can be found in the updated paper (Table 8). Here we show the results of the best performing configuration, denoted by "Feature Clusters"

(to be continued)

---

> ### Author Response · Authors · 2025-11-17
> **Global Reply Regarding SAEs (2/2)**
>
> | Method    | **test 1** |      |       | **test 2** |      |       | **test 3** |      |       | **test 4** |      |       | **test 5** |      |       | **Avg** |
> |-----------|------------|------|-------|------------|------|-------|------------|------|-------|------------|------|-------|------------|------|-------|---------|
> |           | -          | ds   | Var_s | -          | ds   | Var_s | -          | ds   | Var_s | -          | ds   | Var_s | -          | ds   | Var_s |         |
> | Identity  | 0.33       | 0.05 | 0.23  | 0.32       | 0.11 | 0.25  | 0.40       | 0.12 | 0.19  | 0.31       | 0.04 | 0.15  | 0.32       | 0.12 | 0.19  | 0.21    |
> | Random    | 0.36       | 0.10 | 0.11  | 0.36       | 0.16 | 0.16  | 0.32       | 0.11 | 0.12  | 0.33       | 0.10 | 0.12  | 0.39       | 0.16 | 0.18  | 0.21    |
> | PCA 1     | 0.43       | 0.42 | 0.37  | 0.46       | 0.22 | 0.52  | 0.50       | 0.30 | 0.51  | 0.38       | 0.22 | 0.36  | 0.35       | 0.24 | 0.36  | 0.38    |
> | PCA 2     | 0.66       | 0.56 | 0.39  | 0.26       | 0.19 | 0.24  | 0.28       | 0.19 | 0.29  | 0.40       | 0.20 | 0.19  | 0.40       | 0.26 | 0.27  | 0.32    |
> | Feature clusters       | 0.60       | 0.44 | 0.60  | 0.61       | 0.41 | 0.67  | 0.73       | 0.39 | 0.54  | 0.74       | 0.31 | 0.54  | 0.71       | 0.56 | 0.61  | 0.56 |
> | NDM       | 0.89       | 0.89 | 0.90  | 0.71       | 0.72 | 0.83  | 0.63       | 0.70 | 0.38  | 0.50       | 0.55 | 0.79  | 0.72       | 0.67 | 0.78  | **0.71** |
>
>
> Overall, while the SAE-based feature clusers outperform the baselines, NDM still performs better overall.
>
> **In qualitative evaluation**
>
>
> Our qualitative evaluation has two aspects: (i) interpretability of preimages for subspaces, (ii) consistency of the type of information encoded in a subspace across inputs (thus, interpretability of the overall subspace).
> Regarding (i), we think that a single SAE feature would have interpretability comparable to a single subspace preimage, as we know that they are interpretable most of the time but also some features do not have clear and easy interpretation.
> However, regarding (ii), this is not applicable to SAEs, as they provide no clear structure between features; rather, finding encodings of the same type of information requires inspecting each feature individually. In contrast, information in the same subspace found by NDM is of the same "type" across inputs most of the time. Inspecting some of the them can give an idea of the whole feature group, significantly reducing the amount of interpretation work needed.
>
>
> > ### What about training SAEs within subspaces?
>
> Conceptually, if one trained a top-1 SAE (top-k SAE with k=1) within each subspace, we expect that the features would be the cluster centers in the subspace. Inspecting a feature means to find inputs whose activation in the subspace is close to the feature. This is roughly the same as inspecting preimages in InversionView, with query activations taking the role of the "feature". Thus, InversionView can conceptually be thought of as a "parameter-free" top-1 SAE.
>
> [1] Engels, Joshua, et al. "Not all language model features are one-dimensionally linear." arXiv preprint arXiv:2405.14860 (2024).

---

### Author Response · Authors · 2025-11-30
**Summary for AC (2/2)**

4.  **Evaluation Metric and Monosemanticity**
    Reviewers were concerned that our evaluation might not guarantee useful or monosemantic subspaces.
    - MPxT: *"if only a single subspace were used, the condition that the high-level 'variable' lies within the same subspace (L. 261) would trivially hold."*
    - rZmX: *"This criterion could both under under-determine and over-determine useful results. That is, essentially useless decompositions could satisfy this criterion, and useful decompositions could violate it."*
    - EMRg: *"The authors for some tasks that most task-relevant features land in a single subspace but they don't show that this subspace only contains those features, aka is interpretable and monosemantic."*

    Response: We clarified that our evaluation metrics are designed to penalize non-monosemantic subspaces (*"If a single subspace contains all the key intermediate variables, it will unavoidably occupy most of the dimension and variance... resulting low inequality of normalized effect."*) The columns $d_s$ and $Var_s$ specifically measure this (*"where the effects are normalized by subspace dimension and variance"*). Moreover, monosemanticity can also be checked in our qualitative examples.

5. **Constraint of Orthogonality and Limited Granularity**
    - EMRg: *"The number of subspaces is strictly bound by the dimensionality of activation space... This might limit the method's potential and interpretability."*
    - bKFF: *"By construction, NDM can only identify as many subspaces as there are dimensions in the latent space... it is unclear how granular these 'variables'... are."*

    Response: We acknowledged this limitation (*"Empirically, to make things work, managable, and optimizable, we need some constraints, and found orthogonality to work well"*). We also acknowledged that *"the subspaces we currently find are usually big ones, and it will be interesting to find more fine-grained ones."* But we also noted that some "variables" inherently require large subspaces due to their many possible values, and that our method identifies these meaningful, coarse-grained groups.


6.  **Points raised by single reviewers**

    - EMRg: *"'Mutual exclusiveness' doesn't seem like a more fundamental condition, or much different from sparsity at all... So no new theory is needed... I don't understand how 'mutual exclusiveness and feature groups' are fundamentally different from 'sparsity and superposition'."*

    Response: We clarified that we emphasize a different view, i.e., non-uniform vs. uniform, instead of inventing a new theory and added clarification in the paper (*"We do not say that 'mutual exclusiveness and feature groups' is a new theory over 'sparsity and superposition'... The difference is that the concept 'sparsity' treats features uniformly... We would like to emphasize a non-uniform view under which some sets of features can be more orthogonal/superposed than others because of non-uniform dependency between features."*)

    - EMRg: *"This paper mainly proposes a new method and validates it but there's no new mechanistic insight about LLM computation."*

    Response: We highlighted that the mere existence of these "natural" subspaces is itself a finding, and we said that there are scattered mechanistic insights in the Appendix where we inspect subspaces more closely.

    - MPxT: *"I did not find the qualitative examples in Figure 2 particularly convincing."*

    Response: We clarified that Figure 20-23 are the primary illustrations of subspaces as variables, while Figure 2 is showing what kind of variables we can have (*"Our key point here is that related concepts are in the same subspace. So actually Figure 20-23 would be more important qualitative examples to illustrate this idea."*). We also added Figure 30 to better illustrate positional encoding.

    - bKFF: *"Lack of Details on the Computational Aspect of NDM... raises questions about the practicality."*

    Response: We clarified that we do have the information in the appendix of the old version, we also refer to the section in main paper in the new version. In short, *"the compute usage is roughly at the same level as sparse autoencoders (SAEs)."*

    - bKFF: *"It would also be important to evaluate the effectiveness of exploring image models"*

    Response: We acknowledged this limitation, while also emphasizing that we have 4 different evaluations.

---

### Author Response · Authors · 2025-11-30
**Summary for AC (1/2)**

To aid the area chair, we summarize concerns raised by reviewers and our responses. We also refer to the original text so that one can search them with string matching to see the original comment and our complete reply.


1.  **Direct Comparison with Sparse Autoencoders (SAEs)**
    This is a frequent concern raised by EMRg, MPxT, and bKFF.
    - EMRg: *"they never directly compare against SAEs although direct comparisons should be possible."*
    - MPxT: *"In Table 1, it is unclear why no comparison to sparse autoencoders (SAEs) is included. A theoretical justification would be sufficient..."*
    - bKFF: *"Although the intuitive differences... are provided, no practical comparison is made."*

    Response: In the global reply (under "why not compare with SAEs?"), we explained why direct comparison is not meaningful. Nevertheless, due to the strong interest from reviewers, we adapted our evaluation method to SAEs and derive subspaces from them to enable such comparison. This is the new row "Feature clusters" in the Table of the global reply, the full details and results are added in the new Appendix F. We see that our method, NDM, still outperforms baselines. We also explained why, qualitatively, they are not comparable — because one of the two criteria, consistency of the type of information, is not applicable to SAE features (*"regarding (ii), this is not applicable to SAEs, as they provide no clear structure between features"*).


2. **Extracting Features from Subspaces and Testing Mutual Exclusiveness**
    - MPxT: *"...it is not clear why we should expect these subspaces to be inherently interpretable if superposition still occurs within them. Wouldn’t we, in many cases, need an additional method to disentangle or remove the remaining superposition?"*
    - EMRg: *"The authors assume that features within a subspace are mutually exclusive but you never test this (in fact, you don't even extract features within a subspace)."*
    - EMRg: *"The authors for some tasks that most task-relevant features land in a single subspace but they don't show that this subspace only contains those features, aka is interpretable and monosemantic."*
    - EMRg: *"Many things are mutually exclusive... Do you observe this in practice? Doesn't this imply that mutual exclusiveness != interpretability? Doesn't this hurt interpretability of those subspaces and the applicability of this method a lot?"*


    Response: We first explained why inspecting preimages with InversionView is roughly like inspecting features of a top-1 SAE (see "what about training SAEs within subspaces?"). Therefore, the qualitative examples in the paper, e.g., Figures 20-23, can be regarded as observing features within a subspace. They indeed show mutually exclusive concepts within the same subspace. We also argue that mutual exclusiveness itself implies a relation (*"In your example, such strong mutual exclusiveness actually enhances the interpretability of the subspace, making it possible to summarize the subspace as describing the 'topic' of the text."*).


3.  **Hyperparameter Selection and Sensitivity**

    - MPxT: *"The method relies on the mutual information (MI) threshold, but it is not clear how this threshold should be selected in practice."*
    - bKFF: *"How sensitive is NDM to the provided N model activations?"*
    - bkFF: *"...there is a large variability in subsequent performance"*
    - rZmX: *"How were the best hyperparameters selected, and is this method reasonable in practice?"*

    Response: First of all, we clarified that the variance in evaluation metric under different hyperparameters does not necessarily mean variance in quality or interpretability (*"variability in the Gini score can just reflect mismatch in granularity of information partition to the evaluation, not that some decompositions are inherently better than others... Hence, we consider different levels of granularity by changing the hyperparameters, and report the best ones for both NDM and SAE baselines. This can be viewed as the score obtained when the decomposition granularity is best matched to the evaluation suite."*) This motivates us to test different MI thresholds. Meanwhile, we acknowledged that *"some hyperparameter searching is needed"*. Nevertheless, we observe overall strong performance across different hyperparameters (NDM: 0.52-0.71, Feature Clusters: 0.31-0.56, other baselines: 0.21-0.38). Regarding computational cost, we said that *"the widely used SAE approach faces similar challenges (even worse in terms of the number of parameters to be trained), which we think may be inherent to unsupervised methods."*

---

### Meta-Review · Area_Chair_oyB9 · 2025-12-10

**Summary:**

This study proposes a method for decomposing language model activations into more interpretable representations. The method, neighbor distance minimization (NDM), is founded on an assumption that follows from the hypothesis of superposition: features will occur in shared subspaces primarily when they are mutually exclusive. Experiments with GPT-2 (and later experiments with 2B-parameter models) show that these subspaces and the features therein are often interpretable.

Reviewers largely agree that the core ideas are interesting, novel, and well-presented. The breadth of empirical evidence is also admirable. Shared concerns largely centered around a lack of comparison with sparse autoencoders and some concerns around the stability of results with respect to certain hyperparameters. In my view, the first has been sufficiently addressed during the discussion period. Some concerns around the computational expense of hyperparameter tuning and the sensitivity of results to these hyperparameters remain, although I agree with the authors that these may be inherent to unsupervised methods in this space—at least in the current state of the field. Nonetheless, there is an emergent consensus that these findings could be of interest to the ICLR community.

**Reviewer Concerns:**

The shared concerns around comparisons to sparse autoencoders have been sufficiently addressed, both via empirical evidence and a discussion of the non-comparability of SAEs with the proposed method.

Concerns around the stability of results were shared across reviewers. This point was addressed by the authors during the discussion period, but reviewers stated that their primary concerns were not entirely addressed by the revisions and responses.

Overall, I think the authors have done a good job of responding to these concerns, as well as the reviewer-specific concerns and clarifications.

**Reviewer Scores:**

Many reviewers have given decently positive scores. One said that they had changed their score from a 4 to a 6. Another stated that they still had reservations, and would not change their score from a 4. The two remaining reviewers did not have a chance to respond, but their scores were initially positive—and I have found the responses to these reviews largely satisfactory.

---

### Decision · Program_Chairs · 2026-01-26

Accept (Poster)